# Increasing soil nitrous acid emissions driven by climate and fertilization change aggravate global ozone pollution

Yanan Wang[1,5], Qinyi Li [1,2,5], Yurun Wang[1], Chuanhua Ren[1,3], Alfonso Saiz-Lopez [4], Likun Xue[2] & Tao Wang [1] ✉

Soil microbial nitrous acid (HONO) production is an important source of atmospheric reactive nitrogen that affects air quality and climate. However, long-term global soil HONO emissions driven by climate change and fertilizer use have not been quantified. Here, we derive the global soil HONO emissions over the past four decades and evaluate their impacts on ozone ($O_3$) and vegetation. Results show that climate change and the increased fertilizer use enhanced soil HONO emissions from 9.4 Tg N in 1980 to 11.5 Tg N in 2016. Chemistry-climate model simulations show that soil HONO emissions increased global surface $O_3$ mixing ratios by 2.5% (up to 29%) and vegetation risk to $O_3$, with increasing impact during 1980s-2016 in low-anthropogenic-emission regions. With future decreasing anthropogenic emissions, the soil HONO impact on air quality and vegetation is expected to increase. We thus recommend consideration of soil HONO emissions in strategies for mitigating global air pollution.

Nitrous acid (HONO) is a crucial precursor of the highly reactive hydroxyl radical (OH), the dominant oxidant responsible for removing climate- and air quality-relevant gases released into the atmosphere[1–3]. It also plays a key role in the chemical production of ozone ($O_3$) and secondary particulate matter in the polluted regions[4]. $O_3$ pollution has become a worldwide environmental concern because it threatens human health, terrestrial vegetation, and crop production[5,6] and also contributes to global warming[7]. With the decrease in anthropogenic emissions in recent decades, the contribution of natural sources, such as soil emissions, to air pollution and $O_3$ formation has become increasingly significant, particularly in rural and agricultural areas under $NO_x$-limited regimes[8–10].

Soil covers almost the entire Earth's land surface[11], providing crucial services for humans and ecosystems. Soil microbial activities and agricultural management practices, notably fertilizer use, release various gases from soil into the atmosphere, which adversely affect human health, climate, and ecosystems[12–14]. Previous studies on soil

nitrogen (N) emissions mainly focused on nitrous oxide ($N_2O$), nitric oxide (NO), and ammonia ($NH_3$) emissions[15–17]. In recent years, a growing number of studies have demonstrated that soil emissions of HONO contribute 17–80% of atmospheric HONO mixing ratios, with significant impacts on secondary air pollutants such as particulate nitrate and atmospheric $O_3$[18–24]. Soil HONO emissions are largely influenced by the soil water content (SWC), soil temperature (ST), and fertilization[18,19,25,26]. The continuous increase in ST[27,28] and changes in SWC[29] induced by climate change may lead to alterations in soil HONO emissions. Moreover, a growing population and increasing food demand have resulted in a nearly two-fold increase in global N fertilizer usage in the last four decades[30]. However, the long-term trend of soil HONO emissions and their impact on global air quality and vegetation remain unknown.

In this study, we compile global soil emissions of HONO from 1980 to 2016 by establishing a quantitative parameterization scheme that links soil emissions to ST and SWC data for both natural and fertilized

[1]Department of Civil and Environmental Engineering, The Hong Kong Polytechnic University, Hong Kong, China. [2]Environment Research Institute, Shandong University, Qingdao, China. [3]Joint International Research Laboratory of Atmospheric and Earth System Sciences, School of Atmospheric Sciences, Nanjing University, Nanjing, China. [4]Department of Atmospheric Chemistry and Climate, Institute of Physical Chemistry Blas Cabrera, Spanish National Research Council (CSIC), Madrid, Spain. [5]These authors contributed equally: Yanan Wang, Qinyi Li. ✉e-mail: tao.wang@polyu.edu.hk

soils. We incorporate these soil HONO emissions in a global chemistry-climate model, the Community Atmosphere Model with Chemistry (CAM-Chem), to simulate the effects of soil HONO on atmospheric oxidation capacity, air pollution, and vegetation. CAM-Chem simulations reveal that soil HONO emissions significantly enhance the global atmospheric oxidation capacity and $O_3$ mixing ratios, leading to increased damage to vegetation. The impact of soil HONO emissions on $O_3$ has been growing from the early 1980s to 2016 in regions with low anthropogenic emissions. This finding highlights the urgent need for managing soil HONO emissions to mitigate their adverse impacts on air quality and ecosystems.

## Results and discussion

### Spatio–temporal variations in soil HONO emissions

Soil HONO emissions are influenced by soil properties, land-use types, and climatic zones[19,25,26]. We integrated global measurements of soil HONO emissions (Table S4) and developed a parameterization scheme for these emissions based on ST, SWC, and fertilizer use across different regions and land-use types (Methods). Combining this parameterization and the Modern-Era Retrospective Analysis for Research and Applications Version 2 (MERRA2) reanalysis dataset, we quantified the global soil HONO emissions from 1980 to 2016. The total annual soil emissions of HONO in 2016 were estimated to be 13.4 Tg N yr$^{-1}$ without considering canopy reduction. The soil NO emissions estimated using the same method were 7.3 Tg N yr$^{-1}$. We calculated the canopy reduction factor (CRF) using leaf area index (LAI, see Methods), yielding a global average CRF of 0.79, which falls within the range estimated by Vinken et al. (0.87)[31] and Yan et al. (0.67)[32]. The global distribution of CRF shows that it is lowest in the tropical forests of South America and Africa (Supplementary Fig. S2), indicating a substantial reduction in soil emissions due to the dense canopies in these tropical regions. The application of the CRF decreases the global total soil HONO and NO emissions to 11.5 and 6.2 Tg N yr$^{-1}$, respectively. (Fig. 1 and Supplementary Fig. S1). The estimated soil HONO emission is comparable to the value of 9.7 Tg N yr$^{-1}$ obtained by Wu et al.[24]. The soil NO emissions is slightly higher than the estimate of 5.5 Tg N yr$^{-1}$ calculated by Yienger and Levy[33] using the YL95 model, but lower than the 9.0 Tg N yr$^{-1}$ estimated by Hudman et al. [34] using the Berkeley-Dalhousie Soil NO$_x$ Parameterization (BDSNP), and the 12.9 ± 3.9 Tg N yr$^{-1}$ estimated by Vinken et al.[31] using a top-down approach. Overall, the soil NO emissions in our study fall within the range (4–21 Tg N yr$^{-1}$) reported in previous studies[31,32,34]. Note that the top-down estimate method may includes both soil HONO and soil NO as soil NO$_x$, and our estimate for global soil emissions of HONO + NO totals 17.7 Tg N yr$^{-1}$, which closely aligns with the upper estimate of top-down soil NO$_x$ emission (16.8 Tg N yr$^{-1}$). Our estimates indicate that the global soil emissions of HONO exceed those of NO. As most previous studies on soil emissions of reactive oxidized nitrogen focused on soil NO, they likely underestimated the impacts of soil emissions on atmospheric chemistry and air quality.

Figure 1a illustrates the global distribution of HONO emissions exemplarily for 2016. The emission hotspots were mainly located in agricultural areas in India, eastern China, central northern America, Europe, African savannahs, and South America. The spatial distribution of soil HONO emissions in this study is consistent with the results reported by Wu et al. [24], as well as the spatial variations in global soil NO$_x$ emission[32,34]. Approximately 69% of global soil HONO emissions were concentrated in low-latitude regions (30°S to 30°N), while 28% of the emissions occurred poleward of 30°N. Emissions poleward of 30°S accounted for less than 2% of the global total emissions. The northern hemisphere contributed two thirds of the global emissions, while the southern hemisphere contributed the remaining one third. Asia exhibited the highest soil HONO emissions, accounting for 37.2% of the global emissions, with the contributions of India and China being 1.74 Tg N yr$^{-1}$ (15.1%) and 0.55 Tg N yr$^{-1}$

(4.7%), respectively. The contribution of Africa (3.1 Tg N yr$^{-1}$, 28.2%) was the second highest, followed by that of South America (1.5 Tg N yr$^{-1}$, 8.8%) (Fig. 1b). Rasool et al.[35] and Luo et al.[36] employed Day-CENT (Daily time-step version of the CENTURY biogeochemical model) and FEST-C (Fertilizer Emission Scenario Tool for CMAQ) models to assess soil HONO and NO emissions across the United States. Their assessments were based on the proportion of soil HONO to NO emissions, along with the relationship between soil HONO emissions and SWC established by Oswald et al.[26]. Our estimated annual emissions of soil reactive oxidized nitrogen (HONO + NO) in the United States is 0.85 Tg N yr$^{-1}$, which is similar to the value in Luo et al. [36] (0.69 Tg N yr$^{-1}$). Besides, our estimated HONO emissions from soils in North America is 0.90 Tg N yr$^{-1}$, which is close to the estimate of 0.83 Tg N yr$^{-1}$ provided by Wu et al.[24].

Soil HONO emissions are influenced by changes in pedoclimatic conditions and fertilization and, thus, exhibit significant temporal variations. In the short term, soil HONO emissions were observed to vary seasonally. The globally highest emissions occur in July and peak during the summer seasons (June to August in the northern hemisphere and December to February in the southern hemisphere), while the lowest emissions were observed during the winter (Fig. 1d–g, Supplementary Fig. S3). These seasonal variations were primarily caused by differences in ST throughout the year and disparities in crop growing seasons and fertilizer application timing between the northern and southern hemispheres. The peak values of ST in the northern and southern hemispheres were observed in July and January, respectively, with seasonal average temperatures during their respective summers reaching 19.2 °C and 24.0 °C and dropping to 3.2 °C and 14.8 °C in winter (Supplementary Fig. S4). Additionally, fertilization practices in the northern hemisphere are predominantly conducted between March and November, while in the southern hemisphere, such practices are typically implemented from August to March (Supplementary Table S1). Hence, the seasonal variations in soil HONO emissions are influenced by differences in climatic factors and the timing of agricultural activities.

The combined effects of changes in climate and nitrogen fertilizer usage also drive long-term trends in soil HONO emissions. During the period from 1980 to 2016, global soil HONO emissions consistently increased at a rate of 62.9 Gg N yr$^{-1}$ yr$^{-1}$ (0.7% yr$^{-1}$) (Fig. 2a, b). The hotspots for this growth were similar to those of soil HONO emissions, concentrated in agricultural areas in India, eastern China, central northern America, South America, and Africa (Fig. 2a). Africa exhibited the highest growth rate at 21.7 Gg N yr$^{-1}$ yr$^{-1}$ (0.9% yr$^{-1}$), followed by India (8.7 Gg N yr$^{-1}$ yr$^{-1}$, 0.7% yr$^{-1}$) (Fig. 2b). The response of soil HONO emissions to changes in fertilizer application rates and climate varied across regions. Figure 1c shows that the soil emissions during the fertilized and unfertilized periods were 1.13 and 10.39 Tg N yr$^{-1}$, respectively, corresponding to 9.8% and 90.2% of the total emissions, respectively. These proportions are consistent with ratios reported in previous studies[32]. The proportions of soil emissions during the fertilized and unfertilized periods varied across different countries or regions due to differences in fertilizer application rates. The regions with the highest grain production, i.e., China, India, and North America, were characterized by the largest consumption of fertilizers (Fig. 3f). Fertilizer-induced soil HONO emissions in China contributed with 44.5% of the total emissions (Supplementary Fig. S5). India had the second-highest proportion of soil HONO emissions attributable to fertilization, accounting for 25.3%, followed by North America (9.7%). These results indicate that the variation in fertilizer usage considerably affects the long-term trends of soil HONO emissions in China, India, and North America, while the trends in other regions are primarily influenced by changes in ST and SWC resulting from climate change.

Global fertilizer usage has exhibited a consistent increase over the past four decades, with a growth rate of 1.0196 Tg N yr$^{-1}$ (Fig. 3e, f),

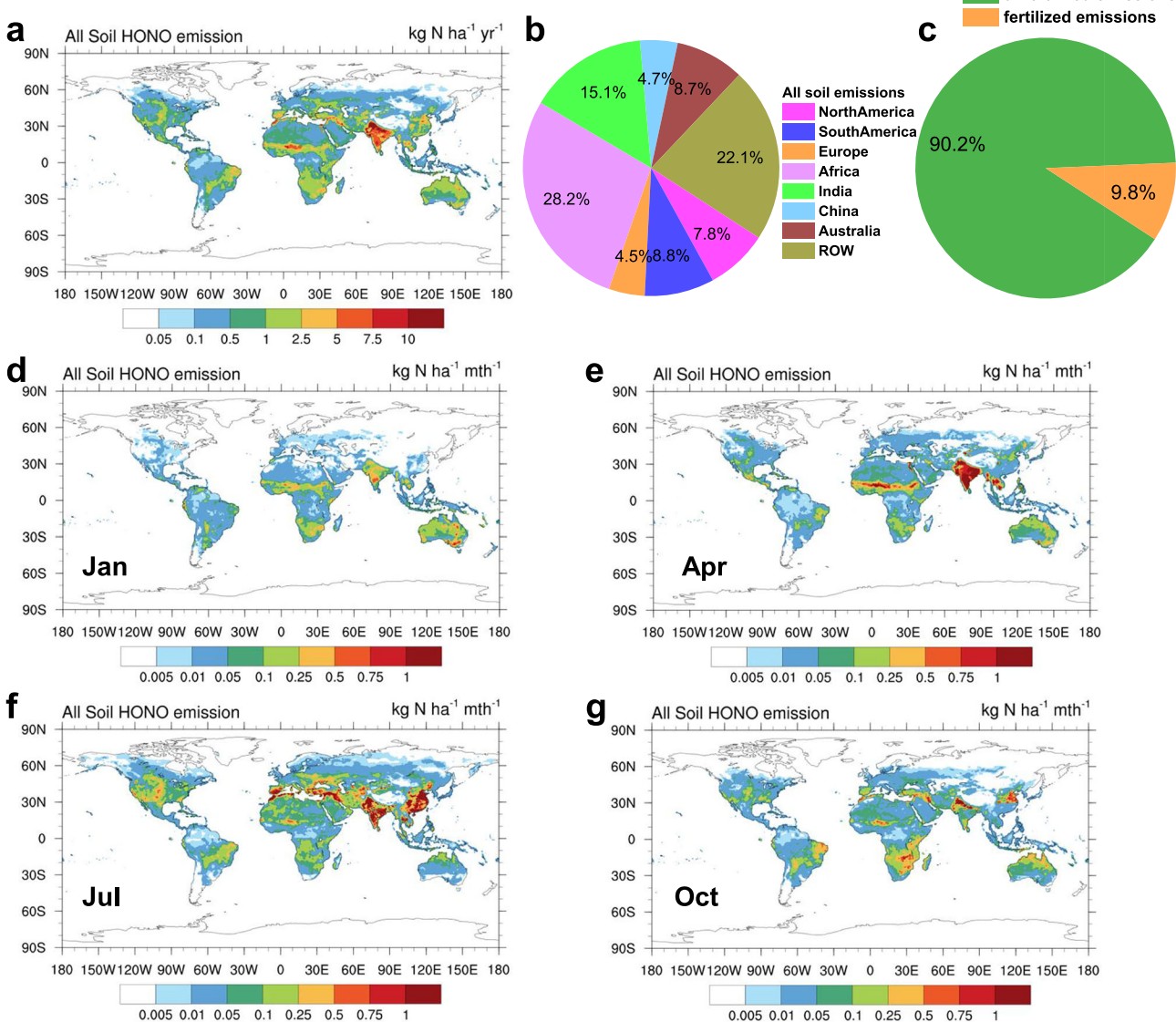

**Fig. 1 | Global soil HONO emissions. a** Global annual average soil HONO emissions in 2016, in units of kg N ha⁻¹ yr⁻¹. **b** Proportions of emissions from different regions. Magenta, blue, orange, purple, green, light blue, brown, and gold sectors indicate proportions of North America, South America, Europe, Africa, India, China, Australia, and the rest of the world (ROW), respectively. **c** Relative contributions of fertilized and unfertilized soil to the total HONO emissions, with orange and green sectors indicating their respective proportions. **d–g** Monthly variation of soil HONO emissions in 2016, in units of kg N ha⁻¹ mth⁻¹.

leading to a rise in global fertilized soil HONO emissions at an average rate of 28.6 Gg N yr⁻¹ yr⁻¹ (Fig. 2c, d). Among the three regions with the highest fertilizer usage (China, India, and North America), the growth rates of fertilizer usage were 0.2706, 0.3629, and 0.1065 Tg N yr⁻¹ yr⁻¹, respectively. This resulted in the highest growth rates in soil HONO emissions after fertilization in these three regions, amounting to 10.1 Gg N yr⁻¹ yr⁻¹ in India, 6.9 Gg N yr⁻¹ yr⁻¹ in China, and 0.9 Gg N yr⁻¹ yr⁻¹ in North America.

Variations in ST and SWC driven by climate change led to changes in the emissions from unfertilized soils. Global ST increased at a rate of 0.018 °C yr⁻¹ from 1980 to 2016. Although the magnitude of this rise varied across regions, a discernible upward trend was observed (Fig. 3a, b). The continuous increase in ST increased the soil HONO emissions. Unlike ST, SWC declined worldwide at a rate of 0.016% yr⁻¹, with trends varying across regions (Fig. 3c, d). SWC increased in Australia and India but continuously decreased in other regions. Laboratory experiments have consistently demonstrated that soil HONO emissions initially rise and then decline as SWC

increases, peaking when the SWC is below 40 WHC%[19,26]. Notably, only Australia exhibited an average SWC below 40 WHC% (Fig. 3e), indicating that an increase in SWC in Australia led to increased soil HONO emissions, whereas the increased SWC in India resulted in reduced unfertilized soil HONO emissions (Fig. 2e, f). Conversely, the decreasing SWC in other regions resulted in increasing soil HONO emissions. Africa demonstrated the highest growth rate at 20.4 Gg N yr⁻¹ yr⁻¹, followed by South America (7.2 Gg N yr⁻¹ yr⁻¹). In summary, the combined effects of changes in fertilizer usage, and of SWC and ST caused by climate change have resulted in an increase of 2.1 Tg N in global soil HONO emissions from 9.4 Tg N in 1980 to 11.5 Tg N in 2016.

## Influence of soil HONO emissions on global air quality
We quantitatively assessed the influence of soil HONO emissions on air quality in 2016 using the CAM-Chem model. In the noSoilHONO case, the traditional sources of HONO were considered, including direct emissions from traffic and biomass burning, homogeneous reactions

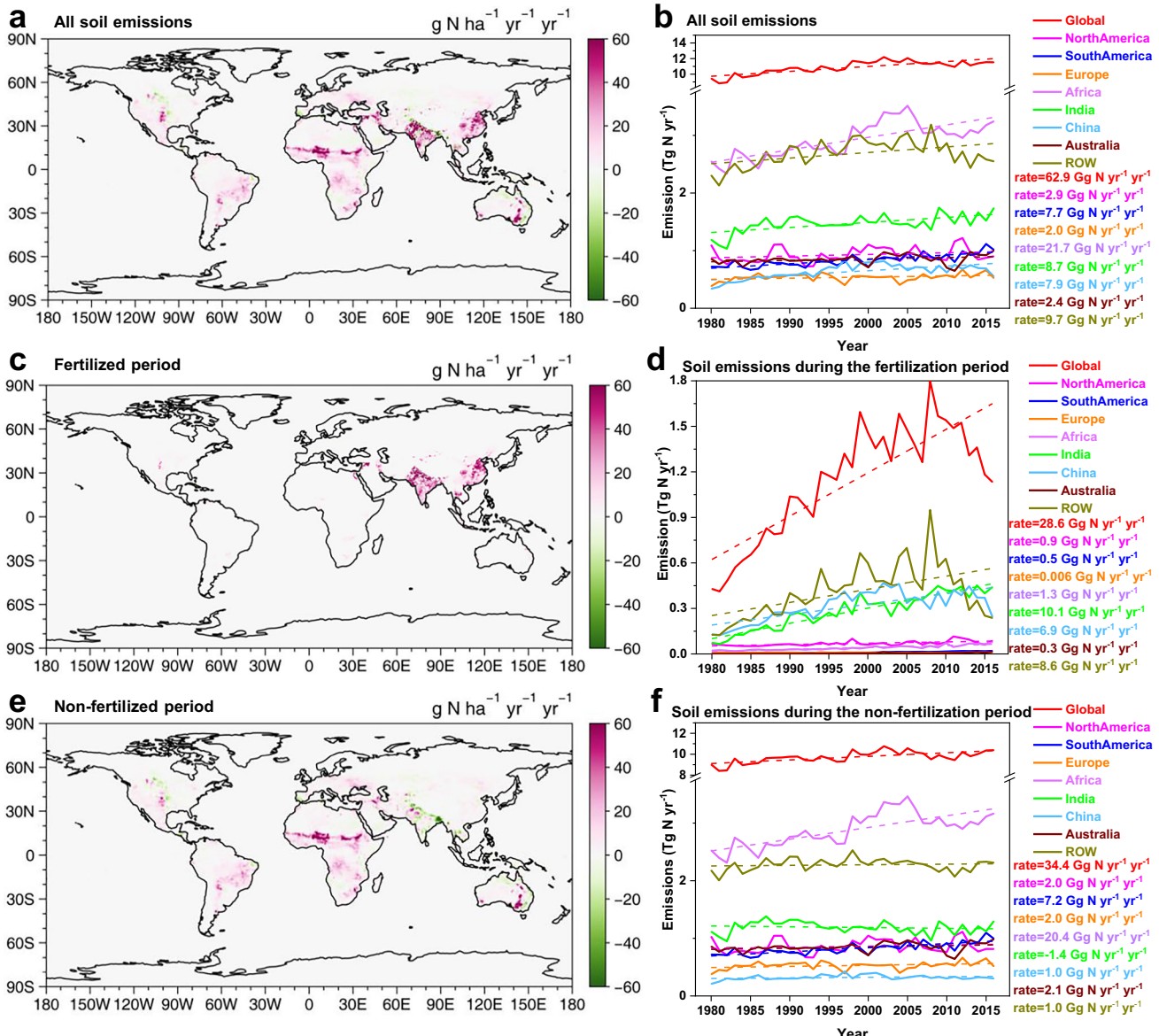

**Fig. 2 | Rates of change in total, fertilized, and unfertilized soil HONO emissions from 1980 to 2016. a, c, e** Rates of change (g N ha⁻¹ yr⁻¹ yr⁻¹) in total soil HONO emissions, fertilization- related emissions, and emissions during non-fertilization period, respectively. The figures display the results of linear regression analysis conducted on emission data for each grid from 1980 to 2016. The slopes of the regression are visually presented in the figures. **b, d, f** Time-series of soil HONO emissions for each region (Gg N ha⁻¹ yr⁻¹ yr⁻¹). Magenta, blue, orange, purple, green, light blue, brown, and gold solid lines represent emissions pertaining to North America, South America, Europe, Africa, India, China, Australia, and the rest of the world (ROW), respectively, while dashed lines represent the rates of change.

of NO and OH, heterogeneous reactions of $NO_2$, and photolysis of particulate nitrate[4,22]. In the SoilHONO case, traditional HONO sources and soil HONO emissions were both included (Methods). Soil NO emissions were considered in both cases (Methods). We compared the model simulations with atmospheric HONO measurements reported in the literature from 36 globally distributed sites across various years (Supplementary Fig. S6, Table S2). The results indicate that our simulated HONO mixing ratios fall within the range of the HONO observations in various years. When the soil HONO emissions were included, the normalized mean bias (NMB) between the simulated average and observed average HONO mixing ratios improved from −49% to −25%. To further evaluate the model performance on the diurnal variation, we selected two measurements with detailed diurnal information, both conducted in 2016 (Supplementary Fig. S7, the same as the simulation year), to compare the simulation results with the observed data. The findings indicate that the model has well reproduced the diurnal variation of ambient HONO mixing ratios after considering soil HONO emissions. By incorporating these soil emissions, the simulation results for 2016 at both sites showed significant improvement, with the NMB values for Beijing (Supplementary Fig. S7a) and Jinan (Supplementary Fig. S7b) improving from −48% to −25.0% and from −57% to −22%, respectively. These results indicate that including soil HONO emissions has improved the capability of the model to simulate atmospheric HONO.

The increase in atmospheric HONO mixing ratios due to soil emissions predominantly occurred in low-latitude regions (Fig. 4a). India exhibited the largest increase in the annual average HONO mixing ratio (962 pptv), followed by Australia (207 pptv). The geographical distribution of the increase in the absolute OH mixing ratio due to soil HONO emissions (Supplementary Fig. S8) was similar to that

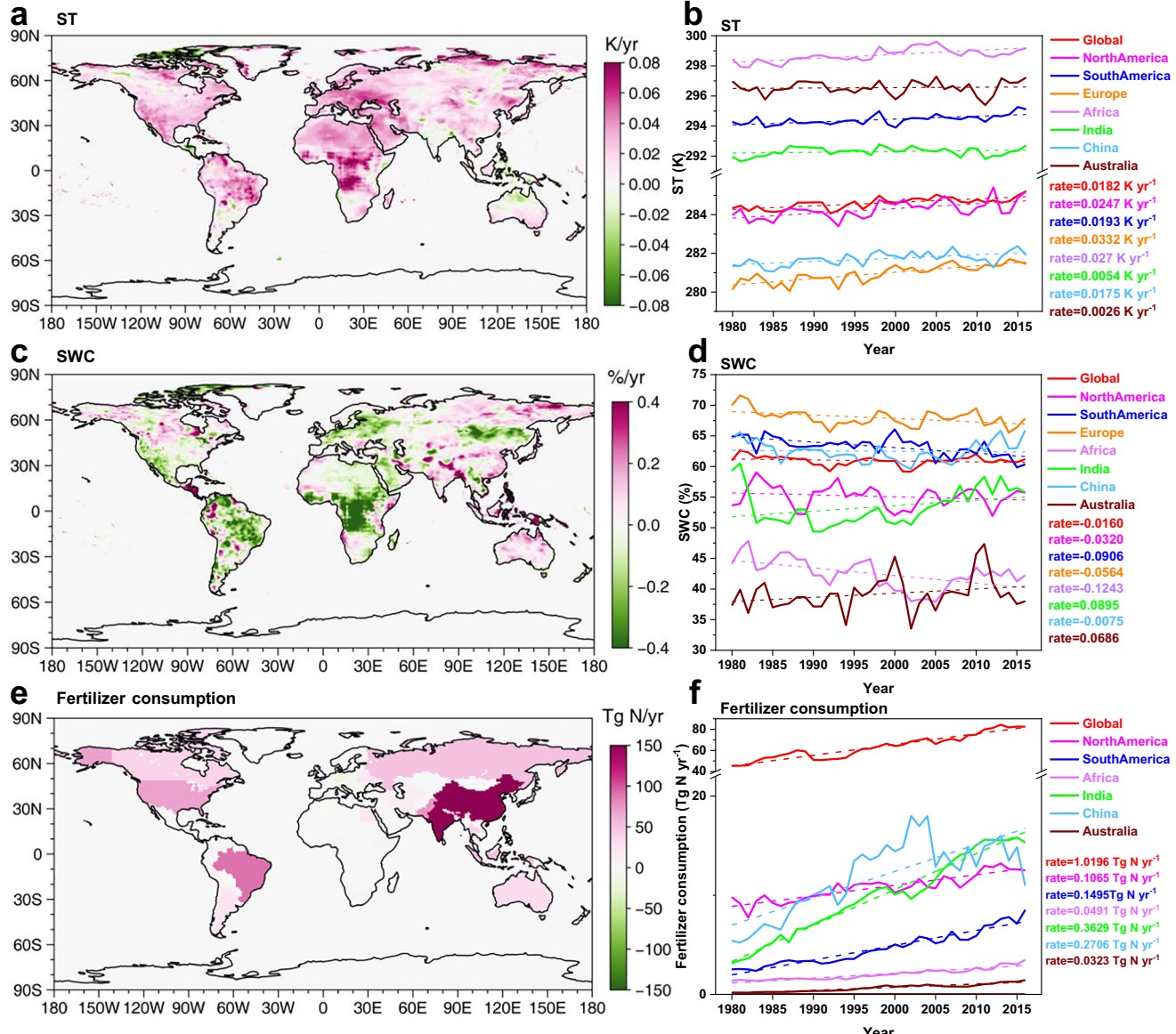

**Fig. 3 | Rates of change in soil temperature (ST), soil water content (SWC), and fertilizer usage from 1980 to 2016. a**, **c**, **e** Distribution of rates of change for ST, SWC, and fertilizer assumptions, respectively. **b**, **d**, **f** present the time-series of these variables from 1980 to 2016. Data for (**e**, **f**) are sourced from International Fertilizer Association statistics (IFASTAT)[30]. **f** does not show changes in fertilizer usage in Europe owing to incomplete data coverage by International Fertilizer Association (IFA) for the entire European region.

of the increase in HONO. Higher fractional increases in atmospheric HONO and OH due to soil HONO emissions were observed in areas with fewer anthropogenic pollution sources (Fig. 4b, Supplementary Fig. S8d). In Australia the annual increases in atmospheric HONO due to soil HONO emissions was 1244%, and the annual OH increased by 52%. In terms of seasonal variation, the enhancement of OH mixing ratios resulting from soil HONO emissions peaked during the respective summer to autumn seasons in both hemispheres (Supplementary Fig. S9b), which was attributed to a combined effect of fertilization and high ST during the growing season. This temporal variation also corresponded to the high emissions of soil HONO during these seasons, as mentioned earlier, indicating that soil HONO emissions considerably affect the atmospheric oxidation capacity, particularly during the growing season. We performed a calculation of the HONO budget for the troposphere in 2016 (Supplementary Fig. S10). The results reveal that the predominant source of HONO within the troposphere is chemical formation, representing a significant 93%. Following this, HONO

emissions from soil contribute 6%. In comparison, anthropogenic sources from vehicle exhaust and biomass burning each represent less than 1%. This suggests that while soil emissions only occur at the surface, their contribution to the overall HONO in the troposphere should not be overlooked.

Soil HONO emissions also increased surface atmospheric $O_3$ mixing ratios globally with an annual average of 2.5% (Fig. 4c, d), especially in regions characterized by lower pollution levels, e.g., the southern hemisphere, with annual increases of 15%, 9%, and 8% in Australia, Africa, and South America, respectively. The maximum increase in the annual average $O_3$ level for a single grid reached up to 29%. The larger percentage increase in $O_3$ mixing ratio due to soil HONO in the southern hemisphere could be explained by differences in the sensitivity of $O_3$ to its precursors. Anthropogenic $NO_x$ emissions are typically higher in the northern hemisphere than in the southern hemisphere[37]. In contrast, biogenic volatile organic compound emissions are considerably higher in the southern hemisphere[38].

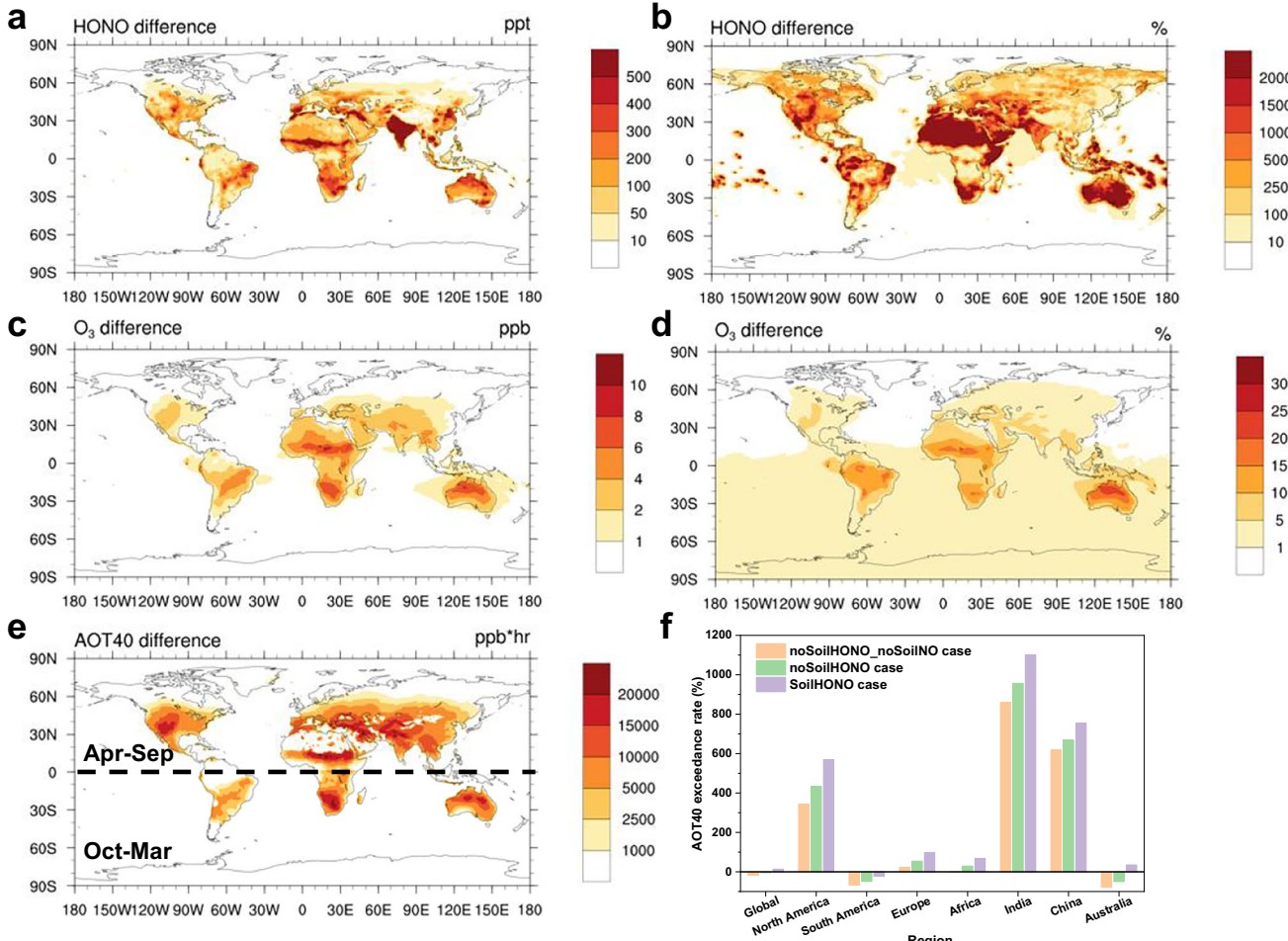

**Fig. 4 | Influence of soil HONO emissions on air quality and vegetation.**
**a**, **c** Absolute increases in HONO (pptv) and $O_3$ (ppbv) mixting ratios owing to soil HONO emissions, respectively. **b**, **d** Relative increases in these species owing to soil HONO emissions. **e** Absolute change in AOT40 (accumulated $O_3$ concentration over a threshold of 40 ppbv in the daytime) caused by soil HONO emissions. **f** Exceedance rates of AOT40 in different regions in 2016.

Consequently, most regions in the southern hemisphere are characterized by a $NO_x$-limited regime for $O_3$ production, and soil HONO emissions promote $O_3$ formation by increasing the $NO_x$ concentrations in these regions (Supplementary Fig. S8a, b), in addition to directly increasing OH radicals. The impact of soil HONO emissions on $O_3$ levels fluctuated with time of year. In the southern hemisphere, soil HONO emissions led to the greatest increase in $O_3$ in Australia during February, reaching up to 28% (Supplementary Fig. S9c). In the northern hemisphere, the highest contribution occurred in India during July (12%).

A sensitivity simulation quantified the overall impact of soil HONO and NO emissions by comparing the results from the noSoilHONO_noSoilNO (without soil HONO and NO emissions) case with those from the SoilHONO case (Supplementary Fig. S11). The effects of soil emissions on air quality extended beyond the regions where soil emissions occurred. Through long-range atmospheric transport, soil emissions led to an approximately 5% rise in near surface $O_3$ mixing ratios over the oceans of the southern hemisphere and the Antarctic region (Fig. 4d and Supplementary Fig. S11f).

To investigate the long-term effects of changing soil HONO emissions on air quality, we also simulated the impact of soil HONO emissions in the early 1980s (1981). In the noSoilHONO case, the mixing ratios of OH and $O_3$ were higher in 2016 compared to 1981 (Supplementary Table S3), consistent with the trend of continuous increases in global OH and $O_3$ mixing ratios reported in previous studies[39,40]. Soil HONO emissions significantly increased mixing ratios

of HONO, OH, and $O_3$ in 1981 (SoilHONO case, Supplementary Fig. S12), indicating a historical impact of soil HONO emissions on air quality that was previously overlooked. With changes in soil emissions and anthropogenic emissions from 1981 to 2016, the impact of soil HONO emissions on air quality varied in different parts of the world. Soil HONO emissions resulted in a larger atmospheric HONO mixing ratio increase in 2016 than in 1981, especially in regions with higher fertilization rates in India and eastern China (Fig. 5a). However, the enhancing effect of soil HONO emissions on OH and $O_3$ in 2016 declined in the Asian region, while increased in regions with low anthropogenic emissions in the southern hemisphere or reduced anthropogenic sources in North America and Europe (Fig. 5b, c). The larger promoting effect of soil HONO emissions in Asia in the early 1980s can be explained by the much smaller anthropogenic emissions in this region in the 1980s (Supplementary Fig. S13), leading to a stronger enhancement effect of soil emissions on OH and $O_3$ compared to 2016. These results suggest that with future reductions in anthropogenic emissions, the impact of soil emissions on air quality will become more significant in Asia, similar to the situation in North America and Europe.

## Influence of soil HONO emissions on vegetation through $O_3$ exposure

High $O_3$ concentrations have negative effects on vegetation[5,6]. To assess the potential vegetation exposure to $O_3$, we adopt the AOT40 metric, which represents the accumulated $O_3$ exposure over a threshold of

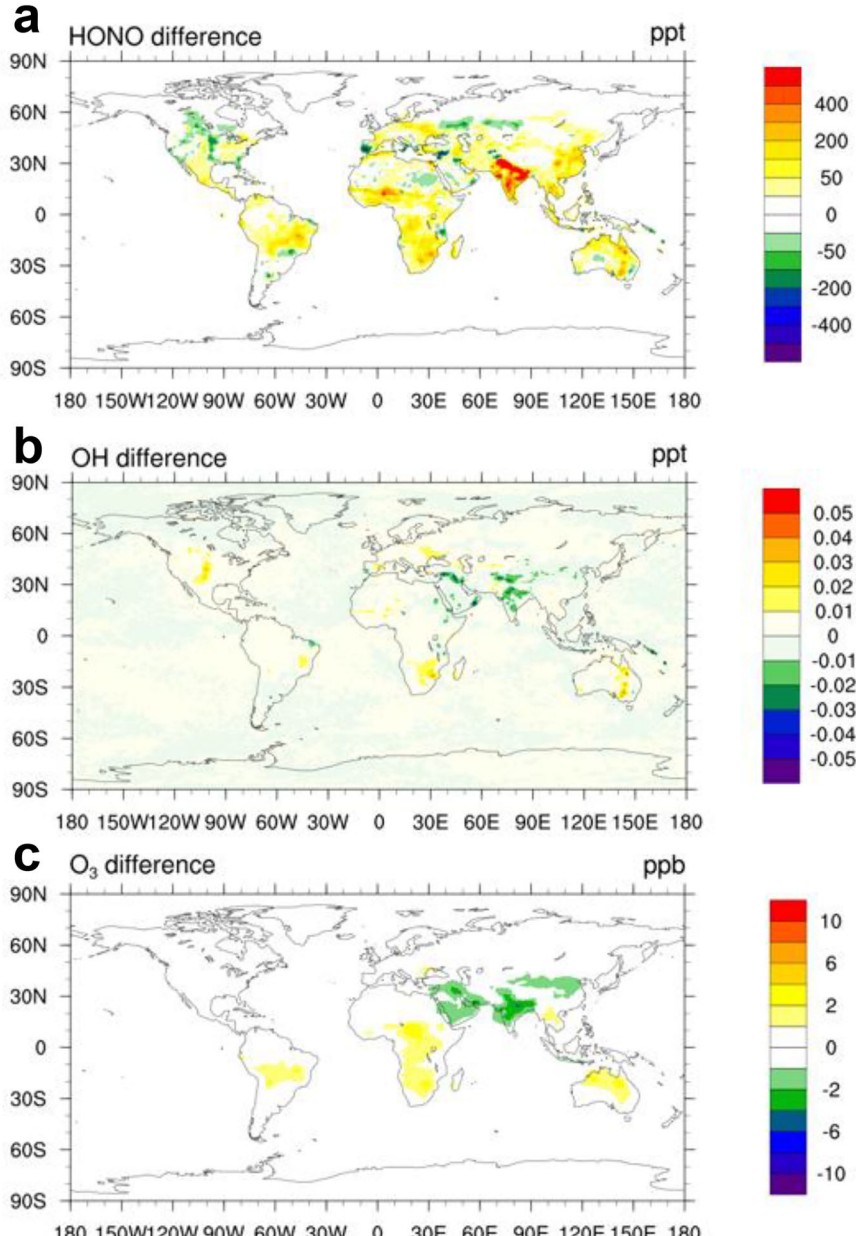

**Fig. 5 | The difference in the enhancement effects of soil HONO emissions on air quality between 2016 and 1980s. a**, **b**, **c** Comparison of the enhancing impact of soil HONO emissions on HONO (pptv), OH (pptv), and $O_3$ (ppbv) mixing ratios between 2016 and the 1980s.

40 ppbv in the daytime throughout a growing season. The selected growing season length has varied significantly across different studies, including 3, 6, and 12 months[41]. We calculated a 6-months AOT40 to compare our results with the AOT40 threshold values for vegetation protection (6 months = 5 ppmv hr) outlined in the United Nations Convention on Long-Range Transboundary Air Pollution[41]. The growing season in the northern hemisphere was from April to September, while that in the southern hemisphere spanned from October to March[41].

The global distribution of crop exposure to $O_3$ according to AOT40 metrics exemplarily in 2016 illustrated that AOT40 ranged from zero to over 100 ppmv hr worldwide (Supplementary Fig. S14). Higher exposure levels were found in Asia due to greater $O_3$ precursor emissions and concentrations. The increase in AOT40 due to soil HONO emissions (Fig. 4e) followed a spatial pattern that closely resembled the pattern of their impact on absolute $O_3$ levels (Fig. 4c). Large changes in $O_3$ exposure were in regions with lower anthropogenic $NO_x$ emissions, including western Asia, the central United

States, Brazil, northern Australia, and most parts of Africa. The highest increase could reach up to 10 ppm hr. In regions with higher anthropogenic $NO_x$ emissions, such as eastern China, the contribution of soil HONO emissions was less pronounced.

The exceedance rates of AOT40 under three different simulation cases varied across regions (Fig. 4f). The three regions with the highest AOT40 exceedance rates, India, China, and North America, also corresponded to the largest populations and highest grain production. In these regions, the threshold (5 ppmv hr) was exceeded by over 500%. Soil HONO emissions contributed significantly to the AOT40 exceedance rate in these three regions, with India showing the highest increase at 145%, followed by North America at 135% and China at 87%. This observation can be explained by the high anthropogenic emissions in these regions leading to high $O_3$ levels, and the additional $O_3$ resulting from soil HONO emissions significantly increased the exceedance rate of AOT40, affecting vegetation and crop production[5,42,43] in these three grain-producing regions. This finding reveals that the increase in

atmospheric $O_3$ levels caused by soil emissions may partly offset the benefits of fertilization on crop yield. In Australia and at the global scale, AOT40 remained below the threshold when soil HONO emissions were not considered. However, including soil HONO emissions in the model resulted in the simulated AOT40 exceeding the threshold values of 5 ppmv hr. This finding shows that soil HONO emissions have a significant impact on atmospheric oxidation capacity and $O_3$ levels, thereby adversely affecting vegetation. The health of vegetation directly influences not only the balance of ecosystems but also the production of food crops. Moreover, a reduced capacity of vegetation to absorb carbon dioxide due to $O_3$ damage can worsen the greenhouse effect, aggravating the effects of climate change[44].

## Implications

Until now the pivotal role of global soil HONO emissions for air quality and vegetation has been underexplored as most previous studies have only focused on soil NO emission. This work presents insights on the effect of climate warming and increased fertilizer application on rising soil HONO emissions and their significant impact on $O_3$ concentrations over the past four decades. The impact of soil emissions is expected to intensify with escalating global warming and increase in fertilizer use. The global surface temperature has increased by 1.09 °C since industrialization and is projected to rise by 1.4–4.4 °C by the end of the 21st century[27]. This will inevitably lead to elevated soil HONO emissions and higher near-surface $O_3$ concentrations, imposing a 'climate penalty'. In addition to global warming, recent years have witnessed an increase in extreme weather events such as heatwaves[45,46], which will further amplify the climate penalty by promoting soil HONO emissions. Besides global warming, the excessive utilization of nitrogen fertilizers, coupled with the decline in nitrogen use efficiency[47], will lead to an elevated loss of soil reactive nitrogen to the atmosphere (Fig. 6).

Consequently, these emissions affect air quality, food security, and human health.

Soil emissions are expected to have a larger impact on atmospheric chemistry and air quality in the future. Anthropogenic emissions are expected to gradually decline in the future due to enhanced pollution control and the rapid adoption of green energy policies. Supplementary Fig. S13 shows the changing trends in direct vehicle HONO emissions, indicating a continuous decrease in emissions from economically developed regions such as the United States and Europe. While developing countries in Asia have exhibited an increase in anthropogenic emissions over the past four decades, this trend is expected to reverse with the adoption of measures promoting low-carbon or zero-carbon emissions[48]. Wildfires are expected to increase in the future under the warming climate, exacerbating the HONO emissions from this sector. The sustained increase in soil HONO emissions combined with decreasing anthropogenic emissions will enhance the relative impact of soil HONO emissions on air quality and crop production in the future.

This work highlights the need for more efficient use of fertilizers in agriculture. While fertilizer use plays a key role in meeting the growing global food demand, it also increases soil emissions of reactive nitrogen, which adversely affects air quality and climate. These outcomes negatively impact ecosystems and crop yields, offsetting part of the benefits of fertilization on crop production and thereby jeopardizing sustainable food production goals. To mitigate these adverse side effects of soil emissions, strategies for efficient fertilizer usage, such as deep fertilizer placement, introduction of precise application rates, and the use of nitrification inhibitors are recommended. These measures will ensure a sustainable agriculture that meets the growing global food demand while preserving the environment.

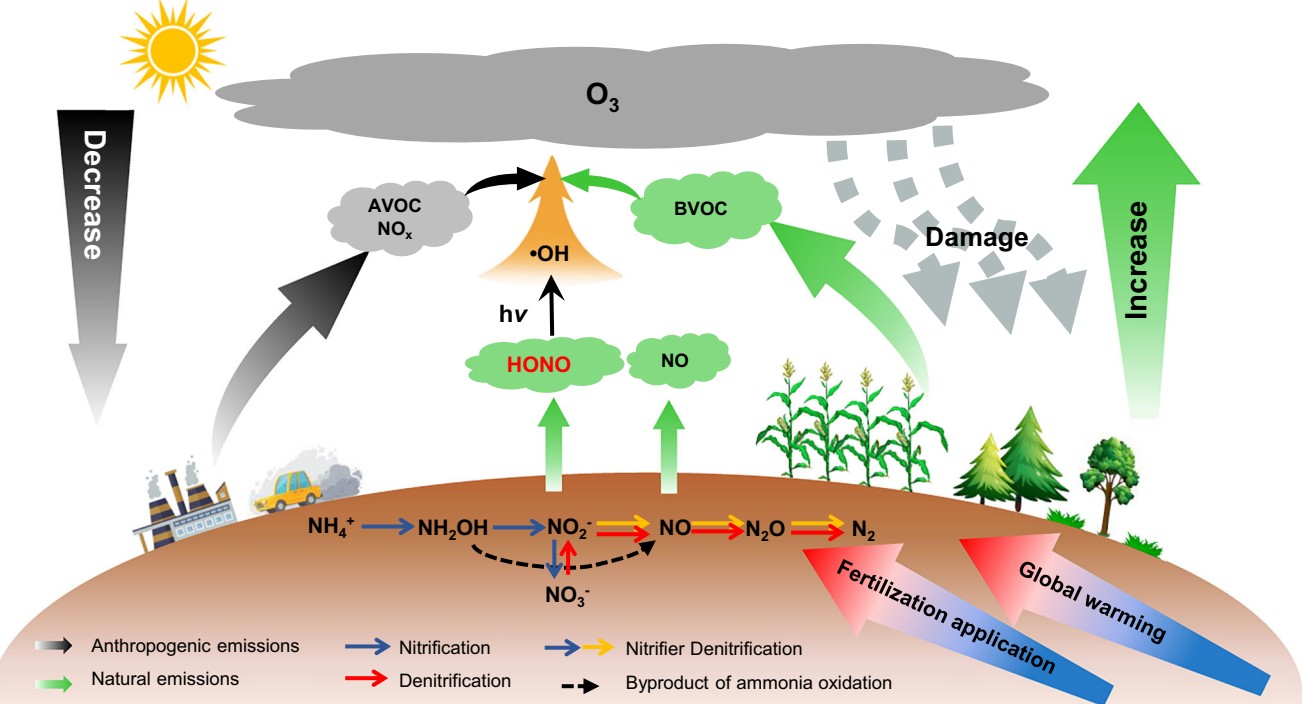

**Fig. 6 | Schematic of soil reactive oxidized nitrogen emissions and their impact on atmospheric composition, air quality, and vegetation.** Soil generates HONO and NO through nitrification (blue arrows), nitrifier denitrification (blue and yellow arrows), and denitrification (yellow arrows), and releases them into the atmosphere through soil–air exchange. The natural emissions (green arrows) exhibit an increasing trend under the combined effects of global warming and increased fertilization. Simultaneously, anthropogenic sources (black arrows) are projected to decrease in the future due to global emission mitigation measures. AVOC and BVOC represent anthropogenic and biogenic volatile organic compounds. The results highlight the growing relative contribution of soil reactive oxidized nitrogen emissions to $O_3$ formation and the subsequent impact on vegetation.

## Methods

### Comprehensive dataset compilation of soil HONO (NO) emissions

To aid the development of the global soil HONO emissions, we compiled a comprehensive dataset of published measurements of soil HONO emissions from diverse ecosystems worldwide (Supplementary Table S4). As SWC, ST and fertilization are the key factors affecting the soil HONO emission[49], we only considered those measurements that included observations of these parameters. Ultimately, the parametric formula we developed was based on emission data collected from five laboratory experiments involving 110 soil samples. We developed the parameterization scheme for soil NO emissions using the same method as that for HONO (Supplementary Fig. S15, Table S5).

Our parameterization scheme considers the effects of SWC, ST, and fertilization on soil HONO emissions, as these factors significantly influence emissions from the soil[49]. Please note that the influence of pH on soil HONO emissions is not explicitly included. Su et al.[50] suggests that acidic conditions favor the combination of $NO_2^-$ and $H^+$, leading to the conversion of HONO released from the soil. However, subsequent laboratory studies by Oswald et al.[26] based on global samples found that HONO emissions from neutral or alkaline soils are higher than those from acidic soils. They propose that ammonia-oxidizing bacteria can directly release HONO, which occurs in greater quantities than expected based on soil acid-base equilibrium and Henry's law balance. Research by Donaldson et al.[51] indicates that the surface acidity of soil minerals, rather than its pKa in bulk water, controls the form of $NO_2^-$ in the soil. Furthermore, pH can also affect soil nitrogen emissions by influencing microbial activity. Consequently, there is currently no consensus on how to quantify the impact of pH on soil HONO emissions. While we are currently unable to accurately quantify the impact of pH on soil HONO emissions, we have established parameterization schemes tailored to specific areas using emission measurement data from various land use types across different regions (see below). This ensures that our parameterization takes into account the three quantifiable variables (SWC, ST, and effects of fertilization), while unquantifiable variables were considered through zoning and land use types.

### Impact of SWC on soil HONO (NO) emissions

Previous studies revealed that with rising SWC, the soil HONO (NO) emissions first increased and then decreased, and the peak emissions occurred below 40% WHC[19,26]. The shapes and peak values of the emission curves collectively define the dependence of HONO (NO) emissions on SWC. The SWC dependence curves of the complied database (Supplementary Fig. S16) were used to derive the influence of SWC on HONO emissions and can be expressed as follows:

$$f(\text{SWC}) = F_{N, \max_{Peak1}} \cdot \exp\left(-\frac{\left(\text{SWC} - \text{SWC}_{C_{Peak1}}\right)^2}{w_{Peak1}^2}\right)$$
$$+ F_{N, \max_{Peak2}} \cdot \exp\left(-\frac{\left(\text{SWC} - \text{SWC}_{C_{Peak2}}\right)^2}{w_{Peak2}^2}\right) \quad (1)$$

where $F_{N, \max_{Peak}}$ denotes the maximum flux that occurs at the optimum SWC ($SWC_{C_{Peak}}$), and $w_{Peak}$ characterizes the width of the curve.

As the SWC dependence of soil HONO emissions varied significantly across different land-use types and regions, we averaged the emission across diverse land-use samples distributed over corresponding latitude bands, including north high latitude (NH, 60–90°N), north middle latitude (NM, 30–60°N), north low latitude (NL, 0–30°N), south low latitude (SL, 0–30°S), south middle latitude (SM, 30–60°S),

and south high latitude (SH, 60–90°S) with global averages used to fill in gaps in regions lacking samples (Supplementary Table S5). For regions like China and Europe, where a wealth of forest and grassland soil samples were available, we opted to utilize average emissions from these specific land-use categories in these two regions rather than relying solely on latitude band averages. The parameters $F_{N, \max_{Peak}}$, $SWC_{C_{Peak}}$, and $w_{Peak}$ in formula (1) specific to different land-use type in different latitude bands are listed in Supplementary Table S6. As shown in Supplementary Fig. S16, the emissions from agricultural soil are higher than those from other land use types, even during non-fertilization periods, which can be attributed to the cumulative effects of residual nitrogen over time.

### Impact of ST on HONO (NO) emissions

In addition to SWC, our study also considered the impact of ST on soil HONO (NO) emissions. We use the Arrhenius function to describe this relationship of exponential increase[18,26]. The $h(T)$ denotes the function of soil emissions on ST ($T$) and can be defined as the ratio of HONO (NO) emissions at $T$ ($F_N(T)$) to those at $T_0$ ($F_N(T_0)$):

$$h(T) = \frac{F_N(T)}{F_N(T_0)} = \frac{A \cdot \exp\left[\left(\frac{-Ea}{R}\right) \cdot \frac{1}{T}\right]}{A \cdot \exp\left[\left(\frac{-Ea}{R}\right) \cdot \frac{1}{T_0}\right]} = \exp\left[\left(\frac{-Ea}{R}\right) \cdot \left(\frac{1}{T} - \frac{1}{T_0}\right)\right] \quad (2)$$

Here, $A$ is a constant, $R$ denotes the gas constant, and $Ea$ represents the activation energy, with average values of 80, 75, and 44 kJ mol$^{-1}$ as reported by Oswald et al. [26] and Wang et al.[19]. Some field observations[10,52] have shown that when ST exceeds 30 °C, soil emissions may plateau or decrease. However, in field observations, changes in ST are typically accompanied by alterations in SWC and other environmental conditions. In contrast, laboratory experiments have the capability to control variables, thereby enabling a more precise assessment of the isolated impact of ST on emissions. The laboratory experiments by Ormeci et al.[53] demonstrated a significant exponential increase in soil NO emissions with increasing ST from 1 to 48 °C. Similarly, our laboratory experiments also revealed that within the temperature range of 5 °C to 55 °C, soil HONO emissions exhibit an exponential increase in response to rising ST. Therefore, we adopted an exponential relationship to describe the effect of ST on soil emissions. Taking both SWC and ST into account, soil HONO emissions were described as:

$$F_N = f(\text{SWC}) \cdot h(T) \quad (3)$$

### Scaling up laboratory results to field conditions

The parameterization scheme in this study relies on 110 laboratory measurements sourced from previous studies. In the laboratory dynamic chamber measurements of soil HONO (NO) emissions, the emission fluxes ($F_N$, ng N m$^{-2}$ s$^{-1}$) are calculated as:

$$F_N = \frac{Q}{A} \cdot [\text{HONO(NO)}]_{measure} \cdot \frac{M_N}{V_m} \quad (4)$$

where $Q$ represents the chamber inlet flow rate in litres per second (L s$^{-1}$), $A$ is the area of the petri dish in square meters (m$^2$), and $M_N$ (g mol$^{-1}$) and $V_m$ (L mol$^{-1}$) represent the molar mass of N and molar volume of the air, respectively.

In the real environment, soil HONO (NO) emission is the result of a bi-directional process. To quantify soil HONO emissions in the real ambient environment ($F_{emis}$) using laboratory results, we employed the following formula, which is derived from a standard formalism that characterizes the atmospheric-soil exchange of

trace gases[50,54].

$$F_{emis} = v_t \times [HONO(NO)]^* = v_t \times [HONO(NO)]_{measured} \qquad (5)$$

where $[HONO(NO)]^*$ is the equilibrium mixing ratio of HONO (NO) at the soil surface; $[HONO(NO)]_{measure}$ is the HONO (NO) mixing ratio we measured (ppbv) and is assumed to equal to $[HONO]^*$ due to the well-mixed air in the chamber; $v_t$ is the transfer velocity of HONO or NO, which was set at 2 and $1\,cm\,s^{-1}$, respectively.

Combining (1)–(5), we derive $F_{emis}$ as follows:

$$F_{emis} = v_t \times [HONO(NO)]^* = v_t \times [HONO(NO)]_{measured}$$

$$= v_t \times \frac{F_N}{\frac{Q}{A} \cdot \frac{M_N}{V_m}} = v_t \times \frac{F_{N,\,max_{Peak1}} \cdot \exp\left(-\frac{\left(SWC - SWC_{C_{Peak1}}\right)^2}{w_{Peak1}^2}\right) + F_{N,\,max_{Peak2}} \cdot \exp\left(-\frac{\left(SWC - SWC_{C_{Peak2}}\right)^2}{w_{Peak2}^2}\right)}{\frac{Q}{A} \cdot \frac{M_N}{V_m}} \times \exp\left[\left(\frac{-Ea}{R}\right) \cdot \left(\frac{1}{T} - \frac{1}{T_0}\right)\right] \qquad (6)$$

## Impact of fertilization on HONO (NO) emissions

To examine the effects of fertilization on HONO (NO) emissions, we considered soil pH as it can affect the transformation of $NH_4^+$-N and $NH_3$ volatilization after fertilizer application, subsequently influencing HONO (NO) emissions[18]. We adopted different parameterization schemes provided by Wang et al.[19] for acidic and alkaline soils. According to the global distribution of soil pH (Supplementary Fig. S17), we used the results for alkaline soils from Wang et al.[19] for soils with a pH >7, while for soils with a pH <7, we used the results for acidic soils from Wang et al.[19]. To consider both the amount and type of fertilizers, we calculated the proportions of different fertilizer types used in each country in 2015 (Supplementary Fig. S18) according to the data on fertilizer application amounts provided by the International Fertilizer Association (IFA)[30]. Based on the gridded N fertilizer rates in 2015 from Houlton et al.[55] and the N proportions of different fertilizer types used in 2015 in different countries from the IFA, we estimated the annual gridded fertilizer rates of different fertilizer types.

We allocated the fertilizer application amount over different periods of the year based on cropping intensity data[56]. We used leaf area index (LAI) in MERRA2 data to determine the start time of the growing season and thus the specific timing for fertilization (Supplementary Table S1). During fertilization, HONO (NO) emissions last for two weeks[15,18,20,21,57–59]. During this period, the soil emissions after the application of different fertilizers depend on ST and SWC, as reported by Wang et al.[18]. We used a linear increase in soil HONO (NO) emissions with rates of different fertilizers, following the IPCC guidelines for estimating soil $N_2O$ emissions and referencing field measurements of soil NO emissions, both of which suggested that emissions rise linearly with the amount of fertilizer applied. Based on this assumption, the HONO (NO) emission flux of different fertilizers at specific fertilization rates was estimated.

To calculate the long-term trend of soil HONO emissions caused by changes in fertilizer usage, we collected data on fertilizer application rates in various countries from 1980 to 2016, which were provided by IFA[30]. As the grid-based fertilizer data are only available for 2015[55], we used 2015 as a baseline year and calculated the annual relative changes in fertilizer application compared with 2015 using IFA data from 1980 to 2016. This allowed us to obtain grid-based fertilizer application data for each year.

Global ST and SWC data from 1980 to 2016 were obtained from the MERRA2 reanalysis dataset (https://gmao.gsfc.nasa.gov/reanalysis/MERRA-2/). Based on these data and the aforementioned parameterization, we derived the global soil HONO emissions between 1980 and 2016 (Figs. 1, 2).

## Impact of canopy reduction

Soil HONO (NO) emissions are impacted by canopy reduction, a process mainly driven by diffusion through plant stomata and direct deposition on the leaf cuticle, which is frequently quantified by the CRF. We used the following formula to adjust $F_{emis}$ to account for canopy reduction effects to yield the soil emissions to the atmosphere ($F_{canopy}$):

$$F_{canopy} = F_{emis} \times CRF = F_{emis} \times \frac{e^{-k_s \times SAI} + e^{-k_c \times LAI}}{2} \qquad (7)$$

where $k_s$ and $k_c$ are stomata and cuticle absorptivity constants, set at 8.75 and $0.24\,m^2\,m^{-2}$, respectively[33]. $SAI$ and $LAI$ are the stomatal area index and leaf area index, respectively. The LAI data is sourced from the MERRA2 dataset, while the SAI is derived from the SAI:LAI ratio across different land cover types according to Yienger and Levy[33], expressed as:

$$SAI = \frac{SAI_{YL95}}{LAI_{YL95}} \times LAI_{MERRA2} \qquad (8)$$

The potential uncertainties of soil emissions are discussed in the Supplementary Information.

## CAM-Chem model simulation

The CAM-Chem model[60,61] was used to simulate the global soil HONO impacts on air quality in early 1980s (1981) and 2016. The simulations were conducted using a horizontal resolution of $0.9° \times 1.25°$ and 56 vertical layers from the surface to the upper stratosphere. The lowest level of the model is about 100 m. A one-year spin-up period was implemented. Anthropogenic and biomass burning emissions were obtained from the Community Emissions Data System[62].

In the standard CAM-Chem model, there is no HONO production nor loss processes. Here, we incorporated direct emission sources and secondary formation pathways for HONO in the CAM-Chem, including traffic emission, biomass burning emission, homogeneous reaction of NO and OH, RH dependent and light-enhancing heterogeneous reactions of $NO_2$ on aerosol and ground surfaces, and photolysis of particulate nitrate[4,22]. The specific reactions and parameters for these sources utilized in the current study are provided in Supplementary Table S7. For HONO emissions from exhaust sources, the HONO to $NO_x$ ratios were found to vary significantly, ranging from 0.29% to 2.3% for vehicle exhausts[63–66] and from 3% to 6% for commercial aircraft[67]. In this study, we assumed a HONO/$NO_x$ ratio of 2.3% for all traffic (land vehicle, ship, and aircraft exhausts) HONO emissions. For biomass burning, HONO/$NO_x$ ratios have been reported to range from 0.025 to 0.23[68–70]. We used the upper bound value of 0.23 for the HONO emissions from biomass burning. The use of the upper limits of these parameters can help that we provide a conservative estimate of the soil HONO impacts compared to these two direct sources. For secondary chemical reactions, the parameters, such as $NO_2$ uptake coefficient ($\gamma_{NO_2}$) and nitrate photolysis rate constant ($J_{pNO_3}$), exhibit a broad range of variation, with differences in an order of magnitude. The secondary reaction parameters used in our research fall within the range reported previously (Supplementary Table S7) which have been adopted by many prior studies. We did not consider HONO emissions from livestock farming[71] nor from biocrusts[72], because so far there is not sufficient relevant data for deriving global

HONO emissions from these two new sources. The HONO absorption cross section used by the CAM-Chem model is based on the values recommended by Burkholder et al. [73], which is consistent with the recent laboratory measurements of Li et al. [72] (with difference <1%) (Supplementary Fig. S19). Considering that these non-soil HONO emissions/sources are potentially subject to uncertainties, we conducted sensitivity tests to evaluate the effects of selection of lower and upper limits of the non-soil emission sources on soil HONO's impact on atmospheric chemistry (Supplementary Text).

Three main simulation cases were designed to determine the effect of soil HONO emissions: noSoilHONO, SoilHONO, and noSoilHONO_noSoilNO. In the noSoilHONO and SoilHONO cases, soil NO and HONO emissions were considered based on the approach outlined above. The difference between noSoilHONO and SoilHONO indicated the impacts of soil HONO emissions, and the difference between noSoilHONO_noSoilNO and SoilHONO represented the impacts of both soil NO and soil HONO.

We also calculated HONO budget in the troposphere in 2016. Please note that due to the vertical transport between different layers in the chemical transport models, it is common to consider the total burden of a certain species within the entire troposphere when calculating the budget.

## Data availability

The data generated in this study is deposited in the Mendeley Data. https://doi.org/10.17632/6wmrvyp5xb.1[74].

## Code availability

The software code for the CESM model is available from http://www.cesm.ucar.edu/models/[75].

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

## Acknowledgements

This research was supported by The Hong Kong Research Grants Council (T24-504/17-N), the National Key Research and Development Program of China (2022YFC3701100), and the National Science Foundation of China (42293322). We thank Dr. Ivonne Trebs and Dr. Dianming Wu for provision of their published soil emission data and Dr. Ivonne Trebs for editing the manuscript.

## Author contributions

T.W. initiated this research. Q.L., Y.W., and T.W. designed the research. Y.W. collected and processed the data and provided input data and parameters for the model. Q.L. performed the model simulations. Q.L. and Y.W., with the help of Y.R.W. and C.R. conducted the data analysis and processing. Y.W., T.W., and Q.L. drafted the paper. L.X. and A.S.-L. provided comments on the paper. All authors contributed to the editing and revision of the paper.

## Competing interests

The authors declare no competing interest.
