## [Transparent Peer Review file · Nature Communications]

Increasing soil nitrous acid emissions driven by climate and fertilization change aggravate global ozone pollution

Corresponding Author: Professor Tao Wang

Version 0:

Reviewer comments:

Reviewer #1

(Remarks to the Author)

This paper develops a parameterization for soil HONO emissions and implements it into a global model to study the impacts on atmospheric composition. They also use information about meteorology and fertilizer use in the last 40 years to estimate how these impacts may be changing on a global scale. The results are interesting and provocative, in part because the authors use an approach to parameterizing the soil emissions that leads to much larger emissions of HONO than of NO (which is traditionally thought to dominate reactive nitrogen oxide emissions). These findings are consistent with some of the authors' laboratory studies on soil samples from across China, but not enough evidence is provided from ambient flux or concentration measurements to confirm that this is true under ambient conditions around the world. Figure S4 provides rather weak, indirect support: simulations of HONO concentrations with and without soil emissions largely overlap, and it is not clear which simulation better matches observations of concentrations at the majority of the sites. The text on lines 185-188 does provide more confidence, but HONO famously has many 'missing sources' so an average reduction in bias from adding a source isn't too convincing. Is there a way to organize the model measurement comparison (perhaps as a function of the relative importance of soil emissions) to gain more insight?

A significant amount of work was done to generate predictions of HONO emissions for soils around the world. The authors develop a parametric representation of emissions, relying on laboratory measurements of soil cores from different land cover types under a range of temperatures and water contents. I found the Methods section (lines 333 – 411) quite difficult to follow. The description mixes an explanation of the analysis of dynamic chamber measurements performed by some of the authors in previous publications, with the treatment of fluxes in a compensation point model, or in an analysis of ambient flux measurements.

Equation 4 emphasizes the fact that in the real environment, HONO fluxes can be bidirectional. Since the gross deposition flux (F_{dep}) requires knowledge of the atmospheric HONO concentration, it cannot be calculated without the chemical transport model. Results from laboratory studies can determine the gross emission flux (F_{emis}) as stated in Equation 6. From Su et al. (2011), and in analogy to bidirectional ammonia fluxes, my understanding is that the equilibrium concentration of gas phase HONO depends explicitly on soil pH, due to its impact on the effective Henry's Law constant. (I was surprised that this paper does not cite Su et al., since it seems very relevant). Furthermore, the parameterization only appears to consider pH as it affects the losses of fertilizer as volatilized ammonia.

Insufficient information is provided about the NO emissions parameterization. The single statement on lines 340-341 (The parameterization scheme for soil NO was the same as that for HONO.) leaves the reader confused – does this mean the NO emissions are the same as HONO? There are a handful of other global soil NO emissions estimates. It would be interesting to know how the magnitude and patterns in this study compares with other inventories.

Ultimately, the emissions parameterization may do a reasonable job, but there's not much evidence to support its quality. A downside of using an empirical, rather than more process-based approach to parameterize the emissions is that it's not clear whether predictions under changing conditions are robust. Some discussion of uncertainties would help to put the results into context.

The impacts of soil HONO emissions on atmospheric composition are (somewhat surprisingly) large in Figure 4. Figures S5-S8 show dramatic impacts on OH and O₃ in the model. Given that HONO rapidly photolyzes to form OH, it's not surprising that OH would increase directly, but that initial OH molecule would likely be quickly lost to surface uptake or reaction with a volatile organic compound. In low NO_x environments, that's unlikely to have a big impact on ozone production. One assumes that the release of NO from photolyzed HONO leads to a bigger (and more regional) impact on the oxidative budget

of the atmosphere in these simulations, but that information is missing from the manuscript. It would be helpful to know more about the treatment of emissions (is there any canopy reduction factor applied?) and what the height of the lowest model layer is.

For Figure 1, consider changing the maximum of the colour scale in panels d-g to saturate at e.g. 1.5 $\mu\text{g N ha}^{-1} \text{mth}^{-1}$ to better match the equivalent amounts in panel a, which is for a year, and to provide more colour contrast in the panels

For Figure 2, the units are very inconsistent and hard to follow. Given that the emissions are being expressed as Tg y^{-1} , why use two very different types of units to visualize the trends? The rates of change of emissions should be by $\text{Tg y}^{-1} \text{y}^{-1}$, or if those result in inconveniently small values, use Gg instead (rather than molecules cm^{-2} , $\text{s}^{-1} \text{y}^{-1}$ or kt y^{-1}).

References

Hang Su et al., Soil Nitrite as a Source of Atmospheric HONO and OH Radicals. *Science* 333,1616-1618 (2011). DOI:10.1126/science.1207687

Reviewer #2

(Remarks to the Author)

This manuscript is an important one in bringing attention to the subject of an overlooked but critical trace gas nitrous acid on global atmospheric chemistry and particularly the oxidation capacity in rural environments. It is definitely a topic that is worthy of publication in *Nature* and, in my opinion, should be published, subject to some considerations outlined below (i.e. minor/moderate revisions). I realize that it may not be feasible to address every single comment, so I am stipulating that not every one is mandatory.

I believe this manuscript would be more effective if it were to directly compare the modeling output with the output from other global models who have tried to include HONO (e.g., Elshorbany et al., 2014; Rasool et al., 2019). It would also be best to explicitly list the many different sources of HONO that are included in the model, as there appear to be a dizzying array of proposed sources in the literature including:

- 1) Abiotic equilibrium from $\text{NO}_2^- + \text{H}^+$ in acidic soils: Su et al. (2011);
- 2) Photosensitized reduction of NO_2 on surfaces with organics like humic acid: Kleffmann (2007); Song et al. (2022)
- 3) Heterogeneous disproportionation of NO_2 with water producing HONO + HNO_3 : Kleffmann (2007);
- 4) Heterogeneous photolysis of NO_3^- on ground or in aerosol phase: Zhou et al. (2003); Ye et al. (2017); Andersen et al. (2023).
- 5) Homogeneous photo-oxidation of HNO_3 (g): Song et al., STE, (2023).

Because global models of HONO are relatively rare, I think the manuscript would benefit tremendously from the addition of a global budget breakdown of HONO sources and sinks and overall lifetimes, etc. This would give the reader a sense of how large, for example, the agricultural soil source is with respect to other leading sources and sinks (e.g. Wang et al., ES&T, 2023; Song et al., *Env. Pol.*, 2022). Specifically, Zhang et al. (2023) claim that direct emissions of HONO (especially from livestock farming) make up ~40% of the total production in rural areas across the North China Plain.

The authors make a rather extraordinary claim that the modern global soil source of HONO is about 13.4 Tg N/yr (up 22% from 1980 levels). I say this is extraordinary because it is a direct source of NO_x to the atmosphere (once photolyzed in the daylight) and most recent models of ozone report overall NO_x budgets that amount to about 50 Tg N/yr (e.g. Hu et al., 2017), with approximately 20% (9.7 TgN/yr) coming from soil NO_x (as NO). Vinken et al. (2014) summarize over a dozen attempts to quantify the global emissions of soil NO_x reporting a range from 4 – 15 TgN/yr , proposing their own best guess at 13.9 TgN/yr . Looking into the model used in this study, CAM-Chem, according to Lamarque et al., GMD, (2012) for the year 2010 they report a soil source of 8 Tg NO-N/yr (out of a total of 41.7 Tg NO-N/yr .) This is very close to the value reported in your study on line 83 (7.3 Tg N/yr). This soil NO_x parameterization is in addition to the new HONO parameterization, which means that the claim of this work is that there is a soil source of nitrogen oxides to the atmosphere that approximately double what is currently assumed in a range of models. This means a change in the relative source strength of worldwide soils from approximately 20% – 40% of the global NO_x budget. This is a remarkable change in our understanding, and I think it is incumbent on the authors to discuss how this shift might be true and might have been overlooked for so long. For example, might many of the “top-down” estimates summarized in Vinken et al. (2014) be counting HONO + NO from soils? Might this mean that there needs to be changes configured into the model soil parameterizations for NO emissions to accommodate this new, large HONO source? Maybe your model’s combined $\text{NO} + \text{HONO}$ emissions (20.7 Tg N/yr) is possibly most in-line with the Steinkamp & Lawrence (2011) result reported using the arithmetic mean value (27.6 Tg N/yr), which has been an outlier in past intercomparisons (Vinken et al., 2014).

Furthermore, I wonder what HONO absorption cross section is used in your model, and what a 30% smaller cross section would do to your analysis (model and obs intercomparisons). Li et al., ES&T (2024) have recently claimed that past photolysis rates of HONO are 20%-30% too large throwing a lot of uncertainty into our understanding of global HONO budgets. I bring all of these issues up in order to urge the authors to strongly consider attempting to present some reasonable estimate of an uncertainty associated with this new global result.

One other modeling effort that I am aware of is the agroecosystem model of Fertilizer Emission Scenario Tool for the CMAQ (FEST- C) developed at Rice University (Rasool et al., 2019; Luo et al., 2022). It seems to me that it would be worthwhile to compare the emissions from this work to that of another one such as Rasool et al. (2019) to see how they might compare. It would also help shore up confidence in the modeling if you were to compare known characteristics of the seasonal and diurnal dependence of the HONO concentrations with those from measurements. That is, are the max:min ratios similar for

night:day and winter:summer to what has been measured, at least at one or two sites.

Specific Comments:

Line 34: Is this increase in global surface O₃ an annual average? 29% is the max for an annual average? or daily average? A maximum daily 8hr average might be more meaningful. Polluted regions will often have higher O₃ maxima but also lower overnight minima (due to greater NO titration), so the 24-hour average is not always the best metric for understanding health or photochemistry impacts.

Lines 82/83: Does your parameterization scheme have a built-in canopy factor to account for deposition of NO₂ in the canopy? Or do you have emissions and dry deposition and the model just adds them together for a net flux? Is it treated the same in the HONO parameterization?

Line 96: Since your model finds that the vast majority of HONO emissions arise from unmanaged ecosystems on the continents, are you aware of the work being done on estimating emissions from biocrusts in arid landscapes (Weber et al., 2015)? Also, how do you treat “legacy” N in the agricultural regions? Or more specifically, does your model generate more HONO emissions in agricultural areas during non-fertilizing periods than in other comparable climates?

Figures 2-5 are too small to be read easily. For the 6-plot figures, I suggest changing from 3 by 2 to 2 by 3 and increasing their size. Perhaps make Figure 5 vertical instead of horizontal. I would also suggest converting all figures to PNG format for better zooming readability.

Line 123: The value of 19.2 C does not match the value in Figure S2 (which has the July peak T over 20 C).

Lines 230-232: These changes need to be defined at a particular model level (e.g. near-surface). If they are near the surface it should be noted that radiative forcing of tropospheric O₃ is weakest at the surface and strongest near the tropopause (Lacis et al., 1990).

Lines 282/283: This is a significant claim that requires significant substantiation. Is there any data on AOT40 impacts on crop yield? There is definitely data on the significant increases in crop yield due to fertilizer amendments, which are very large. In order for there to be a significant offset would require an estimate of the two competing effects.

Figure 6: The figure shows HONO emitted from nitrite in soil, but a more thorough picture of the source parent species might be worthwhile, as shown in Song et al., 2023, coming from NH₂OH and through an unknown enzymatic route from NO₂⁻. Further, aside from having more soil sources of HONO than this figure depicts, NO is also emitted in nitrification and denitrification cycles, not just from NO₂⁻. I therefore recommend a more complete pictorial representation of the processes in soils that generate NO + HONO.

Line 317: I think you should probably lay out what are the “anthropogenic HONO emissions” specifically because there are a wide variety of proposed HONO sources in the literature. For example, Song et al. (2022) report sources like: vehicle exhaust, pNO₃- photolysis, and wildfires. Wildfires are increasing worldwide, so how might that impact future trends in HONO emissions? It might help to lay out a global budget of HONO as captured in your model to give readers an idea of what the approximate relative importance is of all of these varied sources/sinks.

Lines 351/352: How might the use of the average emission from each land-type translate into an uncertainty in the global emissions values reported here? I realize this is a difficult task but it is worthwhile to try to keep track of the uncertainties in your model for the community to better fit them into our understanding. In general, some sort of reckoning of uncertainty needs to be done in this work. I realize it can be quite challenging when there are so many fits and approximations involved, but it is important because this is an extraordinary claim you are making that could change the way the field considers soil NO_x emissions.

Line 369: Oswald et al. (2013) never test at any temperature above 30 C, so I am not sure why there would not be a plateau in HONO emissions at high T as there is for NO. Please see Stark (1996) and Arroyo et al. (2022) for reasons why soil microbial processes tend to develop an optimum (extremum) as opposed to simply monotonically increasing as an abiotic Arrhenius function. As pointed out by Wang et al. (2021), the use of a higher temperature emission peak led to a net increase of soil NO emissions by almost 20% across North America. It would be helpful to state something about how the uncertainty in the high temperature dependence of your HONO emissions parameterization might influence your overall results.

Lines 400/401: Why do you assume a linear increase in HONO? Was HONO shown to have a linear increase with fertilization rates in the studies used to develop your parameterization? If so please cite these studies to justify this claim. Another study of soil NO_x emissions by Oikawa et al. (2015) showed that doubling fertilizer inputs of urea fertilizer increased NO_x fluxes by a factor of 5. While this study was not ostensibly measuring HONO, the use of a linear dependence on fertilizer N should be clarified and justified. Moreover, the authors should clarify the influence that each fertilizer type has on HONO emissions and whether a linear correlation has been reported for each fertilizer type.

SI Figure 14 – What year is this fertilizer data from? Is it an average of all 36 years? This should be specified in the figure. Is there a description of how fertilizer type influences the HONO emission rates? Or is it assumed to not matter?

Another miscellaneous comment:

From Wang et al., *Env. Sci. & Tech.* (2021): “In view of the difficulty of accurately simulating soil moisture during fertilization that typically occurs along with irrigation, we incorporated the average, lowermost and uppermost soil HONO emissions in the SWC range of 60– 100% WHC after using urea (the dominant fertilizer in China) to represent the average case (SoilHONO_avg), the lower limit case (SoilHONO_min), and the upper limit case (SoilHO- NO_max).” In extending this to a global scale, what method was used to overcome this difficulty in determining the appropriate SWC in the model? This again begs the question of what do you expect the overall uncertainty to be in your modeling results? It would help to discuss in detail potential sources of uncertainty in all components of HONO sources for the sake of transparency so that future work may evolve from here. In fact, it would be best to include a designated section discussing the major uncertainties of this modeling approach.

References:

- Andersen, S. T., Carpenter, L. J., Reed, C., Lee, J. D., Chance, R., Sherwen, T., ... & Fomba, K. W. (2023). Extensive field evidence for the release of HONO from the photolysis of nitrate aerosols. *Science Advances*, 9(3), eadd6266.
- Arroyo, J. I., Díez, B., Kempes, C. P., West, G. B., & Marquet, P. A. (2022). A general theory for temperature dependence in biology. *Proceedings of the National Academy of Sciences*, 119(30), e2119872119.
- Elshorbany, Y. F., Crutzen, P. J., Steil, B., Pozzer, A., Tost, H., & Lelieveld, J. (2014). Global and regional impacts of HONO on the chemical composition of clouds and aerosols. *Atmospheric Chemistry and Physics*, 14(3), 1167-1184.
- Hu, L., Jacob, D. J., Liu, X., Zhang, Y., Zhang, L., Kim, P. S., ... & Yantosca, R. M. (2017). Global budget of tropospheric ozone: Evaluating recent model advances with satellite (OMI), aircraft (IAGOS), and ozonesonde observations. *Atmospheric Environment*, 167, 323-334.
- Kleffmann, J. (2007). Daytime sources of nitrous acid (HONO) in the atmospheric boundary layer. *ChemPhysChem*, 8(8), 1137-1144.
- Lacis, A. A., Wuebbles, D. J., & Logan, J. A. (1990). Radiative forcing of climate by changes in the vertical distribution of ozone. *Journal of Geophysical Research: Atmospheres*, 95(D7), 9971-9981.
- Lamarque, J.-F., Emmons, L. K., Hess, P. G., Kinnison, D. E., Tilmes, S., Vitt, F., Heald, C. L., Holland, E. A., Lauritzen, P. H., Neu, J., Orlando, J. J., Rasch, P. J., and Tyndall, G. K.: CAM-chem: description and evaluation of interactive atmospheric chemistry in the Community Earth System Model, *Geosci. Model Dev.*, 5, 369–411, <https://doi.org/10.5194/gmd-5-369-2012>, 2012.
- Li, X., Tian, S., Zu, K., Xie, S., Dong, H., Wang, H., ... & Zhang, Y. (2024). Revisiting the Ultraviolet Absorption Cross Section of Gaseous Nitrous Acid (HONO): New Insights for Atmospheric HONO Budget. *Environmental Science & Technology*, 58(9), 4247-4256.
- Luo, L., Ran, L., Rasool, Q. Z., & Cohan, D. S. (2022). Integrated modeling of US agricultural soil emissions of reactive nitrogen and associated impacts on air pollution, health, and climate. *Environmental Science & Technology*, 56(13), 9265-9276.
- Rasool, Q. Z., Bash, J. O., & Cohan, D. S. (2019). Mechanistic representation of soil nitrogen emissions in the Community Multiscale Air Quality (CMAQ) model v 5.1. *Geoscientific Model Development*, 12(2), 849-878.
- Song, Yifei, Zhang, Y., Xue, C., Liu, P., He, X., Li, X., & Mu, Y. (2022). The seasonal variations and potential sources of nitrous acid (HONO) in the rural North China Plain. *Environmental Pollution*, 311, 119967.
- Song, Yaqui, Wu, D., Ju, X., Dörsch, P., Wang, M., Wang, R., ... & Yu, Y. (2023). Nitrite stimulates HONO and NO_x but not N₂O emissions in Chinese agricultural soils during nitrification. *Science of the Total Environment*, 902, 166451.
- Stark, J. M. (1996). Modeling the temperature response of nitrification. *Biogeochemistry*, 35, 433-445.
- Su, H., Cheng, Y., Oswald, R., Behrendt, T., Trebs, I., Meixner, F. X., ... & Pöschl, U. (2011). Soil nitrite as a source of atmospheric HONO and OH radicals. *Science*, 333(6049), 1616-1618.
- Vinken, G. C. M., Boersma, K. F., Maasakkers, J. D., Adon, M., & Martin, R. V. (2014). Worldwide biogenic soil NO_x emissions inferred from OMI NO₂ observations. *Atmospheric Chemistry and Physics*, 14(18), 10363-10381.
- Wang, Y., Ge, C., Garcia, L. C., Jenerette, G. D., Oikawa, P. Y., & Wang, J. (2021). Improved modelling of soil NO_x emissions in a high temperature agricultural region: role of background emissions on NO₂ trend over the US. *Environmental*

research letters, 16(8), 084061.

Wang, J., Zhang, Y., Zhang, C., Wang, Y., Zhou, J., Whalley, L. K., ... & Ye, C. (2023). Validating HONO as an intermediate tracer of the external cycling of reactive nitrogen in the background atmosphere. *Environmental Science & Technology*, 57(13), 5474-5484.

Weber, B., Wu, D., Tamm, A., Ruckteschler, N., Rodriguez-Caballero, E., Steinkamp, J., ... & Pöschl, U. (2015). Biological soil crusts accelerate the nitrogen cycle through large NO and HONO emissions in drylands. *Proceedings of the National Academy of Sciences*, 112(50), 15384-15389.

Ye, C., Zhang, N., Gao, H., & Zhou, X. (2017). Photolysis of Particulate Nitrate as a Source of HONO and NO_x. *Environmental science & technology*, 51(12), 6849-6856.

Zhang, Q., Liu, P., Wang, Y., George, C., Chen, T., Ma, S., ... & Zeng, Y. (2023). Unveiling the underestimated direct emissions of nitrous acid (HONO). *Proceedings of the National Academy of Sciences*, 120(35), e2302048120.

Zhou, X., Gao, H., He, Y., Huang, G., Bertman, S. B., Civerolo, K., & Schwab, J. (2003). Nitric acid photolysis on surfaces in low-NO_x environments: Significant atmospheric implications. *Geophysical Research Letters*, 30(23).

Reviewer #3

(Remarks to the Author)

Reviewer #4

(Remarks to the Author)

The manuscript derive the global soil HONO emissions from natural and fertilized lands over the past four decades and evaluate the variation of their impacts on global O₃ and vegetation exposure. The results indicated that climate change and the enhanced fertilizer use increased global soil HONO emissions. Simulations with the updated model showed that soil HONO emissions increased global surface O₃ concentrations and the subsequent risk of vegetation exposure to O₃, especially in crop-production regions. The results are very interesting, and have important meaningful for mitigating global air pollution. The article need to be improved, then can be considered to publish.

1. HONO emission from soil is also affected by soil microorganisms, soil pH, light and other factors. Did the author take other factors into account when collecting data? At present, only the two parameters of soil temperature and water content are considered. Are these two factors necessarily the most important factors affecting global soil emissions of HONO?

2. In Figure 1, Global annual average soil HONO emissions 89 in 2016 are overall higher than HONO emissions in these months in d-g, so which months are the main contributors to the high HONO emissions in a? Is it necessary to show the months with high HONO emissions as well?

3. This paper focuses on the situation of soil emission of HONO, then in areas with more exposed soil, there are generally fewer motor vehicles, such as forest areas, farmland, etc., is it unreasonable to choose a high value of 2.3% for motor vehicle emission factors? In this way, the contribution of soil to HONO will be underestimated.

4. About RH-dependent and light-enhancing heterogeneous reactions of NO₂ on various surface, at present, there are many new developments for reference, and the author's reference parameterization scheme needs to be further updated.

5. How reliable is CAM-Chem model? This paper concludes that a large number of simulation results are focused on the model, so the current CAM-Chem model and HONO multi-source parameterization schemes need to be described in more detail. In addition, the parameterization scheme of HONO, including soil emission, NO₂ heterogeneous reaction, vehicle emission and nitrate photolysis, is highly controversial. It is suggested that the author conduct sensitivity analysis on key parameters, explore the changes of important parameters on the amount of HONO emitted by soil, and further evaluate the subsequent impact on O₃.

Version 1:

Reviewer comments:

Reviewer #1

(Remarks to the Author)

The authors have extensively revised the manuscript in response to the reviewers' comments and made the results easier to understand and, in several cases, more robust. I still believe that the application of the laboratory parameterization, which has only been tested against a single set of field observations, to the entire globe is highly uncertain. However this sort of analysis is needed to motivate and contextualize further field observations to evaluate the provocative idea that the authors are putting forward. The methodology they have followed with respect to the parameterization development and model simulations appears sound and is now more clearly explained.

Reviewer #2

(Remarks to the Author)

I have reviewed all of the authors' responses to my first review and find that their actions satisfy what I would consider a good-faith attempt to redress all of my concerns. I therefore recommend that the revised manuscript be published as has been submitted. Thanks to the authors for receiving my comments and critiques and responding to them earnestly.

Reviewer #3

(Remarks to the Author)

Reviewer #4

(Remarks to the Author)

The authors have revised all my comments, the manuscript can be published.

Response to Reviewer #1

We thank the reviewers for the comments and suggestions (in black font) on the manuscript. Our responses (in blue) and the corresponding edits in the manuscript (in red) are shown below.

Comments:

1. This paper develops a parameterization for soil HONO emissions and implements it into a global model to study the impacts on atmospheric composition. They also use information about meteorology and fertilizer use in the last 40 years to estimate how these impacts may be changing on a global scale. The results are interesting and provocative, in part because the authors use an approach to parameterizing the soil emissions that leads to much larger emissions of HONO than of NO (which is traditionally thought to dominate reactive nitrogen oxide emissions). These findings are consistent with some of the authors' laboratory studies on soil samples from across China, but not enough evidence is provided from ambient flux or concentration measurements to confirm that this is true under ambient conditions around the world.

Response 1.1:

We thank the reviewer for the constructive comment. We would like to clarify that our proposed soil HONO and NO emission parameterizations (in which soil HONO emissions exceed soil NO emissions) are supported by laboratory reports by several groups, including independent studies by Huang et al.¹, Xue et al.², as well as our previous research³. (The reference list is attached at the end of this response letter to all reviewers.)

Currently, to our best knowledge, there is only one study⁴ worldwide that has conducted simultaneous field measurements of HONO and NO emission fluxes; these fluxes were observed at a rice-wheat rotation cropland (32°25' N, 116°47' E) using the aerodynamic gradient method. The results show that under non-flooded (dryland) conditions, with a SWC of 45%, the emissions of HONO and NO are 0.26 and 0.53 nmol m⁻² s⁻¹ (equivalent to 3.64 and 7.42 ng N m⁻² s⁻¹), respectively, with NO emissions indeed being higher than HONO emissions.

In Figure R1, we show the relationship between SWC and both HONO and NO emissions in forest and cropland used for the north-mid (NM, 30°N to 60°N) latitude band in our study. The relative levels of HONO versus NO emissions fluctuate with variations in SWC. When SWC reaches 45%, the emissions from cropland soils are 43.8 and 52.1 ng m⁻² s⁻¹ for HONO and NO, respectively; by considering the canopy reduction factor (formula (7) in the revised Method section, see our Response 1.7 for details), we found F_{canopy} values of 5.9 and 7.0 ng m⁻² s⁻¹ for HONO and NO, respectively, indicating that NO emissions are also higher than HONO emissions in this region based on our estimation. This is consistent with the aforementioned field observations⁴, suggesting that under certain soil moisture conditions, NO emissions can indeed exceed those of HONO. However, when averaged over a year for the whole

globe, our results suggest that HONO emissions surpass NO emissions from soil (as shown in Figure 1 and Figure S1 of the revised manuscript).

On the global scale, our estimated soil HONO emissions ($11.5 \text{ Tg N yr}^{-1}$) and NO emissions (6.2 Tg N yr^{-1}) are well within the previously reported range, i.e., $7.36\text{--}11.99 \text{ Tg N yr}^{-1}$ for soil HONO⁵ and $4\text{--}21 \text{ Tg N yr}^{-1}$ for soil NO⁶⁻⁸. Besides, our CAM-Chem simulations suggest that the inclusion of soil HONO sources helps to reproduce the field observations of HONO mixing ratios. Overall, we believe that by considering all available soil HONO measurements and the dominant controlling factors of soil HONO emissions (soil temperature, soil water content, land cover type, fertilizer consumption, etc.), our proposed global soil HONO emissions represent the state-of-the-art knowledge.

That being said, in the future, it would be great to have simultaneous field flux measurements of HONO and NO at multiple sites in different land use types to further investigate the source intensity and distribution of soil reactive nitrogen emissions.

Figure R1. The dependence of soil HONO emissions on soil water content (SWC) in NM latitude band in China. The blue line and orange line represent soil emissions from forests and cropland, respectively. The solid line indicates soil HONO emissions, while the dashed line indicates soil NO emissions.

We have added additional descriptions and discussions in the revised manuscript, as shown below:

Line 84-96 (all line numbers for the revised text indicates those in the clean version of revised manuscript):

“The application of the CRF decreases the global total soil HONO and NO emissions to 11.5 and 6.2 Tg N yr⁻¹, respectively. (Fig.1 and Supplementary Fig. S1). The estimated soil HONO emission is comparable to the value of 9.7 Tg N yr⁻¹ obtained by Wu et al.⁵. The soil NO emissions is slightly higher than the estimate of 5.5 Tg N yr⁻¹ calculated by Yienger and Levy⁹ using the YL95 model, but lower than the 9.0 Tg N yr⁻¹ estimated by Hudman et al.⁶ using the BDSNP model, and the 12.9 ± 3.9 Tg N yr⁻¹ estimated by Vinken et al.⁸ using a top-down approach. Overall, the soil NO emissions in our study fall within the range (4–21 Tg N yr⁻¹) reported in previous studies⁶⁻⁸. Note that the top-down estimate method may include both soil HONO and soil NO as soil NO_x, our estimate for global soil emissions of HONO + NO totals 17.7 Tg N yr⁻¹, which closely aligns with the upper estimate of top-down soil NO_x emission (16.8 Tg N yr⁻¹). Our estimates indicate that the global soil emissions of HONO exceed those of NO. As most previous studies on soil emissions of reactive oxidized nitrogen focused on soil NO, they likely underestimated the impacts of soil emissions on atmospheric chemistry and air quality.”

Line 372-374:

“To aid the development of the global soil HONO emissions, we compiled a comprehensive dataset of published measurements of soil HONO emissions from diverse ecosystems worldwide (Supplementary Table S4).”

2. Figure S4 provides rather weak, indirect support: simulations of HONO concentrations with and without soil emissions largely overlap, and it is not clear which simulation better matches observations of concentrations at the majority of the sites. The text on lines 185-188 does provide more confidence, but HONO famously has many ‘missing sources’ so an average reduction in bias from adding a source isn’t too convincing. Is there a way to organize the model measurement comparison (perhaps as a function of the relative importance of soil emissions) to gain more insight?

Response 1.2:

The original Figure S4 shows the comparison between the observed and simulated average HONO concentrations in the cases with and without soil HONO emissions. We found that the normalized mean bias (NMB) between the observed and simulated HONO concentrations decrease from -49% in the case without soil HONO emissions to -25% in the case with soil HONO, suggesting a significant improvement in reproducing ambient HONO concentrations. In the original Figure S4, we used a logarithmic scale for the Y-axis which may have obscured the distinction between the cases. In the revised manuscript, we have revised this figure utilizing a linear scale for the Y-axis to more clearly illustrate the effects of including soil HONO emissions (shown as Figure R2 below, and Figure S6 in revised SI).

Furthermore, we selected observations with detailed diurnal variation information, i.e., campaigns in Beijing (Figure R3a)¹⁰ and Jinan (Figure R3b)¹¹ which were both conducted in 2016 (the same as the simulation year), to further validate the model performance on simulating the diurnal HONO pattern. The comparison suggests that the diurnal patterns of simulated HONO in both cases align well with the observed data;

when soil HONO is included, there is a significant improvement, with NMB values for Beijing and Jinan improving from -48% to -25% and from -57% to -22%, respectively. We have included Figure R3 in the revised supplementary information as Figure S7 and referenced it in the main text.

Indeed, as the reviewer mentioned, atmospheric HONO has many missing sources. One of these sources is soil emission, which has been proposed as an important HONO source for a long time. However, until the current paper, there has been no study that compiled the long-term trends of global soil HONO emissions and quantified the subsequent soil HONO impacts on global air quality. The authors believe that by considering all available soil HONO measurements and the predominant factors in one coherent framework, the current study is a major step forward in understanding the global HONO budget, soil nitrogen sources, as well as their impacts on atmospheric chemistry.

Figure R2. Comparison of simulated and observed HONO concentrations at 36 observation sites. The blue and red lines illustrate the simulated average values during the campaign period for the noSoilHONO and SoilHONO cases, respectively. The black squares and whiskers represent the observed average mixing ratios and the range, respectively. The information of these sites is shown in Table S2.

Figure R3. Observed and simulated diurnal variations of HONO mixing ratios at (a) Beijing and (b) Jinan sites in 2016. The black dots represent observed mixing ratios, and the blue and red lines represent the simulated results of noSoilHONO and SoilHONO cases, respectively. The observation at Beijing is reported by Wang et al. ¹⁰; that at Jinan by Li et al. ¹¹.

We have added more discussion in the revised manuscript.

Line 203-217:

“We compared the model simulations with atmospheric HONO measurements reported in the literature from 36 globally distributed sites across various years (Supplementary Fig. S6, Table S2). The results indicate that our simulated HONO mixing ratios fall within the range of the HONO observations in various years. When the soil HONO emissions were included, the normalized mean bias (NMB) between the simulated average and observed average HONO mixing ratios improved from -49% to -25%. To further evaluate the model performance on the diurnal variation, we selected two measurements with detailed diurnal information, both conducted in 2016 (Supplementary Fig. S7, the same as the simulation year), to compare the simulation results with the observed data. The findings indicate that the model has well reproduced the diurnal variation of ambient HONO mixing ratios after considering soil HONO emissions. By incorporating these soil emissions, the simulation results for 2016 at both sites showed significant improvement, with the NMB values for Beijing (Supplementary Fig. S7a) and Jinan (Supplementary Fig. S7b) improving from -48% to -25.0% and from -57% to -22%, respectively. These results indicate that including soil HONO emissions has improved the capability of the model to simulate atmospheric HONO.”

3. A significant amount of work was done to generate predictions of HONO emissions for soils around the world. The authors develop a parametric representation of emissions, relying on laboratory measurements of soil cores from different land cover types under a range of temperatures and water contents. I found the Methods section

(lines 333 – 411) quite difficult to follow. The description mixes an explanation of the analysis of dynamic chamber measurements performed by some of the authors in previous publications, with the treatment of fluxes in a compensation point model, or in an analysis of ambient flux measurements.

Response 1.3:

We thank the reviewer for his/her feedback. Our parameterization scheme is based on all available laboratory measurements (110 samples). To apply the laboratory results to real-world environments, we need to consider the bidirectional exchange processes. Therefore, we utilized standard forms of equations that describe atmospheric-soil trace gas exchange for the conversion.

Following the reviewer's comment, we have restructured and revised the Methods section to clarify our approach and added subheadings for better readability.

Line 369-540:

“Global soil HONO emissions

Comprehensive dataset compilation of soil HONO (NO) emissions

To aid the development of the global soil HONO emissions, we compiled a comprehensive dataset of published measurements of soil HONO emissions from diverse ecosystems worldwide (Supplementary Table S4). As SWC, ST and fertilization are the key factors affecting the soil HONO emission¹², we only considered those measurements that included observations of these parameters. Ultimately, the parametric formula we developed was based on emission data collected from five laboratory experiments involving 110 soil samples. We developed the parameterization scheme for soil NO emissions using the same method as that for HONO (Supplementary Fig. S15, Table S5).

Our parameterization scheme considers the effects of SWC, ST, and fertilization on soil HONO emissions, as these factors significantly influence emissions from the soil¹². Please note that the influence of pH on soil HONO emissions is not explicitly included. Su et al.¹³ suggests that acidic conditions favour the combination of NO_2^- and H^+ , leading to the conversion of HONO released from the soil. However, subsequent laboratory studies by Oswald et al.¹⁴ based on global samples found that HONO emissions from neutral or alkaline soils are higher than those from acidic soils. They propose that ammonia-oxidizing bacteria can directly release HONO, which occurs in greater quantities than expected based on soil acid-base equilibrium and Henry's law balance. Research by Donaldson et al.¹⁵ indicates that the surface acidity of soil minerals, rather than its pKa in bulk water, controls the form of NO_2^- in the soil. Furthermore, pH can also affect soil nitrogen emissions by influencing microbial activity. Consequently, there is currently no consensus on how to quantify the impact of pH on soil HONO emissions. While we are currently unable to accurately quantify the impact of pH on soil HONO emissions, we have established parameterization schemes tailored to specific areas using emission measurement data from various land use types across different regions (see below). This ensures that our parameterization takes into account the three quantifiable variables (SWC, ST, and effects of fertilization), while unquantifiable variables were considered through zoning and land use types.

Impact of SWC on soil HONO (NO) emissions

Previous studies revealed that with rising SWC, the soil HONO (NO) emissions first increased and then decreased, and the peak emissions occurred below 40% WHC^{3,14}. The shapes and peak values of the emission curves collectively define the dependence of HONO (NO) emissions on SWC. The SWC dependence curves of the compiled database (Supplementary Fig. S16) were used to derive the influence of SWC on HONO emissions and can be expressed as follows:

$$f(\text{SWC}) = F_{N,\text{maxPeak1}} \cdot \exp\left(-\frac{(\text{SWC}-\text{SWC}_{C\text{Peak1}})^2}{w_{\text{Peak1}}^2}\right) + F_{N,\text{maxPeak2}} \cdot \exp\left(-\frac{(\text{SWC}-\text{SWC}_{C\text{Peak2}})^2}{w_{\text{Peak2}}^2}\right) \quad (1)$$

where $F_{N,\text{maxPeak}}$ denotes the maximum flux that occurs at the optimum SWC ($\text{SWC}_{C\text{Peak}}$), and w_{Peak} characterizes the width of the curve.

As the SWC dependence of soil HONO emissions varied significantly across different land-use types and regions, we averaged the emission across diverse land-use samples distributed over corresponding latitude bands, including north high latitude (NH, 60–90°N), north middle latitude (NM, 30–60°N), north low latitude (NL, 0–30°N), south low latitude (SL, 0–30°S), south middle latitude (SM, 30–60°S), and south high latitude (SH, 60–90°S) with global averages used to fill in gaps in regions lacking samples (Supplementary Table S5). For regions like China and Europe, where a wealth of forest and grassland soil samples were available, we opted to utilize average emissions from these specific land-use categories in these two regions rather than relying solely on latitude band averages. The parameters $F_{N,\text{maxPeak}}$, $\text{SWC}_{C\text{Peak}}$, and w_{Peak} in formula (1) specific to different land-use type in different latitude bands are listed in Supplementary Table S6. As shown in Supplementary Fig. S16, the emissions from agricultural soil are higher than those from other land use types, even during non-fertilization periods, which can be attributed to the cumulative effects of residual nitrogen over time.

Impact of ST on HONO (NO) Emissions

In addition to SWC, our study also considered the impact of ST on soil HONO (NO) emissions. We use the Arrhenius function to describe this relationship of exponential increase^{14,16}. The $h(T)$ denotes the function of soil emissions on ST (T) and can be defined as the ratio of HONO (NO) emissions at T ($F_N(T)$) to those at T_0 ($F_N(T_0)$):

$$h(T) = \frac{F_N(T)}{F_N(T_0)} = \frac{A \cdot \exp\left[\left(\frac{-Ea}{R}\right) \cdot \frac{1}{T}\right]}{A \cdot \exp\left[\left(\frac{-Ea}{R}\right) \cdot \frac{1}{T_0}\right]} = \exp\left[\left(\frac{-Ea}{R}\right) \cdot \left(\frac{1}{T} - \frac{1}{T_0}\right)\right] \quad (2)$$

Here, A is a constant, R denotes the gas constant, and Ea represents the activation energy, with average values of 80, 75, and 44 kJ mol⁻¹ as reported by Oswald et al.¹⁴ and Wang et al.³. Some field observations^{17,18} have shown that when ST exceeds 30°C,

soil emissions may plateau or decrease. However, in field observations, changes in ST are typically accompanied by alterations in SWC and other environmental conditions. In contrast, laboratory experiments have the capability to control variables, thereby enabling a more precise assessment of the isolated impact of ST on emissions. The laboratory experiments by Ormeçi et al.¹⁹ demonstrated a significant exponential increase in soil NO emissions with increasing ST from 1 to 48°C. Similarly, our laboratory experiments also revealed that within the temperature range of 5°C to 55°C, soil HONO emissions exhibit an exponential increase in response to rising ST. Therefore, we adopted an exponential relationship to describe the effect of ST on soil emissions. Taking both SWC and ST into account, soil HONO emissions were described as:

$$F_N = f(\text{SWC}) \cdot h(T) \quad (3)$$

Scaling up laboratory results to field conditions

The parameterization scheme in this study relies on 110 laboratory measurements sourced from previous studies. In the laboratory dynamic chamber measurements of soil HONO (NO) emissions, the emission fluxes (F_N , ng N m⁻² s⁻¹) are calculated as:

$$F_N = \frac{Q}{A} \cdot [\text{HONO}(\text{NO})]_{\text{measure}} \cdot \frac{M_N}{V_m} \quad (4)$$

where Q represents the chamber inlet flow rate in litres per second (L s⁻¹), A is the area of the petri dish in square metres (m²), and M_N (g mol⁻¹) and V_m (L mol⁻¹) represent the molar mass of N and molar volume of the air, respectively.

In the real environment, soil HONO (NO) emission is the result of a bi-directional process. To quantify soil HONO emissions in the real ambient environment (F_{emis}) using laboratory results, we employed the following formula, which is derived from a standard formalism that characterizes the atmospheric-soil exchange of trace gases^{13,20}.

$$F_{\text{emis}} = v_t \times [\text{HONO}(\text{NO})]^* = v_t \times [\text{HONO}(\text{NO})]_{\text{measured}} \quad (5)$$

where $[\text{HONO}(\text{NO})]^*$ is the equilibrium mixing ratio of HONO (NO) at the soil surface; $[\text{HONO}(\text{NO})]_{\text{measured}}$ is the HONO (NO) mixing ratio we measured (ppbv) and is assumed to equal to $[\text{HONO}]^*$ due to the well-mixed air in the chamber; v_t is the transfer velocity of HONO or NO, which was set at 2 and 1 cm s⁻¹, respectively.

Combining (1)–(5), we derive F_{emis} as follows:

$$F_{\text{emis}} = v_t \times [\text{HONO}(\text{NO})]^* = v_t \times [\text{HONO}(\text{NO})]_{\text{measured}} = v_t \times \frac{F_N}{\frac{Q M_N}{A V_m}} =$$

$$v_t \times \frac{F_{N,\text{maxPeak1}} \cdot \exp\left(-\frac{(\text{SWC}-\text{SWC}_{\text{Peak1}})^2}{w_{\text{Peak1}}^2}\right) + F_{N,\text{maxPeak2}} \cdot \exp\left(-\frac{(\text{SWC}-\text{SWC}_{\text{Peak2}})^2}{w_{\text{Peak2}}^2}\right)}{\frac{Q M_N}{A V_m}} \times$$

$$\exp\left[\left(\frac{-E_a}{R}\right) \cdot \left(\frac{1}{T} - \frac{1}{T_0}\right)\right] \quad (6)$$

Impact of fertilization on HONO (NO) emissions

To examine the effects of fertilization on HONO (NO) emissions, we considered soil pH as it can affect the transformation of NH₄⁺-N and NH₃ volatilization after fertilizer application, subsequently influencing HONO (NO) emissions¹⁶. We adopted

different parameterization schemes provided by Wang et al.³ for acidic and alkaline soils. According to the global distribution of soil pH (Supplementary Fig. S17), we used the results for alkaline soils from Wang et al.³ for soils with a pH > 7, while for soils with a pH < 7, we used the results for acidic soils from Wang et al.³. To consider both the amount and type of fertilizers, we calculated the proportions of different fertilizer types used in each country in 2015 (Supplementary Fig. S18) according to data on fertilizer application amounts provided by the International Fertilizer Association (IFA)²¹. Based on the gridded N fertilizer rates in 2015 from Houlton et al.²² and the N proportions of different fertilizer types used in 2015 in different countries from the IFA, we estimated the annual gridded fertilizer rates of different fertilizer types.

We allocated the fertilizer application amount over different periods of the year based on cropping intensity data²³. We used leaf area index (LAI) in MERRA2 data to determine the start time of the growing season and thus the specific timing for fertilization (Supplementary Table S1). During fertilization, HONO (NO) emissions last for two weeks^{16,24-29}. During this period, the soil emissions after the application of different fertilizers depend on ST and SWC, as reported by Wang et al.¹⁶ We used a linear increase in soil HONO (NO) emissions with rates of different fertilizers, following the IPCC guidelines for estimating soil N₂O emissions and referencing field measurements of soil NO emissions, both of which suggested that emissions rise linearly with the amount of fertilizer applied. Based on this assumption, the HONO (NO) emission flux of different fertilizers at specific fertilization rates was estimated.

To calculate the long-term trend of soil HONO emissions caused by changes in fertilizer usage, we collected data on fertilizer application rates in various countries from 1980 to 2016, which were provided by IFA²¹. As the grid-based fertilizer data are only available for 2015²², we used 2015 as a baseline year and calculated the annual relative changes in fertilizer application compared with 2015 using IFA data from 1980 to 2016. This allowed us to obtain grid-based fertilizer application data for each year.

Global ST and SWC data from 1980 to 2016 were obtained from the MERRA2 reanalysis dataset (<https://gmao.gsfc.nasa.gov/reanalysis/MERRA-2/>). Based on these data and the aforementioned parameterization, we derived the global soil HONO emissions between 1980 and 2016 (Fig. 1, 2).

Impact of canopy reduction

Soil HONO (NO) emissions are impacted by canopy reduction, a process mainly driven by diffusion through plant stomata and direct deposition on the leaf cuticle, which is frequently quantified by the canopy reduction factor (CRF). We used the following formula to adjust F_{emis} to account for canopy reduction effects to yield the soil emissions to the atmosphere (F_{canopy}):

$$F_{canopy} = F_{emis} \times CRF = F_{emis} \times \frac{e^{-k_s \times SAI} + e^{-k_c \times LAI}}{2} \quad (7)$$

where k_s and k_c are stomata and cuticle absorptivity constants, set at 8.75 and 0.24 $m^2 m^{-2}$, respectively⁹. SAI and LAI are the stomatal area index and leaf area index, respectively. The LAI data is sourced from the MERRA2 dataset, while the SAI is derived from the SAI:LAI ratio across different land cover types according to Yienger and Levy⁹, expressed as:

$$SAI = \frac{SAI_{YL95}}{LAI_{YL95}} \times LAI_{MERRA2} \quad (8)$$

The potential uncertainties of soil emissions are discussed in the Supplementary Information.

CAM-Chem model simulation

The CAM-Chem model^{30,31} was used to simulate the global soil HONO impacts on air quality in early 1980s (1981) and 2016. The simulations were conducted using a horizontal resolution of $0.9^\circ \times 1.25^\circ$ and 56 vertical layers from the surface to the upper stratosphere. The lowest level of the model is about 100 m. A one-year spin-up period was implemented. Anthropogenic and biomass burning emissions were obtained from the Community Emissions Data System³².

In the standard CAM-Chem model, there is no HONO production nor loss processes. Here, we incorporated direct emission sources and secondary formation pathways for HONO in the CAM-Chem, including traffic emission, biomass burning emission, homogeneous reaction of NO and OH, RH dependent and light-enhancing heterogeneous reactions of NO₂ on aerosol and ground surfaces, and photolysis of particulate nitrate^{33,34}. The specific reactions and parameters for these sources utilized in the current study are provided in Supplementary Table S7. For HONO emissions from exhaust sources, the HONO to NO_x ratios were found to vary significantly, ranging from 0.29% to 2.3% for vehicle exhausts³⁵⁻³⁸ and from 3% to 6% for commercial aircraft³⁹. In this study, we assumed a HONO/NO_x ratio of 2.3% for all traffic (land vehicle, ship, and aircraft exhausts) HONO emissions. For biomass burning, HONO/NO_x ratios have been reported to range from 0.025 to 0.23⁴⁰⁻⁴². We used the upper bound value of 0.23 for the HONO emissions from biomass burning. The use of the upper limits of these parameters can help that we provide a conservative estimate of the soil HONO impacts compared to these two direct sources. For secondary chemical reactions, the parameters, such as NO₂ uptake coefficient (γ_{NO_2}) and nitrate photolysis rate constant (J_{pNO_3}), exhibit a broad range of variation, with differences in an order of magnitude. The secondary reaction parameters used in our research fall within the range reported previously (Supplementary Table S7) which have been adopted by many prior studies. We did not consider HONO emissions from livestock farming⁴³ nor from biocrusts⁴⁴, because so far there is not sufficient relevant data for deriving global HONO emissions from these two new sources. The HONO absorption cross section used by the CAM-Chem model is based on the values recommended by Burkholder et al.⁴⁵, which is consistent with the recent laboratory measurements of Li et al.⁷² (with difference <1%)(Supplementary Fig. S19). Considering that these non-soil HONO emissions/sources are potentially subject to uncertainties, we conducted sensitivity tests to evaluate the effects of selection of lower and upper limits of the non-soil emission sources on soil HONO's impact on atmospheric chemistry (Supplementary Text).

Three main simulation cases were designed to determine the effect of soil HONO emissions: noSoilHONO, SoilHONO, and noSoilHONO_noSoilNO. In the noSoilHONO and SoilHONO cases, soil NO and HONO emissions were considered

based on the approach outlined above. The difference between noSoilHONO and SoilHONO indicate the impacts of soil HONO emissions, and the difference between noSoilHONO_noSoilNO and SoilHONO highlighted the impacts of both soil NO and soil HONO.

We also calculated HONO budget in the troposphere in 2016. Please note that due to the vertical transport between different layers in the chemical transport models, it is common to consider the total burden of a certain species within the entire troposphere when calculating the budget.”

4. Equation 4 emphasizes the fact that in the real environment, HONO fluxes can be bidirectional. Since the gross deposition flux (F_{dep}) requires knowledge of the atmospheric HONO concentration, it cannot be calculated without the chemical transport model. Results from laboratory studies can determine the gross emission flux (F_{emis}) as stated in Equation 6. From Su et al. (2011), and in analogy to bidirectional ammonia fluxes, my understanding is that the equilibrium concentration of gas phase HONO depends explicitly on soil pH, due to its impact on the effective Henry’s Law constant. (I was surprised that this paper does not cite Su et al., since it seems very relevant). Furthermore, the parameterization only appears to consider pH as it affects the losses of fertilizer as volatilized ammonia.

Response 1.4:

Indeed, as mentioned by the referee and in our original manuscript, HONO fluxes are bidirectional. In this process, soil HONO emissions (F_{emis}) are influenced by many factors, including soil physicochemical properties and soil microbial activity. The effect of pH on soil HONO emissions is also influenced from multiple aspects. Su et al.¹³ suggested that acidic conditions favor the combination of NO_2^- and H^+ , leading to the production of HONO that is released from the soil. However, subsequent laboratory studies by Oswald et al.¹⁴ based on global samples found that HONO emissions from neutral or alkaline soils are higher than those from acidic soils, positing that ammonia-oxidizing bacteria can directly release HONO, which occurs in greater quantities than the calculation based on soil acid-base equilibrium and Henry’s law balance. Research by Donaldson et al.¹⁵ further indicated that the surface acidity of soil minerals, rather than its pKa in bulk water, controls the form of NO_2^- in the soil. In addition, pH can also affect soil nitrogen emissions by influencing microbial activity.

To conclude, the effect of pH on soil HONO emissions is not merely related to acid-base balance, and more importantly, there is still no consensus on the impact of pH on soil HONO emissions. Therefore, we have not specifically considered soil pH in our soil HONO emission parameterization. Instead, our study utilized emission measurements from various land use types across different regions to develop a parameterization scheme tailored to each specific area. This approach somewhat accounts for the variations in soil pH values among the regions. We have included a discussion on the exclusion of pH in the parameterization for natural soil HONO emissions and have cited the work of Su et al.¹³ in the revised manuscript.

Line 379-395:

“Our parameterization scheme considers the effects of SWC, ST, and fertilization

on soil HONO emissions, as these factors significantly influence emissions from the soil¹². Please note that the influence of pH on soil HONO emissions is not explicitly included. Su et al.¹³ suggests that acidic conditions favour the combination of NO_2^- and H^+ , leading to the conversion of HONO released from the soil. However, subsequent laboratory studies by Oswald et al.¹⁴ based on global samples found that HONO emissions from neutral or alkaline soils are higher than those from acidic soils. They propose that ammonia-oxidizing bacteria can directly release HONO, which occurs in greater quantities than expected based on soil acid-base equilibrium and Henry's law balance. Research by Donaldson et al.¹⁵ indicates that the surface acidity of soil minerals, rather than its pKa in bulk water, controls the form of NO_2^- in the soil. Furthermore, pH can also affect soil nitrogen emissions by influencing microbial activity. Consequently, there is currently no consensus on how to quantify the impact of pH on soil HONO emissions. While we are currently unable to accurately quantify the impact of pH on soil HONO emissions, we have established parameterization schemes tailored to specific areas using emission measurement data from various land use types across different regions (see below). This ensures that our parameterization takes into account the three quantifiable variables (SWC, ST, and effects of fertilization), while unquantifiable variables were considered through zoning and land use types.”

5. Insufficient information is provided about the NO emissions parameterization. The single statement on lines 340-341 (The parameterization scheme for soil NO was the same as that for HONO.) leaves the reader confused – does this mean the NO emissions are the same as HONO? There are a handful of other global soil NO emissions estimates. It would be interesting to know how the magnitude and patterns in this study compares with other inventories.

Response 1.5:

Here we clarify that in our study, the soil NO emissions are not the same as soil HONO emission; instead, we meant that the methodology used to derive the soil NO emissions parameterization was the same as the approach adopted for HONO, relying on laboratory measurements of 110 global soil samples.

To increase the clarity of the method section, we have introduced the notation “HONO (NO)” in Method section to indicate that the method applied to derive soil NO parameterization is identical to that used for soil HONO. In the revised manuscript, we have explicitly clarified this issue in Line 375-376.

Line 377-378:

“We developed the parameterization scheme for soil NO emissions using the same method as that for HONO (Supplementary Fig. S15, Table S5).”

As to the magnitude of soil NO emission compared to previous estimates, at line 83 of the original manuscript, we mentioned that our estimated soil NO emissions falls well within the range of previous reported values. In the revised manuscript, we have expanded this discussion to offer a more comprehensive comparison of our results with global soil NO emissions estimates.

Line 84-96

“The application of the CRF decreases the global total soil HONO and NO emissions to 11.5 and 6.2 Tg N yr⁻¹, respectively. (Fig.1 and Supplementary Fig. S1). The estimated soil HONO emission is comparable to the value of 9.7 Tg N yr⁻¹ obtained by Wu et al.⁵. The soil NO emissions is slightly higher than the estimate of 5.5 Tg N yr⁻¹ calculated by Yienger and Levy⁹ using the YL95 model, but lower than the 9.0 Tg N yr⁻¹ estimated by Hudman et al.⁶ using the BDSNP model, and the 12.9 ± 3.9 Tg N yr⁻¹ estimated by Vinken et al.⁸ using a top-down approach. Overall, the soil NO emissions in our study fall within the range (4–21 Tg N yr⁻¹) reported in previous studies⁶⁻⁸. Note that the top-down estimate method may include both soil HONO and soil NO as soil NO_x, our estimate for global soil emissions of HONO + NO totals 17.7 Tg N yr⁻¹, which closely aligns with the upper estimate of top-down soil NO_x emission (16.8 Tg N yr⁻¹). Our estimates indicate that the global soil emissions of HONO exceed those of NO. As most previous studies on soil emissions of reactive oxidized nitrogen focused on soil NO, they likely underestimated the impacts of soil emissions on atmospheric chemistry and air quality.”

6. Ultimately, the emissions parameterization may do a reasonable job, but there’s not much evidence to support its quality. A downside of using an empirical, rather than more process-based approach to parameterize the emissions is that it’s not clear whether predictions under changing conditions are robust. Some discussion of uncertainties would help to put the results into context.

Response 1.6:

As noted in our Response 1.1 and 1.2, we have validated the reliability of our method and results from multiple perspectives. We compared our estimated soil HONO and NO emissions with those from previous studies and found that the magnitude, distribution, and seasonal variations are comparable to those reported in earlier research; we also compared the observed and simulated HONO mixing ratios in the cases with and without soil HONO and the results show that the inclusion of soil HONO helped to simulate the magnitude and diurnal pattern of atmospheric HONO; these model performance evaluation provide credibility and reliability for our parameterized method.

Currently, there are only a limited number of studies providing estimates of soil HONO emissions and even fewer for global estimates. To our best knowledge, Rasool et al.⁴⁶ and Luo et al.⁴⁷ were the only studies that utilized the process-based approach, DayCENT (Daily time-step version of the CENTURY biogeochemical model) and FEST-C (Fertilizer Emission Scenario Tool for CMAQ), to estimate soil HONO emissions but only in the United States; nonetheless, their approach was also partly based on an empirical relationship from Oswald et al.¹⁴, which describes the relationship between soil HONO emissions and SWC, as well as the ratio of soil HONO to NO emissions. Although using a different method, our estimated soil reactive oxidized nitrogen (HONO+NO) emissions merely differ from theirs by less than 24%.

We would like to emphasize that our parameterization method is based on laboratory measurements, which span a broad range of soil temperature (5-55°C) and SWC (0-100 %WHC) conditions that can fully reflect the variations in real-world soil

environments. Our parameterization also accounts for the effects of different fertilizer application rates and types on soil HONO (NO) emissions. Hence, we believe that our parameterization represents soil emissions across varying conditions.

Line 84-93:

“The application of the CRF decreases the global total soil HONO and NO emissions to 11.5 and 6.2 Tg N yr⁻¹, respectively. (Fig.1 and Supplementary Fig. S1). The estimated soil HONO emission is comparable to the value of 9.7 Tg N yr⁻¹ obtained by Wu et al.⁵. The soil NO emissions is slightly higher than the estimate of 5.5 Tg N yr⁻¹ calculated by Yienger and Levy⁹ using the YL95 model, but lower than the 9.0 Tg N yr⁻¹ estimated by Hudman et al.⁶ using the BDSNP model, and the 12.9 ± 3.9 Tg N yr⁻¹ estimated by Vinken et al.⁸ using a top-down approach. Overall, the soil NO emissions in our study fall within the range (4–21 Tg N yr⁻¹) reported in previous studies⁶⁻⁸. Note that the top-down estimate method may include both soil HONO and soil NO as soil NO_x, our estimate for global soil emissions of HONO + NO totals 17.7 Tg N yr⁻¹, which closely aligns with the upper estimate of top-down soil NO_x emission (16.8 Tg N yr⁻¹).”

Line 116-124:

“Rasool et al.⁴⁶ and Luo et al.⁴⁷ employed DayCENT (Daily time-step version of the CENTURY biogeochemical model) and FEST-C (Fertilizer Emission Scenario Tool for CMAQ) models to assess soil HONO and NO emissions across the United States. Their assessments were based on the proportion of soil HONO to NO emissions, along with the relationship between soil HONO emissions and SWC established by Oswald et al.¹⁴. Our estimated annual emission of soil reactive oxidized nitrogen (HONO + NO) in the United States is 0.85 Tg N yr⁻¹, which is similar to the value in Luo et al.⁴⁷ (0.69 Tg N yr⁻¹). Besides, our estimated HONO emissions from soils in North America is 0.90 Tg N yr⁻¹, which is close to the estimate of 0.83 Tg N yr⁻¹ provided by Wu et al.⁵.”

That being said, we acknowledge that there is uncertainty in our estimates, and detailed descriptions are provided in the SI text.

Line 501:

“The potential uncertainties of soil emissions are discussed in the Supplementary text.”

Supplementary text in revised SI:

“Our study estimates the long-term trends of global soil HONO emissions based on comprehensive dataset of existing global soil HONO emission measurements and quantifies their subsequent impact on global air quality. However, we acknowledge potential uncertainties in our estimates HONO emissions, particularly in regions lacking direct measurements of the soil HONO flux such as in North America. Only when such measurements become available can we compare our estimated emissions with observed values to quantify the uncertainties. Additionally, our estimates use the average emissions from soil samples of different land-use types within specific latitudinal bands and longitudinal columns to represent corresponding regional and land-use type emissions. This approach introduces uncertainty due to heterogeneous emissions across different locations. Moreover, our parameterization does not include

factors such as soil pH and texture, adding to the uncertainty in our estimates of HONO emissions. Furthermore, when handling soil emissions post-fertilization soil emissions, we directly used MERRA2 soil moisture data, which may not fully reflect changes in soil moisture due to irrigation, thereby introducing an uncertainty in our estimation of global soil HONO emissions. Despite these limitations, our soil HONO emission estimates consider all controlling factors on which there is consensus, including soil temperature, soil water content, land cover, fertilizer consumption, and soil pH (only for fertilizer-induced soil HONO emission). Therefore, we believe that our estimated global soil HONO emissions have included most (if not all) available information across various regions and represent the state-of-the-art knowledge on soil HONO sources.”

7. The impacts of soil HONO emissions on atmospheric composition are (somewhat surprisingly) large in Figure 4. Figures S5-S8 show dramatic impacts on OH and O₃ in the model. Given that HONO rapidly photolyzes to form OH, it’s not surprising that OH would increase directly, but that initial OH molecule would likely be quickly lost to surface uptake or reaction with a volatile organic compound. In low NO_x environments, that’s unlikely to have a big impact on ozone production. One assumes that the release of NO from photolyzed HONO leads to a bigger (and more regional) impact on the oxidative budget of the atmosphere in these simulations, but that information is missing from the manuscript. It would be helpful to know more about the treatment of emissions (is there any canopy reduction factor applied?) and what the height of the lowest model layer is.

Response 1.7:

(1) We agree with the reviewer’s comment on the comprehensive impact of HONO on atmospheric oxidation. In fact, as we wrote in lines 217-219 in the original manuscript (Line 259-262 in the revised manuscript), soil HONO emissions not only increase OH concentrations but also elevate NO_x levels, thereby promoting O₃ formation in low-NO_x environments. We analyzed NO_x concentrations across different cases, and the results show that soil HONO emissions increase NO_x concentrations, especially in the cleaner southern hemisphere (Figure R4). We have added Figure R4 to the revised SI as Figure S8a,b and mentioned it in the revised manuscript.

Figure R4. Global impact of soil HONO emissions on NO_x in 2016. (a) and (b) represent the absolute and relative increases of NO_x due to soil HONO emissions, respectively.

Line 251-254:

“Consequently, most regions in the southern hemisphere are characterized by a NO_x-limited regime for O₃ production, and soil HONO emissions promote O₃ formation by increasing the NO_x concentrations in these regions (Supplementary Fig. S8a,b), in addition to directly increasing OH radicals.”

(2) We thank the reviewer for the insightful comment on the canopy reduction. In our original manuscript, we did not account for the impact of canopy reduction on soil emissions. In the revised manuscript, we have calculated the canopy reduction factor (CRF) using leaf area index (LAI) values following Yan et al.⁷; detailed information of the CRF calculation has been added in the revised manuscript, section “Impact of canopy reduction” in “Methods”.

We calculated a global average CRF of 0.79, which falls within the range estimated by previous studies (0.67 to 0.87)^{8,7}. The global distribution of CRF is also consistent with the previous results, with the smallest CRF in the tropical rainforests of the southern hemisphere (Figure R5). Subsequently, we re-calculated the soil emissions of HONO and NO that can reach the atmosphere post-canopy reduction for all years (1980–2016) and we found that the soil emissions of HONO and NO after considering the canopy effect are 11.5 and 6.2 Tg N yr⁻¹, respectively, which are 86% and 85%, respectively, of the estimates without considering the canopy effect. Following the adjustment of soil emissions, we re-ran all model cases, re-plotted all figures, and re-calculated all values to assess the impact of soil HONO emissions on air quality. Since almost all figures and numbers have been updated, please refer to the revised manuscript for the corresponding edits.

Figure R5. Annual average of the canopy reduction factor (CRF) in 2016. A larger CRF indicates a smaller canopy uptake.

We have added Figure R5 to the revised SI as Figure S2 and included the following description in the Methods and Results section of the revised manuscript.

Line 490-501:

“Impact of canopy reduction

Soil HONO (NO) emissions are impacted by canopy reduction, a process mainly driven by diffusion through plant stomata and direct deposition on the leaf cuticle, which is frequently quantified by the canopy reduction factor (CRF). We used the following formula to adjust F_{emis} to account for canopy reduction effects to yield the soil emissions to the atmosphere (F_{canopy}):

$$F_{canopy} = F_{emis} \times CRF = F_{emis} \times \frac{e^{-k_s \times SAI} + e^{-k_c \times LAI}}{2} \quad (7)$$

where k_s and k_c are stomata and cuticle absorptivity constants, set at 8.75 and 0.24 $m^2 m^{-2}$, respectively⁹. SAI and LAI are the stomatal area index and leaf area index, respectively. The LAI data is sourced from the MERRA2 dataset, while the SAI is derived from the SAI:LAI ratio across different land cover types according to Yienger and Levy⁹, expressed as:

$$SAI = \frac{SAI_{YL95}}{LAI_{YL95}} \times LAI_{MERRA2} \quad (8)$$

The potential uncertainties of soil emissions are discussed in the Supplementary Information.”

Line 79-85:

“We calculated the canopy reduction factor (CRF) using leaf area index (LAI, see Methods), yielding a global average CRF of 0.79, which falls within the range estimated by Vinken et al. (0.87)⁸ and Yan et al. (0.67)⁷. The global distribution of CRF shows that it is lowest in the tropical forests of South America and Africa (Supplementary Fig. S2), indicating a substantial reduction in soil emissions due to the dense canopies in these tropical regions. The application of the CRF decreases the global total soil HONO and NO emissions to 11.5 and 6.2 Tg N yr⁻¹, respectively. (Fig.1 and Supplementary Fig. S1).”

(3) The lowest height of the model is about 100 m, and we have incorporated this information into the revised manuscript.

Line 505:

“The lowest level of the model is about 100 m”

8. For Figure 1, consider changing the maximum of the colour scale in panels d-g to saturate at e.g. 1.5 jg N ha⁻¹ mth⁻¹ to better match the equivalent amounts in panel a, which is for a year, and to provide more colour contrast in the panels

Response 1.8:

We appreciate the reviewer’s feedback and have made the necessary adjustments to Figure 1 as suggested (shown as Figure R6 below).

Figure R6. Global soil HONO emissions. (a) Global annual average soil HONO emissions in 2016, in units of $\text{kg N ha}^{-1} \text{yr}^{-1}$. (b) Proportions of global emissions from different regions. Magenta, blue, orange, purple, green, light blue, brown, and gold sectors indicate proportions of North America, South America, Europe, Africa, India, China, Australia, and the rest of the world (ROW), respectively. (c) Relative contributions of fertilized and unfertilized soil to the total HONO emissions, with orange and green sectors indicating their respective proportions. (d–g) Monthly variation of soil HONO emissions in 2016, in units of $\text{kg N ha}^{-1} \text{mth}^{-1}$.

9. For Figure 2, the units are very inconsistent and hard to follow. Given that the emissions are being expressed as Tg y^{-1} , why use two very different types of units to visualize the trends? The rates of change of emissions should be Tg y^{-1} , or if those result in inconveniently small values, use Gg instead (rather than molecules cm^{-2} , $\text{s}^{-1} \text{y}^{-1}$ or kt y^{-1}).

Response 1.9:

We have revised the unit of Figure 2 to Mg/yr to address the inconsistency in units (shown as Figure R7 below).

Figure R7. Rates of change in total (a, b), fertilized (c, d), and unfertilized (e, f) soil HONO emissions from 1980 to 2016. (a, c, e) Rates of change ($\text{g N ha}^{-1} \text{yr}^{-1} \text{yr}^{-1}$) in total soil HONO emissions, fertilization-introduced emissions, and emissions during non-fertilization period, respectively. The figures display the results of linear regression analysis conducted on emission data for each grid from 1980 to 2016. The slopes of the regression are visually presented in the figures. Panels (b, d, f) Time-series of soil HONO emissions for each region ($\text{Gg N ha}^{-1} \text{yr}^{-1} \text{yr}^{-1}$). Magenta, blue, orange, purple, green, light blue, brown, and gold solid lines represent emissions pertaining to North America, South America, Europe, Africa, India, China, Australia, and the rest of the world (ROW), respectively, while dashed lines represent the rates of change.

Response to Reviewer #2

We thank the reviewers for the comments and suggestions (in black font) on the manuscript. Our responses (in blue) and the corresponding edits in the manuscript (in red) are shown below.

Comments:

This manuscript is an important one in bringing attention to the subject of an overlooked but critical trace gas nitrous acid on global atmospheric chemistry and particularly the oxidation capacity in rural environments. It is definitely a topic that is worthy of publication in Nature and, in my opinion, should be published, subject to some considerations outlined below (i.e. minor/moderate revisions). I realize that it may not be feasible to address every single comment, so I am stipulating that not every one is mandatory.

Response 2.1:

We thank the referee for appreciating our efforts and for the comprehensive review on our manuscript. Following the reviewer's constructive comments and the editor's suggestion, we have taken all of the reviewer's comments/suggestions into consideration when preparing this response and revising the manuscript. Below are our responses and the corresponding edits to each comment.

1. I believe this manuscript would be more effective if it were to directly compare the modeling output with the output from other global models who have tried to include HONO (e.g., Elshorbany et al., 2014; Rasool et al., 2019). It would also be best to explicitly list the many different sources of HONO that are included in the model, as there appear to be a dizzying array of proposed sources in the literature including:

- 1) Abiotic equilibrium from $\text{NO}_2^- + \text{H}^+$ in acidic soils: Su et al. (2011);
- 2) Photosensitized reduction of NO_2 on surfaces with organics like humic acid: Kleffmann (2007); Song et al. (2022)
- 3) Heterogeneous disproportionation of NO_2 with water producing HONO + HNO_3 : Kleffmann (2007);
- 4) Heterogeneous photolysis of NO_3^- on ground or in aerosol phase: Zhou et al. (2003); Ye et al. (2017); Andersen et al. (2023).
- 5) Homogeneous photo-oxidation of HNO_3 (g): Song et al., STE, (2023).

Because global models of HONO are relatively rare, I think the manuscript would benefit tremendously from the addition of a global budget breakdown of HONO sources and sinks and overall lifetimes, etc. This would give the reader a sense of how large, for example, the agricultural soil source is with respect to other leading sources and sinks (e.g. Wang et al., ES&T, 2023; Song et al., Env. Pol., 2022). Specifically, Zhang et al. (2023) claim that direct emissions of HONO (especially from livestock farming) make up ~40% of the total production in rural areas across the North China Plain.

Response 2.2:

(1) So far, there are only four studies (Elshorbany et al.^{48,49}, Wu et al.⁵ and Ha et

al.⁵⁰) that utilized chemical-transport model (CTM) to assess the impact of HONO on global air quality or reported global soil emissions. Below, we introduce the novelty of our work in the context of these four studies.

(a) HONO sources apart from soil emission.

Elshorbany et al.^{48,49} used a fixed atmospheric HONO/NO_x ratio of 0.02 to estimate global HONO concentrations, which inherently introduced significant uncertainties, because as the reviewer suggested, there are many sources for atmospheric HONO and the ambient ratio of HONO/NO_x is not a fixed value. (The reference list is attached at the end of this response letter to all reviewers.)

As mentioned in line 419-423 in the original manuscript (line 531-535 in the revised manuscript), our study incorporated various sources of HONO into the model, including soil emissions, vehicle emissions, biomass burning emission, gas-phase reactions of NO and OH, RH-dependent and light-enhancing heterogeneous reactions of NO₂, and the photolysis of particle nitrate. Overall, our treatment of the HONO sources (apart from soil emissions) includes the dominant sources of HONO and represents a significant improvement in simulating global atmospheric HONO. Detailed reactions and parameters are listed in Table S7 (shown as Table R1 below). We have also modified the corresponding text to clearly state which sources of atmospheric HONO are considered in our study. Please refer to line 529-551 in the revised manuscript for the details.

(b) Soil HONO emissions.

Wu et al.⁵ and Ha et al.⁵⁰ are the two existing studies that reported global soil HONO emissions, but neither of them quantify the soil HONO impact on global air quality. Wu et al.⁵ estimated global soil HONO emissions by considering the number of wet-dry cycles (one precipitation event) in a year without considering the impact of real-time variations in soil moisture on soil emissions; however, Wu et al.⁵ did not report the soil HONO impact on global air quality. Ha et al.⁵⁰ employed a fixed HONO/NO_x ratio of 0.1 for estimating soil HONO emissions based on soil NO sources; but they only report the global effects of HONO from all sources without specifically calculating the global soil HONO impacts. Therefore, the global soil HONO impacts on global air quality remained unquantified until the current work.

In our study, we estimated global soil HONO emissions based on all available soil HONO flux studies and considered all control factors on which there is consensus, including soil temperature, soil water content, land cover, fertilizer consumption, and soil pH (only for fertilizer-induced soil HONO emission). Our treatment on global soil HONO represents the state-of-the-art soil HONO emission estimates. We estimated the global soil HONO emission (after considering the canopy reduction effect) to be 11.5 Tg N yr⁻¹, which is comparable to the value of 9.7 Tg N yr⁻¹ obtained by Wu et al.⁵. We further implemented the global soil HONO emissions in a global model (CAM-Chem) and, for the first time, specifically quantified the atmospheric impacts of soil HONO on a global scale.

On a regional perspective, the reference mentioned by the reviewer, Rasool et al.⁴⁶ and Luo et al.⁴⁷ has estimated the soil HONO emission in the United States based on the ratio of soil HONO to soil NO emissions and the relationship between soil HONO

emissions and SWC from Oswald et al.¹⁴. Our estimated annual emission of soil reactive oxidized nitrogen (HONO + NO) in the United States is 0.85 Tg N yr⁻¹, which is in line with that in Luo et al.⁴⁷ (0.69 Tg N yr⁻¹). Please note that our soil HONO emission estimation considers all critical control factors and therefore in principle provides a realistic representation of ambient conditions.

Table R1 Parameterized HONO source mechanisms included in the model.

Sources	HONO formation reactions	Formula	Parameter range	Parameters in this study
Vehicle emissions ³⁴	The ratio of HONO to NO _x from traffic sources	2.3% NO _x of transportation sources	0.29%–2.3% ³⁵⁻³⁸	2.3%
Biomass burning	The ratio of HONO to NO _x from biomass burning	0.23 NO _x of t biomass burning	0.025–0.23 ⁴⁰⁻⁴²	0.23
Homogeneous reaction ³⁴	NO + OH → HONO	JPL dataset (Burkholder et al. ⁴⁰)		
NO ₂ heterogeneous reaction	NO ₂ + aerosol + RH → HONO	$k = \frac{1}{4} \times v_{NO_2} \times Sa \times \gamma_{NO_2} \times f_{RH} *$	γ_{NO_2} in the range of 1×10^{-6} – 2.6×10^{-5} ⁵¹⁻⁵³	8×10^{-6}
	NO ₂ + aerosol + RH + hv → HONO	$k = \frac{1}{4} \times v_{NO_2} \times Sa \times \gamma_{NO_2} \times f_{RH} \times \frac{\text{light intensity}}{400} **$	γ_{NO_2} in the range of 1.35×10^{-5} – 1.3×10^{-4} ^{51,54}	1×10^{-4}
	NO ₂ + ground + RH → HONO	$k = \frac{1}{8} \times v_{NO_2} \times \frac{2 \times LAI}{H} \times \gamma_{NO_2} \times f_{RH}$	γ_{NO_2} in the range of 1×10^{-6} – 1×10^{-5} ⁵⁵⁻⁵⁷	8×10^{-6}
	NO ₂ + ground + RH + hv → HONO	$k = \frac{1}{8} \times v_{NO_2} \times \frac{2 \times LAI}{H} \times \gamma_{NO_2} \times f_{RH} \times \frac{\text{light intensity} **}{400}$	γ_{NO_2} in the range of 2×10^{-6} – 6×10^{-5} ^{29,58,59}	6×10^{-5}
Particulate photolysis ^{34,60}	NO ₃ ⁻ + hv → HONO	$J_{pNO_3} = EF \times J_{HNO_3(CAM-Chem)}$	1–120 ^{34,60,61}	$\frac{8.3 \times 10^{-5}}{7 \times 10^{-7}}$

$$* f_{RH} = \begin{cases} \frac{RH}{50} & (RH < 50) \\ \frac{RH}{10} - 4 & (50 \leq RH \leq 80) \\ 4 & (RH \geq 80) \end{cases}$$

** To account for the photo enhancing effect, we set 1×10^{-4} for γ_{NO_2} during daytime conditions when the light intensity was below 400 W m⁻²; when the light intensity exceeded 400 W m⁻², we adjusted it linearly in proportion to solar radiation.

Line 508-524

“In the standard CAM-Chem model, there is no HONO production nor loss processes. Here, we incorporated direct emission sources and secondary formation pathways for HONO in the CAM-Chem, including traffic emission, biomass burning emission, homogeneous reaction of NO and OH, RH dependent and light-enhancing heterogeneous reactions of NO₂ on aerosol and ground surfaces, and photolysis of particulate nitrate^{33,34}. The specific reactions and parameters for these sources utilized

in the current study are provided in Supplementary Table S7. For HONO emissions from exhaust sources, the HONO to NO_x ratios were found to vary significantly, ranging from 0.29% to 2.3% for vehicle exhausts³⁵⁻³⁸ and from 3% to 6% for commercial aircraft³⁹. In this study, we assumed a HONO/NO_x ratio of 2.3% for all traffic (land vehicle, ship, and aircraft exhausts) HONO emissions. For biomass burning, HONO/NO_x ratios have been reported to range from 0.025 to 0.23⁴⁰⁻⁴². We used the upper bound value of 0.23 for the HONO emissions from biomass burning. The use of the upper limits of these parameters can help that we provide a conservative estimate of the soil HONO impacts compared to these two direct sources. For secondary chemical reactions, the parameters, such as NO₂ uptake coefficient (γ_{NO_2}) and nitrate photolysis rate constant (J_{pNO_3}), exhibit a broad range of variation, with differences in an order of magnitude. The secondary reaction parameters used in our research fall within the range reported previously (Supplementary Table S7) which have been adopted by many prior studies.”

Line 116-124:

“Rasool et al.⁴⁶ and Luo et al.⁴⁷ employed DayCENT (Daily time-step version of the CENTURY biogeochemical model) and FEST-C (Fertilizer Emission Scenario Tool for CMAQ) models to assess soil HONO and NO emissions across the United States. Their assessments were based on the proportion of soil HONO to NO emissions, along with the relationship between soil HONO emissions and SWC established by Oswald et al.¹⁴. Our estimated annual emission of soil reactive oxidized nitrogen (HONO + NO) in the United States is 0.85 Tg N yr⁻¹, which is similar to the value in Luo et al.⁴⁷ (0.69 Tg N yr⁻¹). Besides, our estimated HONO emissions from soils in North America is 0.90 Tg N yr⁻¹, which is close to the estimate of 0.83 Tg N yr⁻¹ provided by Wu et al.⁵”

(2) In the global modeling practice, due to the vertical transport between different layers, it is common to consider the total burden of a certain species within the entire troposphere when calculating the budget. We calculated HONO budget in the troposphere in 2016 (Figure R8). The results indicate that the primary source of HONO within the troposphere is chemical formation, contributing a substantial 93%. This is followed by HONO emissions from soil, which account for 6%. In contrast, direct emissions of HONO from vehicle exhaust and biomass burning are both less than 1%. This indicates that although soil emissions occur only at the surface layer, their contribution to the tropospheric HONO remains significant and should not be overlooked. We have added Figure R8 to the revised SI as Figure S10 and included descriptions in the revised main text.

Please note that we did not consider livestock farming as a source of HONO in our study, because there is not enough global evidence of this source. Therefore, we could not evaluate the relative importance of this source on a global scale. However, we would like to emphasize that our study focuses on providing the first global soil HONO emission in the past four decades and quantifying soil HONO impacts on global air quality. Our results remain important regardless of considering the livestock farming

HONO. We have clearly clarified it in the revised text.

Figure R8. Tropospheric HONO budget and relative contribution from different sources in 2016. (a) The contributions of different sources to tropospheric HONO in different months of 2016. (b) Annual averaged relative contribution of each HONO source to tropospheric HONO.

Line 236-242:

“We performed a calculation of the HONO budget for the troposphere in 2016 (Supplementary Fig. S10). The results reveal that the predominant source of HONO within the troposphere is chemical formation, representing a significant 93%. Following this, HONO emissions from soil contribute 6%. In comparison, anthropogenic sources from vehicle exhaust and biomass burning each represent less than 1%. This suggests that while soil emissions only occur at the surface, their contribution to the overall HONO in the troposphere should not be overlooked.”

Line 524-526:

“We did not consider HONO emissions from livestock farming⁴³ nor from biocrusts⁴⁴, because so far there is not sufficient relevant data for deriving global HONO emissions from these two new sources.”

2. The authors make a rather extraordinary claim that the modern global soil source of HONO is about 13.4 Tg N/yr (up 22% from 1980 levels). I say this is extraordinary because it is a direct source of NO_x to the atmosphere (once photolyzed in the daylight) and most recent models of ozone report overall NO_x budgets that amount to about 50 Tg N/yr (e.g. Hu et al., 2017), with approximately 20% (9.7 TgN/yr) coming from soil NO_x (as NO). Vinken et al. (2014) summarize over a dozen attempts to quantify the global emissions of soil NO_x reporting a range from 4 – 15 TgN/yr, proposing their own best guess at 13.9 TgN/yr. Looking into the model used in this study, CAM-Chem, according to Lamarque et al., GMD, (2012) for the year 2010 they report a soil source of 8 Tg NO-N/yr (out of a total of 41.7 Tg NO-N/yr.) This is very close to the value reported in your study on line 83 (7.3 Tg N/yr). This soil NO_x parameterization is in

addition to the new HONO parameterization, which means that the claim of this work is that there is a soil source of nitrogen oxides to the atmosphere that approximately double what is currently assumed in a range of models. This means a change in the relative source strength of worldwide soils from approximately 20% – 40% of the global NO_x budget. This is a remarkable change in our understanding, and I think it is incumbent on the authors to discuss how this shift might be true and might have been overlooked for so long. For example, might many of the “top-down” estimates summarized in Vinken et al. (2014) be counting HONO + NO from soils? Might this mean that there needs to be changes configured into the model soil parameterizations for NO emissions to accommodate this new, large HONO source? Maybe your model’s combined NO + HONO emissions (20.7 Tg N/yr) is possibly most in-line with the Steinkamp & Lawrence (2011) result reported using the arithmetic mean value (27.6 Tg N/yr), which has been an outlier in past intercomparisons (Vinken et al., 2014).

Response 2.3:

We thank the reviewer for the insightful comment on soil reactive nitrogen emission in the context of global NO_x budget.

(1) Indeed, the soil HONO emission is a soil reactive nitrogen source (in addition to soil NO) that should be considered in reactive nitrogen budget. This has been proposed by many previous studies^{3,5,13,16}.

The only question that remains open to discussion is how important this additional soil reactive nitrogen source is in the global reactive nitrogen budget. To our best knowledge, we have to rely on laboratory experiments to quantify the impacts of various factors on soil reactive nitrogen fluxes. Among these lab studies, only the ones that use the same experimental setup and simultaneously measure HONO and NO emissions from soil samples could provide insights on the relative importance of HONO and NO in the soil reactive nitrogen emissions. There are only a finite number of this type of studies, and among these studies, even though there were a few samples showing larger soil NO than HONO emissions, there were more samples indicating larger soil HONO than NO or comparable emissions. By examining all samples in respective research, some studies (e.g., Huang et al¹; Xue et al²; Wang et al^{3,16}) concluded that soil HONO emissions exceeds soil NO emissions, and the others (e.g., Oswald et al¹⁴; Wu et al⁶²) indicated that soil HONO and soil NO are comparable. In the current study, all available soil HONO emission experiments (including those mentioned above) have been utilized in compiling the global soil HONO emissions, and our global-integrated annual-average results suggest that soil HONO emission is larger than soil NO emission.

(2) As to the fraction of soil reactive nitrogen source in the global reactive nitrogen budget, here is our calculation. As the reviewer mentioned, the current estimates (without soil HONO) suggest that about 20% (9.7 out of 50 Tg N yr⁻¹) comes from the soil.

Following the reviewer’s suggestion on canopy reduction effect (see our Response 2.7 for details), we have included the canopy reduction factor (CRF) to re-calculate the global soil emissions, and our updated global soil emissions of HONO and NO are 11.5

and 6.2 Tg N yr⁻¹, respectively. Please note that not all of HONO is photolyzed to form NO_x, because it is partly deposited to the ground surface. Nevertheless, if we consider an upper limit (all HONO is instantaneously photolyzed to form NO_x), our calculated upper-limit fraction of soil reactive nitrogen in global NO_x budget would be $(11.5+6.2)/(50-9.7+11.5+6.2)=30.5\%$, which is higher than the current estimate of 20% (without considering soil HONO) but lower than the reviewer's assumption of 40%.

(3) The reviewer's suggestion on the possibility that previous top-down estimate of NO_x budget might have treated both soil HONO and NO as soil NO_x is very interesting. Considering the same emission sector and location of soil HONO and soil NO, we are inclined to believe that the top-down estimates of NO_x based on NO₂ columns include both the direct soil NO emissions and the photolyzed NO from soil HONO emission. Their estimated global NO_x emission is 12.9 ± 3.9 Tg N yr⁻¹ which could be considered as a range from 9.0 to 16.8 Tg N yr⁻¹, while our updated estimate for global soil emission of HONO + NO is 17.7 (11.5 + 6.2) Tg N yr⁻¹, which is actually very close to their upper estimate. We have added a few sentences in the revised manuscript to discuss this possibility.

(4) Regardless of the treatment of top-down estimates on soil reactive nitrogen, we would like to emphasize that including soil HONO emissions in regional and global atmospheric modeling is essential in reproducing atmospheric compositions and oxidation processes. This is the main purpose of our study.

Line 372-374:

“To aid the development of the global soil HONO emissions, we compiled a comprehensive dataset of published measurements of soil HONO emissions from diverse ecosystems worldwide (Supplementary Table S4).”

Line 84-93:

“The application of the CRF decreases the global total soil HONO and NO emissions to 11.5 and 6.2 Tg N yr⁻¹, respectively. (Fig.1 and Supplementary Fig. S1). The estimated soil HONO emission is comparable to the value of 9.7 Tg N yr⁻¹ obtained by Wu et al.⁵. The soil NO emissions is slightly higher than the estimate of 5.5 Tg N yr⁻¹ calculated by Yienger and Levy⁹ using the YL95 model, but lower than the 9.0 Tg N yr⁻¹ estimated by Hudman et al.⁶ using the BDSNP model, and the 12.9 ± 3.9 Tg N yr⁻¹ estimated by Vinken et al.⁸ using a top-down approach. Overall, the soil NO emissions in our study fall within the range (4–21 Tg N yr⁻¹) reported in previous studies⁶⁻⁸. Note that the top-down estimate method may include both soil HONO and soil NO as soil NO_x, our estimate for global soil emissions of HONO + NO totals 17.7 Tg N yr⁻¹, which closely aligns with the upper estimate of top-down soil NO_x emission (16.8 Tg N yr⁻¹).”

3. Furthermore, I wonder what HONO absorption cross section is used in your model, and what a 30% smaller cross section would do to your analysis (model and obs

intercomparisons). Li et al., ES&T (2024) have recently claimed that past photolysis rates of HONO are 20%-30% too large throwing a lot of uncertainty into our understanding of global HONO budgets. I bring all of these issues up in order to urge the authors to strongly consider attempting to present some reasonable estimate of an uncertainty associated with this new global result.

Response 2.4:

The CAM-Chem model used the recommended HONO absorption cross section from the JPL dataset (Burkholder et al.⁴⁵), i.e., the ones reported by Stutz et al.⁶³. Despite some differences at peaks, the results from Stutz et al.⁶³ are generally consistent with those reported by Li et al.⁶⁴ between 360 and 390 nm wavelength (Figure R9a). In CAM-Chem model, absorption cross sections of gaseous species (including HONO) are averaged over 5 nm intervals to enhance computational efficiency. After averaging, the results reported by Li et al.⁶⁴ and those by Stutz et al.⁶³ are actually very similar, with an average difference <1% (Figure R9b).

In fact, the reference mentioned by the reviewer, Li et al.⁶⁴, showed the photolysis rate constants for HONO using their own cross sections and those from previous studies (including those by Stutz et al.⁶³). The difference between the photolysis rates by Li et al.⁶⁴ and Stutz et al.⁶³ is very small, with those from Stutz et al.⁶³ falling within the error range of Li et al.⁶⁴ (Figure R10), indicating that the values used in our study are not overestimated compared to Li et al.⁶⁴.

Figure R9. Comparison of HONO absorption cross section (360–390 nm) from Stutz et al.⁶³ (used in CAM-Chem model) and Li et al.⁶⁴. (a) the original HONO absorption cross section values, (b) the averaged values with a 5 nm interval (used in CAM-Chem).

REDACTED

Figure R10. HONO photolysis rate constants from the studies by Stutz et al.⁶³ and Li et al.⁶⁴. This figure is sourced from Li et al.⁶⁴.

We have included Figure R9 in the revised SI as Figure S19 and incorporated the following discussion in the revised manuscript.

Line 526-528:

“The HONO absorption cross section used by the CAM-Chem model is based on the values recommended by Burkholder et al.⁴⁵, which is consistent with the recent laboratory measurements of Li et al.⁷² (with difference <1%)(Supplementary Fig. S19).”

4. One other modeling effort that I am aware of is the agroecosystem model of Fertilizer Emission Scenario Tool for the CMAQ (FEST- C) developed at Rice University (Rasool et al., 2019; Luo et al., 2022). It seems to me that it would be worthwhile to compare the emissions from this work to that of another one such as Rasool et al. (2019) to see how they might compare.

It would also help shore up confidence in the modeling if you were to compare known characteristics of the seasonal and diurnal dependence of the HONO concentrations with those from measurements. That is, are the max:min ratios similar for night:day and winter:summer to what has been measured, at least at one or two sites.

Response 2.5:

(1) As in the Response 2.2, we have included comparisons of our results with previous studies in the revised manuscript, including comparisons with the results of Rasool et al.⁴⁶ and Luo et al.⁴⁷. Briefly, our method estimates the annual emissions of soil reactive oxidized nitrogen (HONO + NO) in the United States to be 0.85 Tg N yr⁻¹, which is in the same magnitude as that from Luo et al.⁴⁷ (0.69 Tg N yr⁻¹). We also compare our findings and other previous studies. Our calculated global soil HONO emission after considering the canopy reduction factor is 11.5 Tg N yr⁻¹, which is comparable to the value of 9.7 Tg N yr⁻¹ obtained by Wu et al.⁵; for North America, our estimate of 0.90 Tg N yr⁻¹ closely matches 0.83 Tg N yr⁻¹ in their study.

For global soil NO emissions, our estimate is 6.2 Tg N yr⁻¹, which is well within the range reported by previous studies (4-21 Tg N yr⁻¹).

These comparisons demonstrate the reliability of our methodology. We have added the following discussions in the revised manuscript.

Line 84-93:

“The application of the CRF decreases the global total soil HONO and NO emissions to 11.5 and 6.2 Tg N yr⁻¹, respectively. (Fig.1 and Supplementary Fig. S1). The estimated soil HONO emission is comparable to the value of 9.7 Tg N yr⁻¹ obtained by Wu et al.⁵. The soil NO emissions is slightly higher than the estimate of 5.5 Tg N yr⁻¹ calculated by Yienger and Levy⁹ using the YL95 model, but lower than the 9.0 Tg N yr⁻¹ estimated by Hudman et al.⁶ using the BDSNP model, and the 12.9 ± 3.9 Tg N yr⁻¹ estimated by Vinken et al.⁸ using a top-down approach. Overall, the soil NO emissions in our study fall within the range (4–21 Tg N yr⁻¹) reported in previous studies⁶⁻⁸. Note that the top-down estimate method may include both soil HONO and soil NO as soil NO_x, our estimate for global soil emissions of HONO + NO totals 17.7 Tg N yr⁻¹, which closely aligns with the upper estimate of top-down soil NO_x emission (16.8 Tg N yr⁻¹).”

Line 116-124:

“Rasool et al.⁴⁶ and Luo et al.⁴⁷ employed DayCENT (Daily time-step version of the CENTURY biogeochemical model) and FEST-C (Fertilizer Emission Scenario Tool for CMAQ) models to assess soil HONO and NO emissions across the United States. Their assessments were based on the proportion of soil HONO to NO emissions, along with the relationship between soil HONO emissions and SWC established by Oswald et al.¹⁴. Our estimated annual emission of soil reactive oxidized nitrogen (HONO + NO) in the United States is 0.85 Tg N yr⁻¹, which is similar to the value in Luo et al.⁴⁷ (0.69 Tg N yr⁻¹). Besides, our estimated HONO emissions from soils in North America is 0.90 Tg N yr⁻¹, which is close to the estimate of 0.83 Tg N yr⁻¹ provided by Wu et al.⁵.”

(2) As to the comparison of our simulated HONO mixing ratios with the observations, in the original manuscript, we compared the simulated HONO concentrations with the observations from dozens of global locations (Figure R2). We found a significant improvement in the simulation of HONO when soil HONO emissions are incorporated into the model, compared to the simulation without soil HONO. Specifically, when soil HONO emissions are considered, the normalized mean bias (NMB) between the simulated and observed average HONO concentration significantly improves from -49% to -25%.

Furthermore, we selected observations with detailed diurnal variation information, i.e., campaigns in Beijing (Figure R3a)¹⁰ and Jinan (Figure R3b)¹¹ which were both conducted in 2016 (the same as the simulation year), to further validate the model performance on simulating the diurnal HONO pattern (shown as Figure R3 in Response 1.2 to Reviewer #1). The comparison suggests that the diurnal patterns of simulated HONO in both cases align well with the observed data; when soil HONO is included, there is a significant improvement, with NMB values for Beijing and Jinan improving

from -48% to -25% and from -57% to -22%, respectively. We have included Figure R3 in the revised supplementary information as Figure S7 and referenced it in the main text.

The model validation results presented above strongly support the validity of the model results and indicate that incorporating soil HONO emissions enhances the model performance in simulating ambient HONO levels. To clarify the evaluations of our model, we have modified relevant texts and added the following texts in the revised manuscript.

Line 203-217:

“We compared the model simulations with atmospheric HONO measurements reported in the literature from 36 globally distributed sites across various years (Supplementary Fig. S6, Table S2). The results indicate that our simulated HONO mixing ratios fall within the range of the HONO observations in various years. When the soil HONO emissions were included, the normalized mean bias (NMB) between the simulated average and observed average HONO mixing ratios improved from -49% to -25%. To further evaluate the model performance on the diurnal variation, we selected two measurements with detailed diurnal information, both conducted in 2016 (Supplementary Fig. S7, the same as the simulation year), to compare the simulation results with the observed data. The findings indicate that the model has well reproduced the diurnal variation of ambient HONO mixing ratios after considering soil HONO emissions. By incorporating these soil emissions, the simulation results for 2016 at both sites showed significant improvement, with the NMB values for Beijing (Supplementary Fig. S7a) and Jinan (Supplementary Fig. S7b) improving from -48% to -25.0% and from -57% to -22%, respectively. These results indicate that including soil HONO emissions has improved the capability of the model to simulate atmospheric HONO.”

Specific Comments:

5. Line 34: Is this increase in global surface O₃ an annual average? 29% is the max for an annual average? or daily average? A maximum daily 8hr average might be more meaningful. Polluted regions will often have higher O₃ maxima but also lower overnight minima (due to greater NO titration), so the 24-hour average is not always the best metric for understanding health or photochemistry impacts.

Response 2.6:

29% is annual average for a single grid. We have clarified this in the revised manuscript. Since we are interested in the overall ozone concentration at a regional scale, rather than the health effects of ozone exposure, we used the annual or monthly average ozone levels to demonstrate the overall impact of soil emissions.

Line 246-247:

“The maximum increase in the annual average O₃ level for a single grid reached up to 29%.”

6. Lines 82/83: Does your parameterization scheme have a built-in canopy factor to account for deposition of NO₂ in the canopy? Or do you have emissions and dry

deposition and the model just adds them together for a net flux? Is it treated the same in the HONO parameterization?

Response 2.7:

(1) We thank the reviewer for bringing up the canopy factor. In the original manuscript, we did not consider such canopy reduction factor in soil HONO emission.

In the revised manuscript, we have calculated the canopy reduction factor (CRF) using leaf area index (LAI) values following Yan et al.⁷; detailed information of the CRF calculation has been added in the revised manuscript, section “Impact of canopy reduction” in “Methods”.

We calculated a global average CRF of 0.79, which falls within the range estimated by previous studies (0.67 to 0.87)^{8,7}. The global distribution of CRF is also consistent with the previous results, with the smallest CRF in the tropical rainforests of the southern hemisphere (shown as Figure R4 in Response 1.7 to Reviewer #1). Subsequently, we recalculated the soil emissions of HONO and NO that can reach the atmosphere post-canopy reduction for all years (1980-2016) and we found that the soil emissions of HONO and NO after considering the canopy effect are 11.5 and 6.2 Tg N yr⁻¹, respectively, which are 86% and 85%, respectively, of the estimates without considering the canopy effect. Following the adjustment of soil emissions, we re-ran all model cases, re-plotted all figures, and re-calculated all values to assess the impact of soil HONO emissions on air quality. Since almost all figures and numbers have been updated, please refer to the revised manuscript for the corresponding edits.

(2) As to the dry deposition, indeed, CAM-Chem model has a dry deposition module (independent from the emission module) that treats the dry deposition of HONO and other reactive nitrogen species.

7. Line 96: Since your model finds that the vast majority of HONO emissions arise from unmanaged ecosystems on the continents, are you aware of the work being done on estimating emissions from biocrusts in arid landscapes (Weber et al., 2015)? Also, how do you treat “legacy” N in the agricultural regions? Or more specifically, does your model generate more HONO emissions in agricultural areas during non-fertilizing periods than in other comparable climates?

Response 2.8:

(1) We are aware of the work by Weber et al.⁴⁴ about NO and HONO emissions from biocrusts. However, our study focused on HONO emissions from the soil and we do not consider biocrusts as soil. Furthermore, the measurements of HONO emissions from biocrusts are very limited. Therefore, our study did not include HONO emissions from biocrusts. We have clarified this in the revised manuscript.

Line 524-526:

“We did not consider HONO emissions from livestock farming⁴³ nor from biocrusts⁴⁴, because so far there is not sufficient relevant data for deriving global HONO emissions from these two new sources.”

(2) As for the treatment of “legacy” nitrogen in agricultural regions, we did not use specific treatment. Please note that for the key controlling factors that have enough

supporting evidence and quantified relationship with soil HONO, i.e., soil temperature, soil water content, and fertilizer consumption, we use specific parameterizations. But for the soil/climate factors that we cannot specifically parameterized (e.g., the legacy N as the reviewer mentioned), we use different parameterizations for different latitude band, longitude column, and land use types derived from soil samples from corresponding latitude, longitude, and land use type, where possible. This information was included in our original manuscript (Table S5) and is also in the revised SI (Table S5).

Indeed, as proposed in previous studies ^{3,14} and illustrated in original Figure S12 (Figure S16 in the revised SI), the emissions from agricultural soil are higher than those from other land use types, which can be attributed to the cumulative effects of residual nitrogen over time (“legacy” nitrogen as the reviewer termed), even during non-fertilizing periods.

We have added this discussion in the revised manuscript.

Line 414-417:

“As shown in Supplementary Fig. S16, the emissions from agricultural soil are higher than those from other land use types, even during non-fertilization periods, which can be attributed to the cumulative effects of residual nitrogen over time.”

8. Figures 2-5 are too small to be read easily. For the 6-plot figures, I suggest changing from 3 by 2 to 2 by 3 and increasing their size. Perhaps make Figure 5 vertical instead of horizontal. I would also suggest converting all figures to PNG format for better zooming readability.

Response 2.9:

Thanks for the detailed and constructive suggestions. We have modified the figures accordingly.

9. Line 123: The value of 19.2°C does not match the value in Figure S2 (which has the July peak T over 20°C).

Response 2.10:

Thanks for the careful review. The value of 19.2°C value represents the seasonal average of ST in the northern hemispheres. We have revised the sentence to make these points clearer.

Line 142-144:

“The peak values of ST in the northern and southern hemispheres were observed in July and January, respectively, with seasonal average temperatures during their respective summers reaching 19.2 °C and 24.0 °C and dropping to 3.2 °C and 14.8 °C in winter (Supplementary Fig. S4).”

10. Lines 230-232: These changes need to be defined at a particular model level (e.g. near-surface). If they are near the surface it should be noted that radiative forcing of tropospheric O₃ is weakest at the surface and strongest near the tropopause (Lacis et al., 1990).

Response 2.11:

The changes in O₃ described in our paper refer to near surface values. We have clarified this in the revised manuscript.

Line 261-263:

“Through long-range atmospheric transport, soil emissions led to an approximately 5% rise in near surface O₃ mixing ratios over the oceans of the southern hemisphere and the Antarctic region (Fig. 4d and Supplementary Fig. S11f).”

11. Lines 282/283: This is a significant claim that requires significant substantiation. Is there any data on AOT40 impacts on crop yield? There is definitely data on the significant increases in crop yield due to fertilizer amendments, which are very large. In order for there to be a significant offset would require an estimate of the two competing effects.

Response 2.12:

There are many studies indicating that ozone can damage vegetation and reduce crop yield⁶⁵⁻⁶⁷. Given the varying sensitivity of different crops to ozone concentrations, the formulas used to calculate the reduction in crop yield based on AOT40 differ accordingly. To accurately assess changes in crop yield, it is necessary to incorporate the distribution of different crops, which itself merits a full investigation. However, in the current study, we aim to provide the first historical global soil HONO emissions and their impacts on global atmospheric chemistry. Therefore, in the revised manuscript, we have only added the relevant references that link the AOT40 with crop yield without expanding too much discussion on this statement.

We completely agree with the reviewer that it is crucial to quantify the competitive relationship between the positive effects of fertilization on crop yield and the negative effects of ozone-induced yield reduction caused by fertilized soil emissions. Again, this is indeed another very interesting topic that warrants a separate, in-depth discussion. We plan to address this issue in future work. Nonetheless, in the current paper, we have revised the sentence to tune down our statement.

Line 313-315:

“This finding reveals that the increase in atmospheric O₃ levels caused by soil emissions may partly offset the benefits of fertilization on crop yield.”

12. Figure 6: The figure shows HONO emitted from nitrite in soil, but a more thorough picture of the source parent species might be worthwhile, as shown in Song et al., 2023, coming from NH₂OH and through an unknown enzymatic route from NO₂⁻. Further, aside from having more soil sources of HONO than this figure depicts, NO is also emitted in nitrification and denitrification cycles, not just from NO₂⁻. I therefore recommend a more complete pictorial representation of the processes in soils that generate NO + HONO.

Response 2.13:

We thank the reviewer for the detailed suggestions. Accordingly, we have revised Figure 6 (shown as Figure R11 below), which more comprehensively depicts the pathways of NO and HONO formation in the soil⁶⁸.

Figure R11. Schematic of soil reactive oxidized nitrogen emissions and their impact on atmospheric composition, air quality, and vegetation. Soil generates HONO and NO through nitrification (blue arrows), nitrifier denitrification (blue and yellow arrows), and denitrification (yellow arrows), and releases them into the atmosphere through soil–air exchange. The natural emissions (green arrows) exhibit an increasing trend under the combined effects of global warming and increased fertilization. Simultaneously, anthropogenic sources (black arrows) are projected to decrease in the future due to global emission mitigation measures. The results highlight the growing relative contribution of soil reactive oxidized nitrogen emissions to O₃ formation and the subsequent impact on vegetation.

13. Line 317: I think you should probably lay out what are the “anthropogenic HONO emissions” specifically because there are a wide variety of proposed HONO sources in the literature. For example, Song et al. (2022) report sources like: vehicle exhaust, pNO₃- photolysis, and wildfires.

Wildfires are increasing worldwide, so how might that impact future trends in HONO emissions? It might help to lay out a global budget of HONO as captured in your model to give readers an idea of what the approximate relative importance is of all of these varied sources/sinks.

Response 2.14:

(1) In line 317 of the original manuscript, by “anthropogenic HONO emissions”, we meant direct vehicle emissions”. We have clarified it in the revised manuscript.

Line 350-352:

“Supplementary Fig. S13 shows the changing trends in direct vehicle HONO emissions, indicating a continuous decrease in emissions from economically developed regions such as the United States and Europe.”

(2) Indeed, as the reviewer mentioned, wildfires are expected to increase in the future under the warming climate, suggesting that the HONO sources from this sector is expected to increase. We have added this information to the revised manuscript.

Line 355-356:

“Wildfires are expected to increase in the future under the warming climate, exacerbating the HONO emissions from this sector.”

(3) Please refer to Response 2.2 for our calculated global tropospheric budget of HONO.

14. Lines 351/352: How might the use of the average emission from each land-type translate into an uncertainty in the global emissions values reported here? I realize this is a difficult task but it is worthwhile to try to keep track of the uncertainties in your model for the community to better fit them into our understanding. In general, some sort of reckoning of uncertainty needs to be done in this work. I realize it can be quite challenging when there are so many fits and approximations involved, but it is important because this is an extraordinary claim you are making that could change the way the field considers soil NO_x emissions.

Response 2.15:

Please note that our soil HONO estimation integrates all existing global soil HONO emission measurements and considers the predominant controlling factors, e.g., a wide range of soil temperatures (5-55°C) and soil water content (SWC, 0-100%WHC) allowing to reflect the variations in real-world environments, fertilizer type and consumption, as well as soil pH (on fertilizer-induced HONO emission).

We want to emphasize that whenever possible, we use the local emission parameterization for each latitude, longitude, and landuse type. This information was included in our original manuscript (Table S5).

In lines 351-352 in the original manuscript, we wanted to emphasize that for the cropland and forest soils (land use types) in China and Europe (different longitude bands), there are many samples available, so we use the average emission parameterization for this particular land use type in this particular region; when such detailed information is not available, we use the latitude average; when latitude average is not available, we use global average.

Therefore, we are confident that our estimated global soil HONO emissions already include most (if not all) available information across various regions and in principle represent the state-of-the-art knowledge on the controlling factors of soil HONO sources.

That being said, we agree with the reviewer that our estimation has some potential uncertainties for the regions with no direct measurement of soil HONO fluxes. Only when such direct measurements emerge, we can compare our estimated soil HONO emissions to the observation to quantify the uncertainty in our estimates. We have added this information in the revised manuscript.

Line 501:

“The potential uncertainties of soil emissions are discussed in the Supplementary

text.”

Supplementary text in revised SI:

“Our study estimates the long-term trends of global soil HONO emissions based on comprehensive dataset of existing global soil HONO emission measurements and quantifies their subsequent impact on global air quality. However, we acknowledge potential uncertainties in our estimates HONO emissions, particularly in regions lacking direct measurements of the soil HONO flux such as in North America. Only when such measurements become available can we compare our estimated emissions with observed values to quantify the uncertainties. Additionally, our estimates use the average emissions from soil samples of different land-use types within specific latitudinal bands and longitudinal columns to represent corresponding regional and land-use type emissions. This approach introduces uncertainty due to heterogeneous emissions across different locations. Moreover, our parameterization does not include factors such as soil pH and texture, adding to the uncertainty in our estimates of HONO emissions. Furthermore, when handling soil emissions post-fertilization soil emissions, we directly used MERRA2 soil moisture data, which may not fully reflect changes in soil moisture due to irrigation, thereby introducing an uncertainty in our estimation of global soil HONO emissions. Despite these limitations, our soil HONO emission estimates consider all controlling factors on which there is consensus, including soil temperature, soil water content, land cover, fertilizer consumption, and soil pH (only for fertilizer-induced soil HONO emission). Therefore, we believe that our estimated global soil HONO emissions have included most (if not all) available information across various regions and represent the state-of-the-art knowledge on soil HONO sources.”

15. Line 369: Oswald et al. (2013) never test at any temperature above 30 °C, so I am not sure why there would not be a plateau in HONO emissions at high T as there is for NO. Please see Stark (1996) and Arroyo et al. (2022) for reasons why soil microbial processes tend to develop an optimum (extremum) as opposed to simply monotonically increasing as an abiotic Arrhenius function. As pointed out by Wang et al. (2021), the use of a higher temperature emission peak led to a net increase of soil NO emissions by almost 20% across North America. It would be helpful to state something about how the uncertainty in the high temperature dependence of your HONO emissions parameterization might influence your overall results.

Response 2.16:

Indeed, some previous field observation studies, e.g., those by Oikawa et al.¹⁷ and Rolle and Aneja¹⁸, have shown that when soil temperature exceeds 30°C, soil emissions may plateau or decrease. However, under ambient conditions, variations in ST are usually accompanied by changes in SWC and other environmental conditions, so the field flux is a result of the combined effects of multiple factors.

In contrast, in the laboratory, we can control the conditions and allow for assessing the singular impact of temperature on emissions. For instance, the laboratory experiments by Ormeci et al.¹⁹ showed a significant exponential increase in soil NO emissions with increasing ST from 1 to 48°C. Similarly, our previous laboratory

experiments also indicated that in the range of 5°C to 55°C¹⁶, soil HONO emissions exhibit an exponential increase with rising temperature. Therefore, we apply this temperature-dependence of soil HONO emissions for the range of 5 to 55°C in the current study.

In the methods section of the revised manuscript, we have added relevant discussions to further clarify this point.

Line 425-434 in the revised manuscript:

“Some field observations^{17,18} have shown that when ST exceeds 30°C, soil emissions may plateau or decrease. However, in field observations, changes in ST are typically accompanied by alterations in SWC and other environmental conditions. In contrast, laboratory experiments have the capability to control variables, thereby enabling a more precise assessment of the isolated impact of ST on emissions. The laboratory experiments by Ormeçi et al.¹⁹ demonstrated a significant exponential increase in soil NO emissions with increasing ST from 1 to 48°C. Similarly, our laboratory experiments also revealed that within the temperature range of 5°C to 55°C, soil HONO emissions exhibit an exponential increase in response to rising ST. Therefore, we adopted an exponential relationship to describe the effect of ST on soil emissions.”

16. Lines 400/401: Why do you assume a linear increase in HONO? Was HONO shown to have a linear increase with fertilization rates in the studies used to develop your parameterization? If so please cite these studies to justify this claim. Another study of soil NO_x emissions by Oikawa et al. (2015) showed that doubling fertilizer inputs of urea fertilizer increased NO_x fluxes by a factor of 5. While this study was not ostensibly measuring HONO, the use of a linear dependence on fertilizer N should be clarified and justified. Moreover, the authors should clarify the influence that each fertilizer type has on HONO emissions and whether a linear correlation has been reported for each fertilizer type.

Response 2.17:

(1) To our best knowledge, the impact of fertilization rates on soil HONO emissions has not been reported. Our parameterization for post-fertilization HONO emissions follows the IPCC guidelines for soil emissions of N₂O estimation⁶⁹ and field measurements of soil NO emissions^{70,71}, which assume that emissions increase linearly with the amount of applied fertilizer. We have clarified this in the revised manuscript.

(2) Wang et al.¹⁶ reported the effects of various fertilizer types on soil HONO emissions and presented a parameterization scheme for estimating these emissions following the application of different fertilizers. In the present study, we employed the respective parameterization from Wang et al.¹⁶ for various fertilizer types.

Line 474-480:

“During this period, the soil emissions after the application of different fertilizers depend on ST and SWC, as reported by Wang et al.¹⁶ We used a linear increase in soil HONO (NO) emissions with rates of different fertilizers, following the IPCC guidelines for estimating soil N₂O emissions and referencing field measurements of soil NO emissions, both of which suggested that emissions rise linearly with the amount of

fertilizer applied. Based on this assumption, the HONO (NO) emission flux of different fertilizers at specific fertilization rates was estimated.”

17. SI Figure 14 – What year is this fertilizer data from? Is it an average of all 36 years? This should be specified in the figure. Is there a description of how fertilizer type influences the HONO emission rates? Or is it assumed to not matter?

Response 2.18:

(1) The data shown in original Figure S14 presents the proportion of different fertilizer types in each country in 2015. We have clarified this in the revised Figure S18 and manuscript.

Line 465-467:

“To consider both the amount and type of fertilizers, we calculated the proportions of different fertilizer types used in each country in 2015 (Supplementary Fig. S18) according to data on fertilizer application amounts provided by the International Fertilizer Association (IFA)²¹.”

(2) As we elaborated in Response 2.17, we considered the influence of fertilizer types on soil emissions by adopting the respective parameterization schemes for different fertilizer types as reported by Wang et al.¹⁶.

Another miscellaneous comment:

18. From Wang et al., *Env. Sci. & Tech.* (2021): “In view of the difficulty of accurately simulating soil moisture during fertilization that typically occurs along with irrigation, we incorporated the average, lowermost and uppermost soil HONO emissions in the SWC range of 60– 100% WHC after using urea (the dominant fertilizer in China) to represent the average case (SoilHONO_avg), the lower limit case (SoilHONO_min), and the upper limit case (SoilHONO_max).” In extending this to a global scale, what method was used to overcome this difficulty in determining the appropriate SWC in the model? This again begs the question of what do you expect the overall uncertainty to be in your modeling results? It would help to discuss in detail potential sources of uncertainty in all components of HONO sources for the sake of transparency so that future work may evolve from here. In fact, it would be best to include a designated section discussing the major uncertainties of this modeling approach.

Response 2.19:

The work we published in 2021 aimed to precisely evaluate the impact of post-fertilization soil HONO emissions on air quality within a week following a certain fertilization event.

Our current work, however, seeks to assess the long-term changes in soil HONO emissions due to alterations in ST and SWC caused by climate change, and fertilization rates. If we use the fixed average, lowermost, and uppermost values of soil HONO emissions, it would not account for the long-term changes in soil conditions that affect nitrogen emissions. Therefore, this study utilizes MERRA2 data to estimate emissions during fertilization periods. We acknowledge that there is uncertainty in our estimates,

and detailed descriptions are provided in the SI text.

Supplementary text in revised SI:

“Furthermore, when handling soil emissions post-fertilization soil emissions, we directly used MERRA2 soil moisture data, which may not fully reflect changes in soil moisture due to irrigation, thereby introducing an uncertainty in our estimation of global soil HONO emissions.”

Response to Reviewer #3

We thank the reviewers for the comments and suggestions (in black font) on the manuscript. Our responses (in blue) and the corresponding edits in the manuscript (in red) are shown below.

Response:

We thank the reviewer and his/her colleague for their constructive feedback, which significantly improves the manuscript.

Response to Reviewer #4

We thank the reviewers for the comments and suggestions (in black font) on the manuscript. Our responses (in blue) and the corresponding edits in the manuscript (in red) are shown below.

Comments:

The manuscript derives the global soil HONO emissions from natural and fertilized lands over the past four decades and evaluate the variation of their impacts on global O₃ and vegetation exposure. The results indicated that climate change and the enhanced fertilizer use increased global soil HONO emissions. Simulations with the updated model showed that soil HONO emissions increased global surface O₃ concentrations and the subsequent risk of vegetation exposure to O₃, especially in crop-production regions. The results are very interesting, and have important meaningful for mitigating global air pollution. The article needs to be improved, then can be considered to publish.

Response 4.1:

We thank the referee for the constructive comments and the suggestions for improvement.

1. HONO emission from soil is also affected by soil microorganisms, soil pH, light and other factors. Did the author take other factors into account when collecting data? At present, only the two parameters of soil temperature and water content are considered. Are these two factors necessarily the most important factors affecting global soil emissions of HONO?

Response 4.2:

We agree with the reviewer that soil HONO emission is influenced by many factors. The dominant controlling factors include soil temperature, soil moisture, nitrogen availability, as suggested by previous studies, e.g., a comprehensive review by Ludwig et al¹².

We would like to clarify that we consider more than just two factors (soil temperature and soil water content) in compiling the global soil HONO emissions.

(1) For the factors on which there is consensus and quantifiable relationship with soil HONO emissions, including fertilizer type, fertilizer consumption and soil pH (only for fertilizer-induced soil HONO emission), we use specific parameterizations.

(2) For the factors on which there is consensus but no quantifiable relationship, e.g., soil nitrogen availability, microorganisms, landuse type, climate factors, etc., we apply different parameterization for each latitude band, longitude band, and landuse type, where possible. For instance, even under no fertilization period, the soil HONO emissions from agricultural lands are still much larger than those from other land types, suggesting the role of soil microorganisms as the reviewer mentioned as well as soil nitrogen availability (or as Reviewer #2 called it, “legacy” nitrogen). This information was included in the original manuscript (Table S5).

(3) For the factors on which there is no consensus, e.g., soil pH for HONO

emissions from natural soil, we do not consider its impacts in our study.

We have added the following descriptions in the revised manuscript:

Line 379-380 in the revised manuscript:

“Our parameterization scheme considers the effects of SWC, ST, and fertilization on soil HONO emissions, as these factors significantly influence emissions from the soil ¹².”

Line 390-395 in the revised manuscript:

“While we are currently unable to accurately quantify the impact of pH on soil HONO emissions, we have established parameterization schemes tailored to specific areas using emission measurement data from various land use types across different regions (see below). This ensures that our parameterization takes into account the three quantifiable variables (SWC, ST, and effects of fertilization), while unquantifiable variables were considered through zoning and land use types.”

2. In Figure 1, Global annual average soil HONO emissions in 2016 are overall higher than HONO emissions in these months in d-g, so which months are the main contributors to the high HONO emissions in a? Is it necessary to show the months with high HONO emissions as well?

Response 4.3:

From the monthly distribution of soil HONO emissions shown in Figure R12, it is evident that the highest emissions occur in July. This peak is attributed to the combined effects of agricultural fertilization and high temperatures in this month in the northern hemisphere. Generally, the northern hemisphere experiences higher emissions from April to September, which coincides with its growing season. The southern hemisphere exhibits higher emissions from October to April.

Figure R12. Monthly variation of soil HONO emissions in 2016.

We have included Figure R12 in the revised SI as Figure S3 and added the following description in the main text:

Line 137-142:

“The globally highest emissions occur in July and peak during the summer seasons (June to August in the northern hemisphere and December to February in the southern hemisphere), while the lowest emissions were observed during the winter (Fig. 1d–g, Supplementary Fig. S3). These seasonal variations were primarily caused by differences in ST throughout the year and disparities in crop growing seasons and fertilizer application timing between the northern and southern hemispheres.”

3. This paper focuses on the situation of soil emission of HONO, then in areas with more exposed soil, there are generally fewer motor vehicles, such as forest areas,

farmland, etc., is it unreasonable to choose a high value of 2.3% for motor vehicle emission factors? In this way, the contribution of soil to HONO will be underestimated.

Response 4.4:

We thank the reviewer for the insightful comment. Indeed, in regions where soil emissions are higher, the number of vehicles is generally lower, such as in forested areas and farmland. Therefore, the impact of vehicle emissions is expected to be not significant where soil emissions are important.

That being said, we used a maximum HONO/NO_x ratio of 2.3% (upper limit in previous studies) to estimate vehicle HONO emissions, so that our calculated impact of soil HONO emissions on atmospheric HONO is at its most conservative value. If we apply a smaller HONO/NO_x ratio for vehicle HONO emission, the impact of soil HONO emissions on air quality would be even more pronounced, as the reviewer suggested. We have added the discussion in the revised manuscript:

Line 515-520 in the revised manuscript:

“In this study, we assumed a HONO/NO_x ratio of 2.3% for all traffic (land vehicle, ship, and aircraft exhausts) HONO emissions. For biomass burning, HONO/NO_x ratios have been reported to range from 0.025 to 0.23⁴⁰⁻⁴². We used the upper bound value of 0.23 for the HONO emissions from biomass burning. The use of the upper limits of these parameters can help that we provide a conservative estimate of the soil HONO impacts compared to these two direct sources.”

4. About RH-dependent and light-enhancing heterogeneous reactions of NO₂ on various surface, at present, there are many new developments for reference, and the author's reference parameterization scheme needs to be further updated.

Response 4.5:

Please note that in the standard CAM-Chem model, there is no HONO production nor loss processes. Our first step in conducting the current study was to include a proper representation of global HONO sources and losses in CAM-Chem model, which itself is a very time consuming and challenging task. Specifically, we have included direct emissions from vehicles and biomass burning, chemical production from homogeneous reaction, heterogeneous reactions, and photolysis. The formulas and parameters used in our study are listed in Table S1. These formulas and parameters have been included in many previous regional studies including those from our group (Zhang et al.³³ and Fu et al.³⁴) and have been shown to provide a reasonable representation of atmospheric HONO on a regional scale.

The reviewer has a point that HONO chemical production, in particular the heterogeneous NO₂ uptake on various surfaces, is a research field that is fast evolving in the past few years^{34,72,73}. However, implementing the most updated HONO chemical production scheme in a global model is not the focus of our work. Instead, we aim to provide the first global soil HONO emissions in the past four decades and quantify soil HONO impacts on global air quality. Our findings remain important regardless of the choices of NO₂ heterogeneous uptake mechanisms.

We have added the latest HONO chemical production parameterization as Table R1 (in response 2.2 to Reviewer #2) in the revised study to inform the readers of these

latest developments.

Line 508-531 in the revised manuscript:

“In the standard CAM-Chem model, there is no HONO production nor loss processes. Here, we incorporated direct emission sources and secondary formation pathways for HONO in the CAM-Chem, including traffic emission, biomass burning emission, homogeneous reaction of NO and OH, RH dependent and light-enhancing heterogeneous reactions of NO₂ on aerosol and ground surfaces, and photolysis of particulate nitrate^{33,34}. The specific reactions and parameters for these sources utilized in the current study are provided in Supplementary Table S7. For HONO emissions from exhaust sources, the HONO to NO_x ratios were found to vary significantly, ranging from 0.29% to 2.3% for vehicle exhausts³⁵⁻³⁸ and from 3% to 6% for commercial aircraft³⁹. In this study, we assumed a HONO/NO_x ratio of 2.3% for all traffic (land vehicle, ship, and aircraft exhausts) HONO emissions. For biomass burning, HONO/NO_x ratios have been reported to range from 0.025 to 0.23⁴⁰⁻⁴². We used the upper bound value of 0.23 for the HONO emissions from biomass burning. The use of the upper limits of these parameters can help that we provide a conservative estimate of the soil HONO impacts compared to these two direct sources. For secondary chemical reactions, the parameters, such as NO₂ uptake coefficient (γ_{NO_2}) and nitrate photolysis rate constant (J_{pNO_3}), exhibit a broad range of variation, with differences in an order of magnitude. The secondary reaction parameters used in our research fall within the range reported previously (Supplementary Table S7) which have been adopted by many prior studies. We did not consider HONO emissions from livestock farming⁴³ nor from biocrusts⁴⁴, because so far there is not sufficient relevant data for deriving global HONO emissions from these two new sources. The HONO absorption cross section used by the CAM-Chem model is based on the values recommended by Burkholder et al.⁴⁵, which is consistent with the recent laboratory measurements of Li et al.⁷² (with difference <1%). Considering that these non-soil HONO emissions/sources are potentially subject to uncertainties, we conducted sensitivity tests to evaluate the effects of selection of lower and upper limits of the non-soil emission sources on soil HONO’s impact on atmospheric chemistry (Supplementary Text).”

5.How reliable is CAM-Chem model? This paper concludes that a large number of simulation results are focused on the model, so the current CAM-Chem model and HONO multi-source parameterization schemes need to be described in more detail. In addition, the parameterization scheme of HONO, including soil emission, NO₂ heterogeneous reaction, vehicle emission and nitrate photolysis, is highly controversial. It is suggested that the author conduct sensitivity analysis on key parameters, explore the changes of important parameters on the amount of HONO emitted by soil, and further evaluate the subsequent impact on O₃.

Response 4.6:

Thank you for your valuable comments.

(1) As mentioned in line 419-423 in the original manuscript (line 531-535 in the

revised manuscript), our study incorporated various sources of HONO into the model, including soil emissions, vehicle emissions, biomass burning emission, gas-phase reactions of NO and OH, RH-dependent and light-enhancing heterogeneous reactions of NO₂, and the photolysis of particle nitrate. Detailed reactions and parameters are listed in Table S7 (shown as Table R1 in response 2.2 to Reviewer #2). We have also modified the corresponding text to clearly state which sources of atmospheric HONO are considered in our study. Please refer to line 529-551 in the revised manuscript for the details.

(2) Indeed, there is considerable variability in the parameters used for HONO sources across different studies. We have conducted a statistical analysis of the range of HONO source parameters (as shown in Table R1) and ran CAM-Chem simulations (both with and without soil HONO) while using the maximum and minimum values of these parameters to assess the impact of different parameter choices on soil emissions' effect on air quality. The results indicate that if we select smaller parameters, the impact of soil HONO emissions on air quality would be even more pronounced than those presented in the current main text (Table R2). To ensure a conservative estimate of the impact of soil HONO emissions, the parameters used in our study, as shown in Table R1, are at their maximum or median values, and we have not used the minimum values to evaluate the impact of soil HONO emissions. We have added the description in the revised SI.

Table R2 Differences in the impact of soil HONO emissions on air quality with different parameter selections. The data listed in the table represent the increase in pollutant concentrations (%) due to soil HONO emissions.

	Species	Global	North America	South America	Europe	Africa	India	China	Australia
Parameters in this study	HONO	302.7	453.8	324.0	331.7	287.8	800.0	106.5	1243.5
	O ₃	2.5	1.8	8.3	2.2	9.0	5.1	2.9	14.5
	OH	7.3	12.4	18.3	14.8	27.1	19.5	8.0	51.9
maximum value of the parameters	HONO	263.9	397.8	306.8	262.6	276.9	652.5	79.0	1213.0
	O ₃	2.5	1.8	8.3	2.2	9.0	5.1	2.9	14.5
	OH	7.3	12.4	18.3	14.8	27.1	19.6	8.0	51.9
Minimum value of the parameter	HONO	1569.1	1986.5	2084.5	1538.2	2115.2	4252.9	482.5	6983.8
	O ₃	2.7	2.4	8.8	2.8	9.6	5.9	2.6	18.4
	OH	8.0	14.9	19.7	16.9	29.7	22.9	7.4	58.1

Line 508-531 in the revised manuscript:

“In the standard CAM-Chem model, there is no HONO production nor loss processes. Here, we incorporated direct emission sources and secondary formation pathways for HONO in the CAM-Chem, including traffic emission, biomass burning emission, homogeneous reaction of NO and OH, RH dependent and light-enhancing heterogeneous reactions of NO₂ on aerosol and ground surfaces, and photolysis of particulate nitrate^{33,34}. The specific reactions and parameters for these sources utilized

in the current study are provided in Supplementary Table S7. For HONO emissions from exhaust sources, the HONO to NO_x ratios were found to vary significantly, ranging from 0.29% to 2.3% for vehicle exhausts³⁵⁻³⁸ and from 3% to 6% for commercial aircraft³⁹. In this study, we assumed a HONO/NO_x ratio of 2.3% for all traffic (land vehicle, ship, and aircraft exhausts) HONO emissions. For biomass burning, HONO/NO_x ratios have been reported to range from 0.025 to 0.23⁴⁰⁻⁴². We used the upper bound value of 0.23 for the HONO emissions from biomass burning. The use of the upper limits of these parameters can help that we provide a conservative estimate of the soil HONO impacts compared to these two direct sources. For secondary chemical reactions, the parameters, such as NO₂ uptake coefficient (γ_{NO_2}) and nitrate photolysis rate constant (J_{pNO_3}), exhibit a broad range of variation, with differences in an order of magnitude. The secondary reaction parameters used in our research fall within the range reported previously (Supplementary Table S7) which have been adopted by many prior studies. We did not consider HONO emissions from livestock farming⁴³ nor from biocrusts⁴⁴, because so far there is not sufficient relevant data for deriving global HONO emissions from these two new sources. The HONO absorption cross section used by the CAM-Chem model is based on the values recommended by Burkholder et al.⁴⁵, which is consistent with the recent laboratory measurements of Li et al.⁷² (with difference <1%)(Supplementary Fig. S19). Considering that these non-soil HONO emissions/sources are potentially subject to uncertainties, we conducted sensitivity tests to evaluate the effects of selection of lower and upper limits of the non-soil emission sources on soil HONO's impact on atmospheric chemistry (Supplementary Text).”

Supplementary text in revised SI:

“In addition to the uncertainties in soil HONO emissions, the production and emission of HONO involve multiple processes and environmental factors, including traffic emissions, biomass burning, and atmospheric chemical reactions. For these sources, the formulas and parameter values used in the estimation process also carry uncertainties. To evaluate the influence of parameter selection for these HONO sources on the impact of soil HONO emissions, we conducted sensitivity tests by using the maximum and minimum values for the parameters (Table S7) of HONO sources. The results (Table S8) indicate that if we select smaller parameters, the impact of soil HONO emissions on air quality would be more pronounced. To ensure a conservative estimate of the impact of soil HONO emissions, the parameters used in our study, as shown in Table S7, are at their maximum or median values, and we have not used the minimum values to evaluate the impact of soil HONO emissions.”

Reference

- 1 Huang, L. *et al.* Nitrous Acid and Nitric Oxide Emissions from Agricultural Soils in

- Guangdong Province: Laboratory Measurement and Emission Estimation. *ACS Earth and Space Chemistry*, doi:10.1021/acsearthspacechem.4c00048 (2024).
- 2 Xue, C. *et al.* Reducing Soil-Emitted Nitrous Acid as a Feasible Strategy for Tackling Ozone Pollution. *Environ Sci Technol* **58**, 9227-9235, doi:10.1021/acs.est.4c01070 (2024).
- 3 Wang, Y. *et al.* Large contribution of nitrous acid to soil-emitted reactive oxidized nitrogen and its effect on air quality. *Environ Sci Technol* **57**, 3516-3526, doi:10.1021/acs.est.2c07793 (2023).
- 4 Meng, F. *et al.* Measurement report: Surface exchange fluxes of HONO during the growth process of paddy fields in the Huaihe River Basin, China. *EGUsphere*, 1-29, doi:10.5194/egusphere-2024-2127 (2024).
- 5 Wu, D. *et al.* Global and regional patterns of soil nitrous acid emissions and their acceleration of rural photochemical reactions. *Journal of Geophysical Research: Atmospheres* **127**, e2021JD036379, doi:10.1029/2021jd036379 (2022).
- 6 Hudman, R. C. *et al.* Steps towards a mechanistic model of global soil nitric oxide emissions: implementation and space based-constraints. *Atmospheric Chemistry and Physics* **12**, 7779-7795, doi:10.5194/acp-12-7779-2012 (2012).
- 7 Yan, X., Ohara, T. & Akimoto, H. Statistical modeling of global soil NO_x emissions. *Global Biogeochemical Cycles* **19**, doi:10.1029/2004gb002276 (2005).
- 8 Vinken, G. C. M., Boersma, K. F., Maasakkers, J. D., Adon, M. & Martin, R. V. Worldwide biogenic soil NO_x emissions inferred from OMI NO₂ observations. *Atmospheric Chemistry and Physics* **14**, 10363-10381, doi:10.5194/acp-14-10363-2014 (2014).
- 9 Yienger, J. J. & Levy II, H. Empirical model of global soil-biogenic NO_x emissions. *J. Geophys. Res* **100**, 11447-11464 (1995).
- 10 Wang, J., Zhang, X., Guo, J., Wang, Z. & Zhang, M. Observation of nitrous acid (HONO) in Beijing, China: Seasonal variation, nocturnal formation and daytime budget. *Sci Total Environ* **587-588**, 350-359, doi:10.1016/j.scitotenv.2017.02.159 (2017).
- 11 Li, D. *et al.* Characteristics and sources of nitrous acid in an urban atmosphere of northern China: Results from 1-yr continuous observations. *Atmospheric Environment* **182**, 296-306, doi:10.1016/j.atmosenv.2018.03.033 (2018).
- 12 Ludwig, J., Meixner, F. X., Vogel, B. & Forstner, J. Soil-air exchange of nitric oxide: An overview of processes, environmental factors, and modeling studies. *Biogeochemistry* **52**, 33 (2001).
- 13 Su, H. *et al.* Soil nitrite as a source of atmospheric HONO and OH radicals. *Science* **333**, p.1616-1618 (2011).
- 14 Oswald, R. *et al.* HONO emissions from soil bacteria as a major source of atmospheric reactive nitrogen. *Science* **341**, 3 (2013).
- 15 Donaldson, M. A., Bish, D. L. & Raff, J. D. Soil surface acidity plays a determining role in the atmospheric-terrestrial exchange of nitrous acid. *Proc Natl Acad Sci U S A* **111**, 18472-18477, doi:10.1073/pnas.1418545112 (2014).
- 16 Wang, Y. *et al.* Agricultural fertilization aggravates air pollution by stimulating soil nitrous acid emissions at high soil moisture. *Environ Sci Technol* **55**, 14556-14566, doi:10.1021/acs.est.1c04134 (2021).
- 17 Oikawa, P. Y. *et al.* Unusually high soil nitrogen oxide emissions influence air quality in a high-temperature agricultural region. *Nat Commun* **6**, 8753, doi:10.1038/ncomms9753

- (2015).
- 18 Paul A. Roelle & Aneja, V. P. Nitric oxide emissions from soils amended with municipal waste biosolids. *Atmospheric Environment* **36**, 137-147 (2002).
- 19 Ormeci, B., Sanin, S. L. & Peirce, J. J. Laboratory study of NO flux from agricultural soil: Effects of soil moisture, pH, and temperature. *Journal of Geophysical Research: Atmospheres* **104**, 1621-1629, doi:10.1029/98jd02834 (1999).
- 20 Meusel, H. *et al.* Emission of nitrous acid from soil and biological soil crusts represents an important source of HONO in the remote atmosphere in Cyprus. *Atmospheric Chemistry and Physics* **18**, 799-813, doi:10.5194/acp-18-799-2018 (2018).
- 21 IFA (International Fertilizer Association). World nitrogen fertilizer consumption. Available at <http://www.fertilizer.org> (accessed 9th June 2024) (1980-2016).
- 22 Houlton, B. Z. *et al.* A world of co-benefits: Solving the global nitrogen challenge. *Earths Future* **7**, 1-8, doi:10.1029/2019EF001222 (2019).
- 23 Liu, X. *et al.* Annual dynamic dataset of global cropping intensity from 2001 to 2019. *Sci Data* **8**, 283, doi:10.1038/s41597-021-01065-9 (2021).
- 24 Bouwman, A. F., Boumans, L. J. M. & Batjes, N. H. Emissions of N₂O and NO from fertilized fields: Summary of available measurement data. *Global Biogeochemical Cycles* **16**, 6-1-6-13 (2002).
- 25 Tian, D., Zhang, Y., Mu, Y., Liu, J. & He, K. Effect of N fertilizer types on N₂O and NO emissions under drip fertigation from an agricultural field in the North China Plain. *Sci Total Environ* **715**, 136903, doi:10.1016/j.scitotenv.2020.136903 (2020).
- 26 Tang, K. *et al.* A dual dynamic chamber system based on IBBCEAS for measuring fluxes of nitrous acid in agricultural fields in the North China Plain. *Atmospheric Environment* **196**, 10-19, doi:10.1016/j.atmosenv.2018.09.059 (2019).
- 27 Xue, C. *et al.* Development and application of a twin open-top chambers method to measure soil HONO emission in the North China Plain. *Science of the Total Environment* **659**, 621-631, doi:10.1016/j.scitotenv.2018.12.245 (2019).
- 28 Xue, C. *et al.* Evidence for strong HONO emission from fertilized agricultural fields and its remarkable impact on regional O₃ pollution in the summer North China Plain. *ACS Earth and Space Chemistry* **5**, 340-347, doi:10.1021/acsearthspacechem.0c00314 (2021).
- 29 Liu, Y. *et al.* A comprehensive model test of the HONO sources constrained to field measurements at rural North China Plain. *Environmental Science & Technology* **53**, 3517-3525, doi:10.1021/acs.est.8b06367 (2019).
- 30 Saiz-Lopez, A. *et al.* Natural short-lived halogens exert an indirect cooling effect on climate. *Nature* **618**, 967-973, doi:10.1038/s41586-023-06119-z (2023).
- 31 Li, Q. *et al.* Global environmental implications of atmospheric methane removal through chlorine-mediated chemistry-climate interactions. *Nat Commun* **14**, 4045, doi:10.1038/s41467-023-39794-7 (2023).
- 32 McDuffie, E. E. *et al.* A global anthropogenic emission inventory of atmospheric pollutants from sector- and fuel-specific sources (1970–2017): an application of the Community Emissions Data System (CEDS). *Earth System Science Data* **12**, 3413-3442, doi:10.5194/essd-12-3413-2020 (2020).
- 33 Zhang, L. *et al.* Potential sources of nitrous acid (HONO) and their impacts on ozone: A WRF-Chem study in a polluted subtropical region. *Journal of Geophysical Research:*

- Atmospheres* **121**, 3645–3662, doi:10.1002/2015jd024468 (2016).
- 34 Fu, X., Wang, T., Zhang, L., Li, Q. & Wang, Z. The significant contribution of HONO to secondary pollutants during a severe winter pollution event in southern China. *Atmospheric Chemistry Physics* **19**, 1–14 (2019).
- 35 Gutzwiller, L., F. Arens, Baltensperger, U., Gäggeler, H. W. & Ammann, M. Significance of semivolatile diesel exhaust organics for secondary HONO formation. *Environ. Sci. Technol.* **36**, 677–682, doi:doi:10.1021/es015673b (2002).
- 36 Xu, Z. *et al.* Nitrous acid (HONO) in a polluted subtropical atmosphere: Seasonal variability, direct vehicle emissions and heterogeneous production at ground surface. *Atmospheric Environment* **106**, 100–109, doi:10.1016/j.atmosenv.2015.01.061 (2015).
- 37 Kurtenbach, R. *et al.* Investigations of emissions and heterogeneous formation of HONO in a road traffic tunnel. *Atmospheric Environment* **35**, 3385–3394 (2001).
- 38 Kirchstetter, T. W., Harley, R. A. & Littlejohn, D. Measurement of nitrous acid in motor vehicle exhaust. *Environ. Sci. Technol.* **30**, 2843–2849, doi:doi:10.1021/es960135y (1996).
- 39 Lee, B. H. *et al.* Measurements of nitrous acid in commercial aircraft exhaust at the Alternative Aviation Fuel Experiment. *Environ Sci Technol* **45**, 7648–7654, doi:10.1021/es200921t (2011).
- 40 Burling, I. R. *et al.* Laboratory measurements of trace gas emissions from biomass burning of fuel types from the southeastern and southwestern United States. *Atmospheric Chemistry and Physics* **10**, 11115–11130, doi:10.5194/acp-10-11115-2010 (2010).
- 41 Akagi, S. K. *et al.* Emission factors for open and domestic biomass burning for use in atmospheric models. *Atmospheric Chemistry and Physics* **11**, 4039–4072, doi:10.5194/acp-11-4039-2011 (2011).
- 42 Keene, W. C. *et al.* Emissions of major gaseous and particulate species during experimental burns of southern African biomass. *Journal of Geophysical Research* **111**, doi:10.1029/2005jd006319 (2006).
- 43 Zhang, Q. *et al.* Unveiling the underestimated direct emissions of nitrous acid (HONO). *Proc Natl Acad Sci U S A* **120**, e2302048120, doi:10.1073/pnas.2302048120 (2023).
- 44 Weber, B., Wu, D., Tamm, A., Ruckteschler, N. & Pschl, U. Biological soil crusts accelerate the nitrogen cycle through large NO and HONO emissions in drylands. *Proceedings of the National Academy of Sciences of the United States of America* **112**, 201515818 (2015).
- 45 Burkholder, J. B. *et al.* Chemical kinetics and photochemical data for use in atmospheric studies; evaluation number 19. *Pasadena, CA: Jet Propulsion Laboratory, National Aeronautics and Space Administration*, 4-74-74-77 (2020).
- 46 Rasool, Q. Z., Bash, J. O. & Cohan, D. S. Mechanistic representation of soil nitrogen emissions in the Community Multiscale Air Quality (CMAQ) model v 5.1. *Geoscientific Model Development* **12**, 849–878, doi:10.5194/gmd-12-849-2019 (2019).
- 47 Luo, L., Ran, L., Rasool, Q. Z. & Cohan, D. S. Integrated Modeling of U.S. Agricultural Soil Emissions of Reactive Nitrogen and Associated Impacts on Air Pollution, Health, and Climate. *Environ Sci Technol* **56**, 9265–9276, doi:10.1021/acs.est.1c08660 (2022).
- 48 Elshorbany, Y. F., Steil, B., Brühl, C. & Lelieveld, J. Impact of HONO on global atmospheric chemistry calculated with an empirical parameterization in the EMAC model. *Atmospheric Chemistry and Physics* **12**, 9977–10000, doi:10.5194/acp-12-9977-2012 (2012).
- 49 Elshorbany, Y. F. *et al.* Global and regional impacts of HONO on the chemical composition

- of clouds and aerosols. *Atmospheric Chemistry and Physics* **14**, 1167-1184, doi:10.5194/acp-14-1167-2014 (2014).
- 50 Ha, P. T. M. *et al.* Implementation of HONO into the chemistry–climate model CHASER (V4.0): roles in tropospheric chemistry. *Geoscientific Model Development* **16**, 927-960, doi:10.5194/gmd-16-927-2023 (2023).
- 51 Xue, C. *et al.* HONO Budget and Its Role in Nitrate Formation in the Rural North China Plain. *Environ Sci Technol* **54**, 11048-11057, doi:10.1021/acs.est.0c01832 (2020).
- 52 Zheng, J. *et al.* Contribution of nitrous acid to the atmospheric oxidation capacity in an industrial zone in the Yangtze River Delta region of China. *Atmospheric Chemistry and Physics* **20**, 5457-5475, doi:10.5194/acp-20-5457-2020 (2020).
- 53 Zhang, X. *et al.* Elucidating HONO formation mechanism and its essential contribution to OH during haze events. *npj Climate and Atmospheric Science* **6**, doi:10.1038/s41612-023-00371-w (2023).
- 54 Xue, C. *et al.* Atmospheric measurements at Mt. Tai – Part II: HONO budget and radical (RO_2 + NO_3) chemistry in the lower boundary layer. *Atmospheric Chemistry and Physics* **22**, 1035-1057, doi:10.5194/acp-22-1035-2022 (2022).
- 55 Gu, R. *et al.* Nitrous acid in the polluted coastal atmosphere of the South China Sea: Ship emissions, budgets, and impacts. *Sci Total Environ* **826**, 153692, doi:10.1016/j.scitotenv.2022.153692 (2022).
- 56 Zhang, W. *et al.* Aging of pollution air parcels acts as the dominant source for nocturnal HONO. *Sci Total Environ* **881**, 163438, doi:10.1016/j.scitotenv.2023.163438 (2023).
- 57 Shi, X. *et al.* Budget of nitrous acid and its impacts on atmospheric oxidative capacity at an urban site in the central Yangtze River Delta region of China. *Atmospheric Environment* **238**, doi:10.1016/j.atmosenv.2020.117725 (2020).
- 58 Zhang, X. *et al.* The Levels and Sources of Nitrous Acid (HONO) in Winter of Beijing and Sanmenxia. *Journal of Geophysical Research: Atmospheres* **127**, doi:10.1029/2021jd036278 (2022).
- 59 Liu, J. *et al.* Detailed budget analysis of HONO in Beijing, China: Implication on atmosphere oxidation capacity in polluted megacity. *Atmospheric Environment* **244**, doi:10.1016/j.atmosenv.2020.117957 (2021).
- 60 Ye, C., Zhang, N., Gao, H. & Zhou, X. Photolysis of particulate nitrate as a source of HONO and NO_x . *Environmental Science & Technology* **51**, 6849-6856, doi:10.1021/acs.est.7b00387 (2017).
- 61 Romer, P. S. *et al.* Constraints on Aerosol Nitrate Photolysis as a Potential Source of HONO and NO_x . *Environ Sci Technol* **52**, 13738-13746, doi:10.1021/acs.est.8b03861 (2018).
- 62 Wu, D. *et al.* Soil HONO emissions at high moisture content are driven by microbial nitrate reduction to nitrite: tackling the HONO puzzle. *The ISME Journal* **13**, 1688-1699, doi:10.1038/s41396-019-0379-y (2019).
- 63 Stutz, J. *et al.* UV-visible absorption cross sections of nitrous acid. *Journal of Geophysical Research: Atmospheres* **105**, 14585-14592, doi:10.1029/2000jd900003 (2000).
- 64 Li, X. *et al.* Revisiting the Ultraviolet Absorption Cross Section of Gaseous Nitrous Acid (HONO): New Insights for Atmospheric HONO Budget. *Environ Sci Technol* **58**, 4247-4256, doi:10.1021/acs.est.3c08339 (2024).

- 65 Mills, G. *et al.* Evidence of widespread effects of ozone on crops and (semi-)natural vegetation in Europe (1990-2006) in relation to AOT40- and flux-based risk maps. *Global Change Biology* **17**, 592-613, doi:10.1111/j.1365-2486.2010.02217.x (2011).
- 66 Wang, Y. *et al.* Reductions in crop yields across China from elevated ozone. *Environ Pollut* **292**, 118218, doi:10.1016/j.envpol.2021.118218 (2022).
- 67 Avnery, S., Mauzerall, D. L., Liu, J. & Horowitz, L. W. Global crop yield reductions due to surface ozone exposure: 1. Year 2000 crop production losses and economic damage. *Atmospheric Environment* **45**, 2284-2296, doi:10.1016/j.atmosenv.2010.11.045 (2011).
- 68 Zhu, X., Burger, M., Doane, T. A. & Horwath, W. R. Ammonia oxidation pathways and nitrifier denitrification are significant sources of N₂O and NO under low oxygen availability. *Proceedings of the National Academy of Sciences of the United States of America* **110**, 6328-6333, doi:10.1073/pnas.1219993110 (2013).
- 69 de Klein C, e. a. I. P. o. C. C. IPCC Guidelines for National Greenhouse Gas Inventories. *Agriculture, Forestry and Other Land Use ((Institute for Global Environmental Strategies, Kanagawa, Japan)* **4**, 11.11-11.54 (2006).
- 70 Zhao, M. *et al.* Mitigating gaseous nitrogen emissions intensity from a Chinese rice cropping system through an improved management practice aimed to close the yield gap. *Agriculture, Ecosystems & Environment* **203**, 36-45, doi:10.1016/j.agee.2015.01.014 (2015).
- 71 Liu, X. J., Mosier, A. R., Halvorson, A. D. & Zhang, F. S. Tillage and Nitrogen Application Effects on Nitrous and Nitric Oxide Emissions from Irrigated Corn Fields. *Plant and Soil* **276**, 235-249, doi:10.1007/s11104-005-4894-4 (2005).
- 72 Zhang, Y. *et al.* Concentration and source changes of nitrous acid (HONO) during the COVID-19 lockdown in Beijing. *Atmospheric Chemistry and Physics* **24**, 8569-8587, doi:10.5194/acp-24-8569-2024 (2024).
- 73 Lu, X., Wang, Y., Li, J., Shen, L. & Fung, J. C. H. Evidence of heterogeneous HONO formation from aerosols and the regional photochemical impact of this HONO source. *Environmental Research Letters* **13**, doi:10.1088/1748-9326/aae492 (2018).